# Asymptotically Unbiased Instance-wise Regularized Partial AUC Optimization: Theory and Algorithm

**Huiyang Shao**[1,2]    **Qianqian Xu**[1*]    **Zhiyong Yang**[2]
**Shilong Bao**[3,4]    **Qingming Huang**[1,2,5,6*]

[1] Key Lab of Intell. Info. Process., Inst. of Comput. Tech., CAS
[2] School of Computer Science and Tech., University of Chinese Academy of Sciences
[3] State Key Lab of Info. Security, Inst. of Info. Engineering, CAS
[4] School of Cyber Security, University of Chinese Academy of Sciences
[5] BDKM, University of Chinese Academy of Sciences
[6] Peng Cheng Laboratory
shaohuiyang21@mails.ucas.ac.cn xuqianqian@ict.ac.cn
yangzhiyong21@ucas.ac.cn baoshilong@iie.ac.cn qmhuang@ucas.ac.cn

## Abstract

The Partial Area Under the ROC Curve (PAUC), typically including One-way Partial AUC (OPAUC) and Two-way Partial AUC (TPAUC), measures the average performance of a binary classifier within a specific false positive rate and/or true positive rate interval, which is a widely adopted measure when decision constraints must be considered. Consequently, PAUC optimization has naturally attracted increasing attention in the machine learning community within the last few years. Nonetheless, most of the existing methods could only optimize PAUC approximately, leading to inevitable biases that are not controllable. Fortunately, a recent work presents an unbiased formulation of the PAUC optimization problem via distributional robust optimization. However, it is based on the pair-wise formulation of AUC, which suffers from the limited scalability w.r.t. sample size and a slow convergence rate, especially for TPAUC. To address this issue, we present a simpler reformulation of the problem in an asymptotically unbiased and instance-wise manner. For both OPAUC and TPAUC, we come to a nonconvex strongly concave minimax regularized problem of instance-wise functions. On top of this, we employ an efficient solver enjoys a linear per-iteration computational complexity w.r.t. the sample size and a time-complexity of $O(\epsilon^{-1/3})$ to reach a $\epsilon$ stationary point. Furthermore, we find that the minimax reformulation also facilitates the theoretical analysis of generalization error as a byproduct. Compared with the existing results, we present new error bounds that are much easier to prove and could deal with hypotheses with real-valued outputs. Finally, extensive experiments on several benchmark datasets demonstrate the effectiveness of our method.

## 1 Introduction

AUC refers to the Area Under the Receiver Operating Characteristic (ROC) curve [1], where the ROC curve is obtained by plotting the True Positive Rate (TPR) against the False Positive Rate (FPR) of a given classifier for all possible thresholds. Since it is insensitive to the class distribution, AUC has become one of the standard metrics for long-tail, and imbalanced datasets [1, 22, 38]. Consequently, AUC optimization has attracted increasing attention in the machine learning community ever since the early 2000s [13, 5, 35, 17]. Over the last two decades, research on AUC optimization has evolved from the simplest linear models and decision trees [27, 10, 29, 41] to state-of-the-art deep

---

*Corresponding authors.

36th Conference on Neural Information Processing Systems (NeurIPS 2022).

Table 1: Comparison with existing partial AUC algorithms. The convergence rate represents the number of iterations after which an algorithm can find an $\epsilon$-stationary point, where $\epsilon$-sp is $\epsilon$-stationary point. $\triangle$ implies a natural result of non-convex SGD. $n_+^B$ ($n_-^B$ resp.) is the number of positive (negative resp.) instances for each mini-batch $B$.

| | SOPA [44] | SOPA-S [44] | TPAUC [39] | Ours |
|---|---|---|---|---|
| Convergence Rate (OPAUC) | $O(\epsilon^{-4})$ | $O(\epsilon^{-4})$ | $O(\epsilon^{-4})^{\triangle}$ | $O(\epsilon^{-3})$ |
| Convergence Rate (TPAUC) | $O(\epsilon^{-6})$ | $O(\epsilon^{-4})$ | $O(\epsilon^{-4})^{\triangle}$ | $O(\epsilon^{-3})$ |
| Convergence Measure | $\epsilon$-sp (non-smooth) | $\epsilon$-sp | $\epsilon$-sp | $\epsilon$-sp |
| Smoothness | non-smooth | smooth | smooth | smooth |
| Unbiasedness | $\checkmark$ | $\times$ | $\times$ | with bias $O(1/\kappa)$ when $\omega = 0$ |
| Per-Iteration Time Complexity | $O(n_+^B n_-^B)$ | $O(n_+^B n_-^B)$ | $O(n_+^B n_-^B)$ | $O(n_+^B + n_-^B)$ |

learning architectures [21, 14, 37, 43, 42, 33]. With such remarkable success, one can now easily apply AUC optimization to deal with various real-world problems ranging from financial fraud detection[16, 4, 23], spam email detection [24], to medical diagnosis [24, 38, 37, 43], etc.

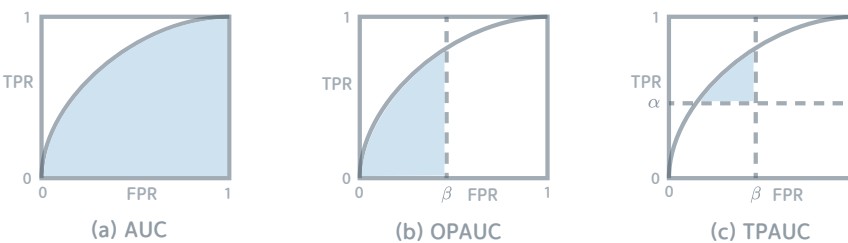

(a) AUC  (b) OPAUC  (c) TPAUC

Figure 1: The comparison among different AUC variants. (a) The area under the ROC curve (AUC). (b) The one-way partial AUC (OPAUC). (c) The two-way partial AUC (TPAUC).

However, in such long-tailed applications, we are often interested in a specific region in the ROC curve where its area is called Partial AUC (PAUC). As illustrated in Fig.1, there are two types of PAUC. Here, One-way Partial AUC (OPAUC) measures the area within an FPR interval ( $0 \leq \text{FPR} \leq \beta$); while Two-way Partial AUC (TPAUC) measures the area with $\text{FPR} \leq \beta$, $\text{TPR} \geq \alpha$. Unlike the full AUC, optimizing PAUC requires selecting top-ranked or/and bottom-ranked instances, leading to a hard combinatorial optimization problem. Many efforts have been made to solve the problem [6, 24, 25, 20, 39]. However, the majority of them rely on full-batch optimization and the approximation of the top (bottom)-$k$ ranking process, which suffers from immeasurable biases and undesirable efficiency. Most recently, researchers have started to explore mini-batch PAUC optimization for deep learning models. [39] proposed a novel end-to-end optimization framework for PAUC. This formulation has a fast convergence rate with the help of a stochastic optimization algorithm, but the estimation of PAUC is still biased. Later, [44] proposed a Distributional Robust Optimization (DRO) framework for PAUC optimization, where the bias can be eliminated by a clever reformulation and the compositional SGD algorithms [28]. However, they adopt the pair-wise loss function which has limited scalability w.r.t. sample size and $O(\epsilon^{-4})/O(\epsilon^{-6})$ time complexity to reach the $\epsilon$-stationary point for OPAUC/TPAUC. Considering the efficiency bottleneck comes from the pair-wise formulation, we will explore the following question in this paper:

> ***Can we design a simpler, nearly asymptotically unbiased and instance-wise formulation to optimize OPAUC and TPAUC in an efficient way?***

To answer this question, we propose an efficient and nearly unbiased optimization algorithm (the bias vanishes asymptotically when $\kappa \to 0$) for regularized PAUC maximization with a faster convergence guarantee. The comparison with previous results are listed in Tab.1. We consider both OPAUC and TPAUC maximization, where for OPAUC, we focus on maximizing PAUC in the region: $\text{FPR} \leq \beta$ and for TPAUC we focus on the region: $\text{FPR} \leq \beta$ and $\text{TPR} \geq \alpha$. We summarize our contributions below.

- With a proper regularization, we propose a nonconvex strongly concave minimax instance-wise formulation for OPAUC and TPAUC maximization. On top of our proposed formulation, we employ an efficient stochastic minimax algorithm that finds a $\epsilon$-stationary point within $O(\epsilon^{-3})$ iterations.

- We conduct a generalization analysis of our proposed methods. Our instance-wise reformulation can overcome the interdependent issue of the original pair-wise generalization analysis. The proof is much easier than the existing results. Moreover, compared with [25, 39], it can be applied to any real-valued hypothesis functions other than the hard-threshold functions.

- We conduct extensive experiments on multiple imbalanced image classification tasks. The experimental results speak to the effectiveness of our proposed methods.

## 2 Preliminaries

**Notations.** In this section, we give some definitions and preliminaries about OPAUC and TPAUC. Let $\mathcal{X} \subseteq \mathbb{R}^d$ be the input space, $\mathcal{Y} = \{0, 1\}$ be the label space. We denote $\mathcal{D}_\mathcal{P}$ and $\mathcal{D}_\mathcal{N}$ as positive and negative instance distribution, respectively. Let $S = \{\boldsymbol{z} = (\boldsymbol{x}_i, y_i)\}_{i=1}^n$ be a set of training data drawn from distribution $\mathcal{D}_\mathcal{Z}$, where $n$ is the number of samples. Let $\mathcal{P}$ ($\mathcal{N}$ resp.) be a set of positive (negative resp.) instances in the dataset. **In this paper we only focus on the scoring functions** $f : \mathcal{X} \mapsto [0, 1]$. The output range can be simply implemented by any deep neural network with sigmoid outputs.

**Standard AUC.** The standard AUC calculates the entire area under the ROC curve. For mathematical convenience, our arguments start with the pair-wise reformulation of AUC. Specifically, as shown in [1], AUC measures the probability of a positive instance having a higher score than a negative instance:

$$\text{AUC}(f) = \Pr_{\boldsymbol{x} \sim \mathcal{D}_\mathcal{P}, \boldsymbol{x}' \sim \mathcal{D}_\mathcal{N}} [f(\boldsymbol{x}) > f(\boldsymbol{x}')]. \tag{1}$$

**OPAUC.** As mentioned in the introduction, instead of considering the entire region of ROC, we focus on two forms of PAUC, namely TPAUC and OPAUC. According to [6], OPAUC is equivalent to the probability of a positive instance $\boldsymbol{x}$ being scored higher than a negative instance $\boldsymbol{x}'$ within the specific range $f(\boldsymbol{x}') \in [\eta_\beta(f), 1]$ $s.t.$ $\Pr_{\boldsymbol{x}' \sim \mathcal{D}_\mathcal{N}}[f(\boldsymbol{x}') \geq \eta_\beta] = \beta$:

$$\text{OPAUC}(f) = \Pr_{\boldsymbol{x} \sim \mathcal{D}_\mathcal{P}, \boldsymbol{x}' \sim \mathcal{D}_\mathcal{N}} [f(\boldsymbol{x}) > f(\boldsymbol{x}'), f(\boldsymbol{x}') \geq \eta_\beta(f)]. \tag{2}$$

Practically, we do not know the exact data distributions $\mathcal{D}_\mathcal{P}, \mathcal{D}_\mathcal{N}$ to calculate Eq.(2). Therefore, we turn to the empirical estimation of Eq.(2). Given a finite dataset $S$ with $n$ instances, let $n_+, n_-$ be the numbers of positive/negative instances, respectively. For the OPAUC, its empirical estimation could be expressed as [24]:

$$\hat{\text{AUC}}_\beta(f, S) = 1 - \sum_{i=1}^{n_+} \sum_{j=1}^{n_-^\beta} \frac{\ell_{0,1}\Big(f(\boldsymbol{x}_i) - f(\boldsymbol{x}'_{[j]})\Big)}{n_+ n_-^\beta}, \tag{3}$$

where $n_-^\beta = \lfloor n_- \cdot \beta \rfloor$; $\boldsymbol{x}'_{[j]}$ denotes the $j$-th largest score among negative samples; $\ell_{0,1}(t)$ is the $0 - 1$ loss, which returns 1 if $t < 0$ and 0 otherwise.

**TPAUC.** More recently, [36] argued that an efficient classifier should have low FPR and high TPR simultaneously. Therefore, we also study a more general variant called Two-way Partial AUC (TPAUC), where the restricted regions satisfy TPR $\geq \alpha$ and FPR $\leq \beta$. Similar to OPAUC, TPAUC measures the probability that a positive instance $\boldsymbol{x}$ ranks higher than a negative instance $\boldsymbol{x}'$ where $f(\boldsymbol{x}) \in [0, \eta_\alpha(f)]$, $s.t.$ $\Pr_{\boldsymbol{x} \sim \mathcal{D}_\mathcal{P}}[f(\boldsymbol{x}) \leq \eta_\alpha] = \alpha$, $f(\boldsymbol{x}') \in [\eta_\beta(f), 1]$ $s.t.$ $\Pr_{\boldsymbol{x}' \sim \mathcal{D}_\mathcal{N}}[f(\boldsymbol{x}') \geq \eta_\beta] = \beta$.

$$\text{TPAUC}(f) = \Pr_{\boldsymbol{x} \sim \mathcal{D}_\mathcal{P}, \boldsymbol{x}' \sim \mathcal{D}_\mathcal{N}} [f(\boldsymbol{x}) > f(\boldsymbol{x}'), f(\boldsymbol{x}) \leq \eta_\alpha(f), f(\boldsymbol{x}') \geq \eta_\beta(f)]. \tag{4}$$

Similarly to OPAUC, for the TPAUC, we adopt its empirical estimation [36, 39]:

$$\hat{\text{AUC}}_{\alpha,\beta}(f, S) = 1 - \sum_{i=1}^{n_+^\alpha} \sum_{j=1}^{n_-^\beta} \frac{\ell_{0,1}\Big(f(\boldsymbol{x}_{[i]}) - f(\boldsymbol{x}'_{[j]})\Big)}{n_+^\alpha n_-^\beta}, \tag{5}$$

where $n_+^\alpha = \lfloor n_+ \cdot \alpha \rfloor$ and $\boldsymbol{x}_{[i]}$ is $i$-th smallest score among all positive instances.

# 3 Problem Formulation

In this section, we introduce how to optimize OPAUC and TPAUC in an asymptotically unbiased instance-wise manner. Note that Eq.(3) and Eq.(5) are hard to optimize since it is complicated to determine the positive quantile function $\eta_\alpha(f)$ and the negative quantile function $\eta_\beta(f)$. So we can not obtain the bottom-ranked positive instances and top-ranked negative instances directly. In this section, we will elaborate on how to tackle these challenges.

## 3.1 Optimizing the OPAUC

According to Eq.(3), given a surrogate loss $\ell$ and the finite dataset $S$, maximizing OPAUC and $\hat{\text{AUC}}_\beta(f, S)$ is equivalent to solving the following problems, respectively:

$$\min_f \mathcal{R}_\beta(f) = \mathbb{E}_{\boldsymbol{x}\sim\mathcal{D}_\mathcal{P},\boldsymbol{x}'\sim\mathcal{D}_\mathcal{N}} \left[ \mathbb{I}_{f(\boldsymbol{x}')\geq\eta_\beta(f)} \cdot \ell(f(\boldsymbol{x}) - f(\boldsymbol{x}')) \right], \tag{6}$$

$$\min_f \hat{\mathcal{R}}_\beta(f, S) = \sum_{i=1}^{n_+} \sum_{j=1}^{n_-^\beta} \frac{\ell\left(f(\boldsymbol{x}_i) - f(\boldsymbol{x}'_{[j]})\right)}{n_+ n_-^\beta}. \tag{7}$$

**Step 1: Instance-wise Reformulation.** Here, to simplify the reformulation, we will use the most popular **surrogate squared loss** $\ell(x) = (1-x)^2$. Under this setting, the following theorem shows an instance-wise reformulation of the OPAUC optimization problem (please see Appendix.F for the proof):

**Theorem 1.** *Assuming that $f(\boldsymbol{x}) \in [0,1]$, $\forall \boldsymbol{x} \in \mathcal{X}$, $F_{op}(f, a, b, \gamma, t, \boldsymbol{z})$ is defined as:*

$$F_{op}(f, a, b, \gamma, t, \boldsymbol{z}) = [(f(\boldsymbol{x}) - a)^2 - 2(1+\gamma)f(\boldsymbol{x})]y/p - \gamma^2 \\ [(f(\boldsymbol{x}) - b)^2 + 2(1+\gamma)f(\boldsymbol{x})] \cdot [(1-y)\mathbb{I}_{f(\boldsymbol{x})\geq t}]/[(1-p)\beta], \tag{8}$$

*where $y = 1$ for positive instances, $y = 0$ for negative instances and we have the following conclusions:*

*(a) (**Population Version**.) We have:*

$$\min_f \mathcal{R}_\beta(f) \Leftrightarrow \min_{f,(a,b)\in[0,1]^2} \max_{\gamma\in[-1,1]} \mathbb{E}_{\boldsymbol{z}\sim\mathcal{D}_\mathcal{Z}} \left[ F_{op}(f, a, b, \gamma, \eta_\beta(f), \boldsymbol{z}) \right], \tag{9}$$

*where $\eta_\beta(f) = \arg\min_{\eta_\beta\in\mathbb{R}} \left[ \mathbb{E}_{\boldsymbol{x}'\sim\mathcal{D}_\mathcal{N}}[\mathbb{I}_{f(\boldsymbol{x}')\geq \eta_\beta}] = \beta \right]$.*

*(b) (**Empirical Version**.) Moreover, given a training dataset $S$ with sample size $n$, denote:*

$$\hat{\mathbb{E}}_{\boldsymbol{z}\sim S}[F_{op}(f, a, b, \gamma, \hat{\eta}_\beta(f), \boldsymbol{z})] = \frac{1}{n} \sum_{i=1}^n F_{op}(f, a, b, \gamma, \hat{\eta}_\beta(f), \boldsymbol{z}_i),$$

*where $\hat{\eta}_\beta(f)$ is the empirical quantile of the negative instances in $S$. We have:*

$$\min_f \hat{\mathcal{R}}_\beta(f, S) \Leftrightarrow \min_{f,(a,b)\in[0,1]^2} \max_{\gamma\in[-1,1]} \hat{\mathbb{E}}_{\boldsymbol{z}\sim S} \left[ F_{op}(f, a, b, \gamma, \hat{\eta}_\beta(f), \boldsymbol{z}) \right], \tag{10}$$

**Step 2: Differentiable Sample Selection.** Thm.1 provides a support to convert the pair-wise loss into instance-wise loss for OPAUC. However, the minimax problem Eq.(10) is still difficult to solve due to the operation $\mathbb{I}_{f(\boldsymbol{x}')\geq\eta_\beta(f)}$, which requires selecting top-ranked negative instances. To make the sample selection process differentiable, we adopt the following lemma.

**Lemma 1.** $\sum_{i=1}^k x_{[i]}$ *is a convex function of* $(x_1, \cdots, x_n)$ *where* $x_{[i]}$ *is the top-i element of a set* $\{x_1, x_2, \cdots, x_n\}$. *Furthermore, for* $x_i, i = 1, \cdots, n$, *we have* $\frac{1}{k}\sum_{i=1}^k x_{[i]} = \min_s\{s + \frac{1}{k}\sum_{i=1}^n[x_i - s]_+\}$, *where* $[a]_+ = \max\{0, a\}$. *The population version is* $\mathbb{E}_x[x \cdot \mathbb{I}_{x\geq\eta(\alpha)}] = \min_s \frac{1}{\alpha}\mathbb{E}_x[\alpha s + [x - s]_+]$, *where* $\eta(\alpha) = \arg\min_{\eta\in\mathbb{R}}[\mathbb{E}_x[\mathbb{I}_{x\geq\eta}] = \alpha]$ *(please see Appendix.F for the proof).*

Lem.1 proposes an Average Top-$k$ (ATk) loss which is the surrogate loss for top-$k$ loss to eliminate the sorting problem. Optimizing the ATk loss is equivalent to selecting top-ranked instances. Actually, for $\mathcal{R}_\beta(f)$, we can just reformulate it as an Average Top-$k$ (ATk) loss. Denote $\ell_-(\boldsymbol{x}') = (f(\boldsymbol{x}') - b)^2 + 2(1+\gamma)f(\boldsymbol{x}')$. In the proof of the next theorem, we will show that $\ell_-(\boldsymbol{x}')$ is an increasing function w.r.t. $f(\boldsymbol{x}')$, namely:

$$\mathbb{E}_{\boldsymbol{x}'\sim\mathcal{D}_\mathcal{N}}[\mathbb{I}_{f(\boldsymbol{x}')\geq\eta_\beta(f)}\cdot\ell_-(\boldsymbol{x}')|f(\boldsymbol{x}')\geq\eta_\beta(f))] = \min_s \frac{1}{\beta}\cdot\mathbb{E}_{\boldsymbol{x}'\sim\mathcal{D}_\mathcal{N}}[\beta s + [\ell_-(\boldsymbol{x}') - s]_+]. \quad (11)$$

The similar result holds for $\hat{\mathcal{R}}_\beta(f, S)$. Then, we can reach to Thm.2 (please see Appendix.F for the proof):

**Theorem 2.** *Assuming that $f(\boldsymbol{x}) \in [0, 1]$, for all $\boldsymbol{x} \in \mathcal{X}$, we have the equivalent optimization for OPAUC:*

$$\min_{f,(a,b)\in[0,1]^2} \max_{\gamma\in[-1,1]} \mathbb{E}_{\boldsymbol{z}\sim\mathcal{D}_\mathcal{Z}}[F_{op}(f,a,b,\gamma,\eta_\beta(f),\boldsymbol{z})]$$
$$\Leftrightarrow \min_{f,(a,b)\in[0,1]^2} \max_{\gamma\in\Omega_\gamma} \min_{s'\in\Omega_{s'}} \mathbb{E}_{\boldsymbol{z}\sim\mathcal{D}_\mathcal{Z}}[G_{op}(f,a,b,\gamma,\boldsymbol{z},s')], \quad (12)$$

$$\min_{f,(a,b)\in[0,1]^2} \max_{\gamma\in[-1,1]} \hat{\mathbb{E}}_{\boldsymbol{z}\sim S}[F_{op}(f,a,b,\gamma,\hat{\eta}_\beta(f),\boldsymbol{z})]$$
$$\Leftrightarrow \min_{f,(a,b)\in[0,1]^2} \max_{\gamma\in\Omega_\gamma} \min_{s'\in\Omega_{s'}} \hat{\mathbb{E}}_{\boldsymbol{z}\sim S}[G_{op}(f,a,b,\gamma,\boldsymbol{z},s')], \quad (13)$$

*where $\Omega_\gamma = [b-1, 1]$, $\Omega_{s'} = [0, 5]$ and*

$$G_{op}(f,a,b,\gamma,\boldsymbol{z},s') = [(f(\boldsymbol{x}) - a)^2 - 2(1+\gamma)f(\boldsymbol{x})]y/p - \gamma^2$$
$$+ \left(\beta s' + [(f(\boldsymbol{x}) - b)^2 + 2(1+\gamma)f(\boldsymbol{x}) - s']_+\right)(1-y)/[\beta(1-p)]. \quad (14)$$

**Step 3: Asymptotically Unbiased Smoothing.** Even with Thm.2, it is hard to optimize the min-max-min formulation in Eq.(13). A solution is to swap the order $\max_\gamma$ and $\min_{s'}$ to reformulate it as a min-max problem. The key obstacle to this idea is the non-smooth function $[\cdot]_+$. To avoid the $[\cdot]_+$, we apply the `softplus` function [12]:

$$r_\kappa(x) = \frac{\log(1 + \exp(\kappa\cdot x))}{\kappa}, \quad (15)$$

as a smooth surrogate. It is easy to show that $r_\kappa(x) \overset{\kappa\to\infty}{\to} [x]_+$. **Denote $G_{op}^\kappa(f,a,b,\gamma,\boldsymbol{z},s')$ the surrogate objective where the $[\cdot]_+$ in $G_{op}(f,a,b,\gamma,\boldsymbol{z},s')$ is replaced with $r_\kappa(\cdot)$.** We then proceed to solve the surrogate problem:

$$\min_{f,(a,b)\in[0,1]^2} \max_{\gamma\in\Omega_\gamma} \min_{s'\in\Omega_{s'}} \mathbb{E}_{\boldsymbol{z}\sim\mathcal{D}_\mathcal{Z}}[G_{op}^\kappa(f,a,b,\gamma,\boldsymbol{z},s')]$$
$$\min_{f,(a,b)\in[0,1]^2} \max_{\gamma\in\Omega_\gamma} \min_{s'\in\Omega_{s'}} \hat{\mathbb{E}}_{\boldsymbol{z}\sim S}[G_{op}^\kappa(f,a,b,\gamma,\boldsymbol{z},s')], \quad (16)$$

respectively for the population and empirical version. In Appendix.B, we will proof that such a approximation has a convergence rate $O(1/\kappa)$.

**Step 4: The Regularized Problem.** It is easy to check that $r_\kappa(x)$ has a bounded second-order derivation. In this way, we can regard $G_{op}^\kappa(f,a,b,\gamma,\boldsymbol{z},s')$ as a weakly-concave function [3] of $\gamma$. By employing an $\ell_2$ regularization, we turn to a regularized form:

$$G_{op}^{\kappa,\omega}(f,a,b,\gamma,\boldsymbol{z},s') = G_{op}^\kappa(f,a,b,\gamma,\boldsymbol{z},s') - \omega\cdot\gamma^2,$$

With a sufficiently large $\omega$, $G_{op}^{\kappa,\omega}(f,a,b,\gamma,\boldsymbol{z},s')$ is strongly-concave w.r.t. $\gamma$ when all the other variables are fixed. Note that the regularization scheme will inevitably bias. As a very general result, regularization will inevitably induce bias. However, it is known to be a necessary building block to stabilize the solutions and improve generalization performance. We then reach a minimax problem in the final step.

**Step 5: Min-Max Swapping.** According to min-max theorem [3], if we replace $G_{op}^\kappa(f,a,b,\gamma,\boldsymbol{z},s')$ with $G_{op}^{\kappa,\omega}(f,a,b,\gamma,\boldsymbol{z},s')$, the surrogate optimization problem satisfies:

$$\min_{f,(a,b)\in[0,1]^2} \max_{\gamma\in\Omega_\gamma} \min_{s'\in\Omega_{s'}} \mathbb{E}_{\boldsymbol{z}\sim\mathcal{D}_\mathcal{Z}}[G_{op}^{\kappa,\omega}] \Leftrightarrow \min_{f,(a,b)\in[0,1]^2,s'\in\Omega_{s'}} \max_{\gamma\in\Omega_\gamma} \mathbb{E}_{\boldsymbol{z}\sim\mathcal{D}_\mathcal{Z}}[G_{op}^{\kappa,\omega}], \quad (17)$$

$$\min_{f,(a,b)\in[0,1]^2} \max_{\gamma\in\Omega_\gamma} \min_{s'\in\Omega_{s'}} \hat{\mathbb{E}}_{z\sim S}[G_{op}^{\kappa,\omega}] \Leftrightarrow \min_{f,(a,b)\in[0,1]^2,s'\in\Omega_{s'}} \max_{\gamma\in\Omega_\gamma} \hat{\mathbb{E}}_{z\sim S}[G_{op}^{\kappa,\omega}], \tag{18}$$

where $G_{op}^{\kappa,\omega} = G_{op}^{\kappa,\omega}(f,a,b,\gamma,z,s')$. In this sense, we come to a regularized non-convex strongly-concave problem. In Sec.4, we will employ an efficient solver to optimize the parameters.

### 3.2 Optimizing the TPAUC

According to Eq.(5), given a surrogate loss $\ell$ and finite dataset $S$, maximizing TPAUC and $\hat{\mathrm{AUC}}_{\alpha,\beta}(f,S)$ is equivalent to solving the following problems, respectively:

$$\min_f \mathcal{R}_{\alpha,\beta}(f) = \mathbb{E}_{\boldsymbol{x}\sim\mathcal{D}_\mathcal{P},\boldsymbol{x}'\sim\mathcal{D}_\mathcal{N}}\left[\mathbb{I}_{f(\boldsymbol{x}')\geq\eta_\beta(f)}\cdot\mathbb{I}_{f(\boldsymbol{x})\leq\eta_\alpha(f)}\cdot\ell(f(\boldsymbol{x})-f(\boldsymbol{x}'))\right]. \tag{19}$$

$$\min_f \hat{\mathcal{R}}_{\alpha,\beta}(f,S) = \sum_{i=1}^{n_+^\alpha}\sum_{j=1}^{n_-^\beta} \frac{\ell\left(f(\boldsymbol{x}_{[i]})-f(\boldsymbol{x}'_{[j]})\right)}{n_+^\alpha n_-^\beta}. \tag{20}$$

Due to the limited space, we present the result directly, please refer to Appendix.D for more details.

Similar to OPAUC, we apply the function $r_\kappa(x)$, regularization $\omega\gamma^2$ and min-max theorem to solve the problem. In this sense, we can use

$$\min_{f,(a,b)\in[0,1]^2,s\in\Omega_s,s'\in\Omega_{s'}} \max_{\gamma\in\Omega_\gamma} \mathbb{E}_{z\sim\mathcal{D}_\mathcal{Z}}\left[G_{tp}^{\kappa,\omega}(f,a,b,\gamma,z,s,s')\right], \tag{21}$$

where $\Omega_\gamma = [\max\{-a,b-1\},1]$ and

$$\min_{f,(a,b)\in[0,1]^2,s\in\Omega_s,s'\in\Omega_{s'}} \max_{\gamma\in\Omega_\gamma} \hat{\mathbb{E}}_{z\sim S}\left[G_{tp}^{\kappa,\omega}(f,a,b,\gamma,z,s,s')\right], \tag{22}$$

to minimize $\mathcal{R}_{\alpha,\beta}(f)$, and $\hat{\mathcal{R}}_{\alpha,\beta}(f)$, respectively. Here:

$$\begin{aligned}
G_{tp}^{\kappa,\omega}(f,a,b,\gamma,z,s,s') &= \left(\alpha s + r_\kappa\left((f(\boldsymbol{x})-a)^2 - 2(1+\gamma)f(\boldsymbol{x})-s\right)\right)y/(\alpha p) - (\omega+1)\gamma^2 \\
&\quad + \left(\beta s' + r_\kappa\left((f(\boldsymbol{x})-b)^2 + 2(1+\gamma)f(\boldsymbol{x})-s'\right)\right)(1-y)/[\beta(1-p)].
\end{aligned} \tag{23}$$

According to Thm.2 of [32], we have the following corollary:

**Corollary 1.** *We can reformulate Eq.*(18) *and Eq.*(23) *as an off-the-shelf minimax problem where the coupled constraint is replaced with the Lagrange multipliers ($\theta_b$ for OPAUC, $\theta_b,\theta_a$ for TPAUC). For* OPAUC*:*

$$\min_{f,(a,b)\in[0,1]^2,s\in\Omega_s} \max_{\gamma\in[b-1,1]} \mathbb{E}_{z\sim\mathcal{D}_\mathcal{Z}}[G_{op}^{\kappa,\omega}] \Leftrightarrow \min_{f,(a,b)\in[0,1]^2,s\in\Omega_s,\theta_b\in[0,M_1]} \max_{\gamma\in[-1,1]} \mathbb{E}_{z\sim\mathcal{D}_\mathcal{Z}}[G_{op}^{\kappa,\omega}]. \\ - \theta_b(b-1-\gamma) \tag{24}$$

*For* TPAUC*:*

$$\begin{aligned}
&\min_{f,(a,b)\in[0,1]^2,s\in\Omega_s,s'\in\Omega_{s'}} \max_{\gamma\in[\max\{-a,b-1\},1]} \mathbb{E}_{z\sim\mathcal{D}_\mathcal{Z}}[G_{tp}^{\kappa,\omega}] \\
\Leftrightarrow &\min_{f,(a,b)\in[0,1]^2,s\in\Omega_s,s'\in\Omega_{s'},\theta_a\in[0,M_2],\theta_b\in[0,M_3]} \max_{\gamma\in[-1,1]} \mathbb{E}_{z\sim\mathcal{D}_\mathcal{Z}}[G_{tp}^{\kappa,\omega}] \\
&- \theta_b(b-1-\gamma) - \theta_a(-a-\gamma).
\end{aligned} \tag{25}$$

The tight constraint $\theta_b\in[0,M_1]/\theta_b\in[0,M_2], \theta_a\in[0,M_3]$ comes from the fact that optimum $\theta_b,\theta_a$ are both finite since the objective function is bounded from above. In the experiments, to make sure that $M_1,M_2,M_3$ are large enough, we set them as $M_1 = M_2 = M_3 = 10^9$.

## 4 Training Algorithm

According to the derivations in the previous sections, our goal is then to solve the resulting empirical minimax optimization problems in Eq.(18) and Eq.(22). It is easy to check that they are strongly-concave w.r.t $\gamma$ whenever $\kappa \leq 2+2\omega$, when $(f(\boldsymbol{x}),a,b)\in[0,1]^3, \gamma\in[-1,1], s\in\Omega_s, s'\in\Omega_{s'}$.

Therefore, we can adopt the nonconvex strongly concave minimax optimization algorithms to solve these problems [15]. In this section, following the work [15], we employ an accelerated stochastic gradient descent ascent (ASGDA) method to solve the minimax optimization problem. We denote $\boldsymbol{\theta} \in \mathbb{R}^d$ as the parameters of function $f$, $\boldsymbol{\tau} = \{\boldsymbol{\theta}, a, b, s, s', \theta_a, \theta_b\} \in \Omega_\tau$ as the variables for the outer min-problem. Alg.1 shows the framework of our algorithm (we adopt the accelerated algorithm in [15] to solve our problem). There are two key steps. At Line 5-6 of Algorithm 1, variables $\boldsymbol{\tau}_{t+1}$ and $\gamma_{t+1}$ are updated in a momentum way. Moreover, the convex combination ensures that they are always feasible given that the initial solution is feasible. At Line 9-10, using the momentum-based variance reduced technique, we can estimate the stochastic first-order partial gradients $\boldsymbol{v}_t$ and $w_t$ in a more stable manner.

---

**Algorithm 1** Accelerated Stochastic Gradient Descent Ascent Algorithm

---

1: **Input**: Dataset $\mathcal{X}$, learning parameters $\{\nu, \lambda, k, m, c_1, c_2, T\}$
2: **Initialize:** Randomly select $\boldsymbol{\tau}_0 = \{\boldsymbol{\theta}_0, a_0, b_0, s_0, s'_0, \theta_a, \theta_b\}$ from $\Omega_\tau$, $\boldsymbol{v}_0 = \mathbf{0}^{d+6}$, $w_0 = 0$.
       Randomly select $\gamma_0$ from $\Omega_\gamma$, $t = 0$,
3: **for** $t = 0, 1, \cdots, T$ **do**
4:    Compute the learning rate $\eta_t = \frac{k}{(m+t)^{1/3}}$;
5:    Update $\boldsymbol{\tau}_{t+1} = (1 - \eta_t)\boldsymbol{\tau}_t + \eta_t \mathcal{P}_{\Omega_\tau}(\boldsymbol{\tau}_t - \nu \boldsymbol{v}_t)$;
6:    Update $\gamma_{t+1} = (1 - \eta_t)\gamma_t + \eta_t \mathcal{P}_{\Omega_\gamma}(\gamma_t + \lambda w_t)$;
7:    Compute $\rho_{t+1} = c_1 \eta_t^2$ and $\xi_{t+1} = c_2 \eta_t^2$;
8:    Sampling mini-batch data $\mathcal{B}_{t+1}$ from dataset $\mathcal{X}$;
9:    Update $\boldsymbol{v}_{t+1} = \nabla_{\boldsymbol{\tau}} G_{(\cdot)}^{\kappa,\omega}(\boldsymbol{\tau}_{t+1}, \gamma_{t+1}; \mathcal{B}_{t+1}) + (1 - \rho_{t+1})[\boldsymbol{v}_t - \nabla_{\boldsymbol{\tau}} G_{(\cdot)}^{\kappa,\omega}(\boldsymbol{\tau}_t, \gamma_t, \mathcal{B}_{t+1})]$;
10:   Update $w_{t+1} = \nabla_\gamma G_{(\cdot)}^{\kappa,\omega}(\boldsymbol{\tau}_{t+1}, \gamma_{t+1}; \mathcal{B}_{t+1}) + (1 - \xi_{t+1})[w_t - \nabla_\gamma G_{(\cdot)}^{\kappa,\omega}(\boldsymbol{\tau}_t, \gamma_t, \mathcal{B}_{t+1})]$;
11: **end for**
12: **Return** $\boldsymbol{\theta}_{T+1}$

---

With the following smoothness assumption, we can get the convergence rate in Thm.3.

**Assumption 1.** $G_{(\cdot)}^{\kappa,\omega}(\boldsymbol{\tau}, \gamma; \mathcal{B})$ *has Lipschitz continuous gradients, i.e., there is a positive scalar $L_G$ such that for any $\boldsymbol{\tau}, \boldsymbol{\tau}' \in \Omega_\tau$, $\gamma, \gamma' \in \Omega_\gamma$,*

$$\|\nabla G_{(\cdot)}^{\kappa,\omega}(\boldsymbol{\tau}, \gamma; \mathcal{B}) - \nabla G_{(\cdot)}^{\kappa,\omega}(\boldsymbol{\tau}', \gamma'; \mathcal{B})\| \leq L_G(\|\boldsymbol{\tau} - \boldsymbol{\tau}'\| + \|\gamma - \gamma'\|). \tag{26}$$

**Theorem 3.** *(Theorem 9 [15]) Supposing that Asm.1 holds, let $\{\boldsymbol{\tau}_t, \gamma_t\}$ be a sequence generated by our method, if the learning rate satisfies:*

$$c_1 \geq \frac{2}{3k^3} + \frac{9\tau^2}{4}, \qquad c_2 \geq \frac{2}{3k^3} + \frac{75L_G^2}{2}$$

$$m \geq \max(2, k^3, (c_1 k)^3, (c_2 k)^3), \lambda \leq \min\left(\frac{1}{6L_G}, \frac{27b\mu}{16}\right) \tag{27}$$

$$\nu \leq \min(\frac{\lambda\tau}{2L_G}\sqrt{\frac{2b}{8\lambda^2 + 75(L_G/\mu)^2 b}}, \frac{m^{1/3}}{2(L_G + \frac{L_G^2}{\mu})k}).$$

*Then we have:*

$$\frac{1}{T}\sum_{t=1}^{T} \mathbb{E}\left[\left\|\frac{1}{\nu}(\boldsymbol{\tau}_t - \mathcal{P}_{\Omega_\tau}(\boldsymbol{\tau}_t - \nu\boldsymbol{v}_t))\right\|\right] \leq \frac{2\sqrt{3M''}m^{1/6}}{T^{1/2}} + \frac{2\sqrt{3M''}}{T^{1/3}}, \tag{28}$$

*where $\|\frac{1}{\nu}(\boldsymbol{\tau}_t - \mathcal{P}_{\Omega_\tau}(\boldsymbol{\tau}_t - \nu\nabla F_{(\cdot)}(\boldsymbol{\tau}_t)))\|$ is the $l_2$-norm of gradient mapping metric for the outer problem [7, 11, 30] with $F_{(\cdot)}(\boldsymbol{\tau}_t) = \max_{\gamma \in \Omega_\gamma} G_{(\cdot)}^{\kappa,\omega}(\boldsymbol{\tau}_t, \gamma)$. When $b = 1$, it is easy to verify that $k = O(1)$, $\lambda = O(\mu)$, $\nu^{-1} = O(L_G/\mu)$, $c_1 = O(1)$, $c_2 = O(L_G^2)$ and $m = O(L_G^6)$. Then we have $M'' = O(L_G^3/\mu^3)$. Thus, the algorithm has a convergence rate of $O(\frac{(L_G/\mu)^{3/2}}{T^{1/3}})$. By $\frac{(L_G/\mu)^{3/2}}{T^{1/3}} \leq \epsilon$, then the iteration number to achieve $\epsilon$-first-order saddle point which satisfies: $T \geq (L_G/\mu)^{4.5}\epsilon^{-3}$.*

## 5   Generalization Analysis

In this section, we theoretically analyze the generalization performance of our proposed estimators for OPAUC (please see the Appendix.G for the TPAUC). According to Thm.2 in Sec.3, we know

that the generalization error of OPAUC with a surrogate loss $\ell$ can be measured as:

$$\mathcal{R}_\beta(f) \propto \min_{(a,b)\in[0,1]^2} \max_{\gamma\in\Omega_\gamma} \min_{s'\in\Omega_{s'}} \mathbb{E}_{\boldsymbol{z}\sim\mathcal{D}_\mathcal{Z}}[G_{op}(f,a,b,\gamma,\boldsymbol{z},s')], \tag{29}$$

and

$$\hat{\mathcal{R}}_\beta(f) \propto \min_{(a,b)\in[0,1]^2} \max_{\gamma\in\Omega_\gamma} \min_{s'\in\Omega_{s'}} \hat{\mathbb{E}}_{\boldsymbol{z}\sim S}[G_{op}(f,a,b,\gamma,\boldsymbol{z},s')], \tag{30}$$

Following the ERM paradigm, to prove the uniform convergence result over a hypothesis class $\mathcal{F}$ of the scoring function $f$, we need to show that:

$$\sup_{f\in\mathcal{F}}\left[\mathcal{R}_\beta(f) - \hat{\mathcal{R}}_\beta(f)\right] \leq \epsilon.$$

According to the aforementioned discussion, we only need to prove that:

$$\sup_{f\in\mathcal{F}}\left[\min_{(a,b)\in[0,1]^2} \max_{\gamma\in\Omega_\gamma} \min_{s'\in\Omega_{s'}} \mathbb{E}_{\boldsymbol{z}\sim\mathcal{D}_\mathcal{Z}}[G_{op}(f,a,b,\gamma,\boldsymbol{z},s')]\right.$$
$$\left. - \min_{(a,b)\in[0,1]^2} \max_{\gamma\in\Omega_\gamma} \min_{s'\in\Omega_{s'}} \hat{\mathbb{E}}_{\boldsymbol{z}\sim S}[G_{op}(f,a,b,\gamma,\boldsymbol{z},s')]\right] \leq \epsilon.$$

To prove this, we need to define the measure of the complexity of the class $\mathcal{F}$. Here we adopt the Radermacher complexity $\Re$ as in [2]. Specifically, we come to the following definition:

**Definition 1.** *The empirical Rademacher complexity of positive and negative instances with respect to $S$ is defined as:*

$$\hat{\Re}_+(\mathcal{F}) = \mathbb{E}_{\boldsymbol{\sigma}}\left[\sup_{f\in\mathcal{F}} \frac{1}{n_+}\sum_{i=1}^{n_+} \sigma_i f(\boldsymbol{x}_i)\right], \tag{31}$$

$$\hat{\Re}_-(\mathcal{F}) = \mathbb{E}_{\boldsymbol{\sigma}}\left[\sup_{f\in\mathcal{F}} \frac{1}{n_-}\sum_{j=1}^{n_-} \sigma_j f(\boldsymbol{x}'_j)\right] \tag{32}$$

*where $(\sigma_1,\cdots,\sigma_{n_+})$ and $(\sigma_1,\cdots,\sigma_{n_-})$ are independent uniform random variables taking values in $\{-1,+1\}$.*

Finally, we come to the generalization bound as follows:

**Theorem 4.** *For any $\delta > 0$, with probability at least $1-\delta$ over the draw of an i.i.d. sample set $S$ of size $n$, for all $f \in \mathcal{F}$ we have:*

$$\min_{(a,b)\in[0,1]^2} \max_{\gamma\in\Omega_\gamma} \min_{s'\in\Omega_{s'}} \mathbb{E}_{\boldsymbol{z}\sim\mathcal{D}_\mathcal{Z}}[G_{op}(f,a,b,\gamma,\boldsymbol{z},s')] \leq \min_{(a,b)\in[0,1]^2} \max_{\gamma\in\Omega_\gamma} \min_{s'\in\Omega_{s'}} \hat{\mathbb{E}}_{\boldsymbol{z}\sim S}[G_{op}(f,a,b,\gamma,\boldsymbol{z},s')]$$
$$+ O(\hat{\Re}_+(\mathcal{F}) + \hat{\Re}_-(\mathcal{F})) + O(n_+^{-1/2} + \beta^{-1}n_-^{-1/2})$$

**Remark 1.** *Although the results we obtain are similar to some previous studies. [39, 25], our generalization analysis is simpler and does not require complex error decomposition. Moreover, our results hold for all real-valued hypothesis class with outputs in $[0,1]$, while the previous results [39, 25] only hold for hard-threshold functions.*

## 6 Experiment

In this section, we conduct a series of experiments on different datasets for both OPAUC and TPAUC optimization. Due to space limitations, please refer to the Appendix.E for the details of implementation and competitors. The source code is available in https://github.com/Shaocr/PAUCI.

### 6.1 Setups

We adopt three imbalanced binary classification datasets: CIFAR-10-LT [8], CIFAR-100-LT [19] and Tiny-ImgaeNet-200-LT following the instructions in [39], where the binary datasets are constructed by selecting one super category as positive class and the other categories as negative class. Please see Appendix.E for more details. The evaluation metrics in experiments are $\hat{\text{AUC}}_\beta$ and $\hat{\text{AUC}}_{\alpha,\beta}$.

## 6.2 Overall Results

In Tab.2, Tab.3, we record the performance on test sets of all the methods on three subsets of CIFAR-10-LT, CIFAR-100-LT, and Tiny-Imagent-200-LT. Each method is tuned independently for OPAUC and TPAUC metrics. From the results, we make the following remarks: (1) Our proposed methods outperform all the competitors in most cases. Even for failure cases, our methods attain fairly competitive results compared with the competitors for OPAUC and TPAUC. (2) In addition, we can see that the normal AUC optimization method AUC-M has less reasonable performance under PAUC metric. This demonstrates the necessity of developing the PAUC optimization algorithm. (3) Approximation methods SOPA-S, AUC-poly, and AUC-exp have lower performance than the unbiased algorithm SOPA and our instance-wise algorithm PAUCI in most cases. Above all, the experimental results show the effectiveness of our proposed method.

Table 2: OPAUC (FPR $\leq$ 0.3) on testing data of different imbalanced datasets. The highest and the second best results are highlighted in orange and blue, respectively.

| Methods | CIFAR-10-LT | | | CIFAR-100-LT | | | Tiny-Imagenet-LT | | |
| --- | --- | --- | --- | --- | --- | --- | --- | --- | --- |
| | Subset 1 | Subset 2 | Subset 3 | Subset 1 | Subset 2 | Subset 3 | Subset 1 | Subset 2 | Subset 3 |
| SOPA [44] | 0.7659 | 0.9688 | 0.7651 | 0.9108 | 0.9875 | 0.8483 | 0.8157 | 0.9037 | 0.9066 |
| SOPA-S [44] | 0.7548 | 0.9674 | 0.7542 | 0.9033 | 0.9860 | 0.8449 | 0.8180 | 0.9087 | 0.9095 |
| AGD-SBCD [44] | 0.7526 | 0.9615 | 0.7497 | 0.9105 | 0.9814 | 0.8406 | 0.8135 | 0.9081 | 0.9057 |
| AUC-poly [39] | 0.7542 | 0.9672 | 0.7538 | 0.9027 | 0.9859 | 0.8441 | 0.8185 | 0.9084 | 0.9100 |
| AUC-exp [39] | 0.7347 | 0.9620 | 0.7457 | 0.8987 | 0.9850 | 0.8407 | 0.8127 | 0.9026 | 0.9049 |
| CE | 0.7417 | 0.9431 | 0.7428 | 0.8903 | 0.9695 | 0.8321 | 0.8023 | 0.8917 | 0.8878 |
| MB [18] | 0.7492 | 0.9648 | 0.7500 | 0.9003 | 0.9804 | 0.8575 | 0.8193 | 0.9072 | 0.9091 |
| AUC-M [41] | 0.7334 | 0.9609 | 0.7442 | 0.8996 | 0.9845 | 0.8403 | 0.8102 | 0.9011 | 0.9043 |
| PAUCI | 0.7721 | 0.9716 | 0.7746 | 0.9155 | 0.9889 | 0.8492 | 0.8267 | 0.9214 | 0.9217 |

Table 3: TPAUC (TPR $\geq$ 0.5, FPR $\leq$ 0.5) on testing data of different imbalanced datasets.

| Methods | CIFAR-10-LT | | | CIFAR-100-LT | | | Tiny-Imagenet-LT | | |
| --- | --- | --- | --- | --- | --- | --- | --- | --- | --- |
| | Subset 1 | Subset 2 | Subset 3 | Subset 1 | Subset 2 | Subset 3 | Subset 1 | Subset 2 | Subset 3 |
| SOPA [44] | 0.7096 | 0.9593 | 0.7220 | 0.8714 | 0.9855 | 0.7485 | 0.7417 | 0.8681 | 0.8650 |
| SOPA-S [44] | 0.6603 | 0.9456 | 0.6917 | 0.8617 | 0.9812 | 0.7419 | 0.7354 | 0.8666 | 0.8628 |
| AUC-poly [39] | 0.6804 | 0.9543 | 0.6974 | 0.8618 | 0.9835 | 0.7431 | 0.7349 | 0.8676 | 0.8627 |
| AUC-exp [39] | 0.6669 | 0.9493 | 0.6930 | 0.8613 | 0.9827 | 0.7447 | 0.7328 | 0.8672 | 0.8626 |
| CE | 0.6420 | 0.9353 | 0.6798 | 0.8467 | 0.9603 | 0.7311 | 0.7223 | 0.8517 | 0.8478 |
| MB [18] | 0.6437 | 0.9492 | 0.6913 | 0.8665 | 0.9677 | 0.7583 | 0.7348 | 0.8651 | 0.8624 |
| AUC-M [41] | 0.6520 | 0.9381 | 0.6821 | 0.8505 | 0.9822 | 0.7324 | 0.7361 | 0.8517 | 0.8598 |
| PAUCI | 0.7192 | 0.9663 | 0.7305 | 0.8814 | 0.9874 | 0.7497 | 0.7618 | 0.8875 | 0.8860 |

## 6.3 Convergence Analysis

In the convergence experiments, for sake of fairness, we did not use warm-up. All algorithms use hyperparameters in the performance experiments. We show the plots of training convergence in Fig.4 and Fig.5 on CIFAR-10 for both OPAUC and TPAUC. Due to the space limitation, the other results could be found in Appendix.E. According to the figures, we can make the following observations: (1) Our algorithm and SOPA converge faster than other methods for OPAUC. However, for TPAUC optimization, the SOPA converges very slowly due to its complicated algorithm, while our method still shows the best convergence property in most cases. (2) It's notable that our algorithm converges to stabilize after twenty epochs in most cases. That means our method has better stability in practice.

## 7 Conclusion

In this paper, we focus on designing an efficient and asymptotically unbiased algorithm for PAUC. We propose a nonconvex strongly concave minimax instance-wise formulation for OPAUC and

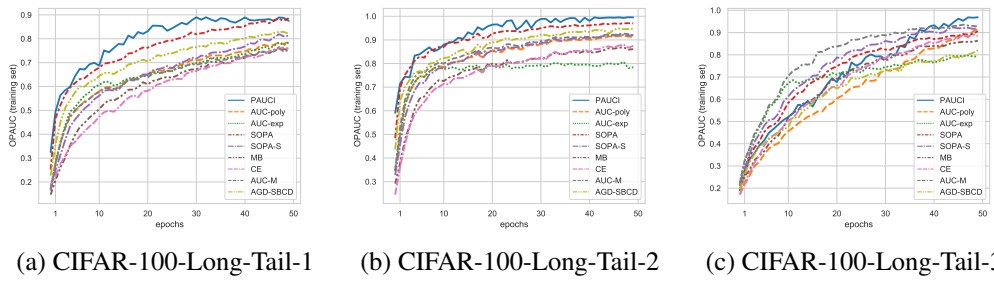

| (a) CIFAR-100-Long-Tail-1 | (b) CIFAR-100-Long-Tail-2 | (c) CIFAR-100-Long-Tail-3 |

Figure 2: Convergence of OPAUC optimization.

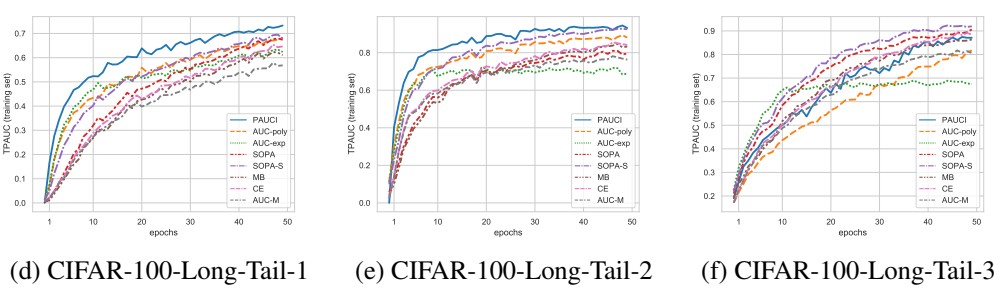

| (d) CIFAR-100-Long-Tail-1 | (e) CIFAR-100-Long-Tail-2 | (f) CIFAR-100-Long-Tail-3 |

Figure 3: Convergence of TPAUC optimization.

TPAUC. In this way, we incorporate the instances selection into the loss calculation to eliminate the score ranking challenge. For OPAUC and TPAUC, we employ an efficient stochastic minimax algorithm that ensures we can find a $\epsilon$-first order saddle point after $O(\epsilon^{-3})$ iterations. Moreover, we present a theoretical analysis of the generalization error of our formulation. Our conclusion may contribute to future work about AUC generalization. Finally, empirical studies over a range of long-tailed benchmark datasets speak to the effectiveness of our proposed algorithm.

# 8 Acknowledgements

This work was supported in part by the National Key R&D Program of China under Grant 2018AAA0102000, in part by National Natural Science Foundation of China: U21B2038, 61931008, 61836002, 6212200758 and 61976202, in part by the Fundamental Research Funds for the Central Universities, in part by Youth Innovation Promotion Association CAS, in part by the Strategic Priority Research Program of Chinese Academy of Sciences, Grant No. XDB28000000, in part by the National Postdoctoral Program for Innovative Talents under Grant BX2021298, and in part by mindspore, which is a new AI computing framework [2].

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
