# List of Appendix

# Appendix A    Related work

**Deep AUC Optimization.** In the past few decades, AUC optimization has already achieved remarkable success in the long-tailed/imbalanced learning task [38]. A partial list of the related literature includes [13, 5, 35, 17, 27, 10, 29, 41]. In recent age, many studies focused on AUC optimization with stochastic gradient method. For example, on top of the square surrogate loss, [41] first proposed a minimax reformulation of the AUC. With a strongly convex regularizer, [26] improved the convergence rate of the stochastic learning algorithm for AUC to $O(1/T)$. In succession, [21, 14, 42] proposed some AUC optimization methods that can be applied to nonconvex deep neural networks.

**Partial AUC (PAUC) Optimization.** [22] first introduced the concept of PAUC. Earlier studies related to PAUC only paid attention to the simplest linear models. In [27], the PAUC is first optimized by a distribution-free rank-based method. [34] developed a non-parametric estimate of the PAUC, and selected features at each step to build the final classifier. [24] develops a cutting plane algorithm to find the most violated constraint instance, decomposing PAUC optimization into subproblems and solving them by an efficient structural SVM-based approach. However, most of the above approaches often fall into the non-differentiable property or intractable optimization problems, posing a significant obstacle to the end-to-end implementation. Using the Implicit Function Theorem, [20] formulated a rate-constrained optimization problem that modeled the quantile threshold as the output of a function of model parameters. As a milestone study, [39] simplifies the challenging sample-selected problem involved in PAUC optimization in a bi-level manner and thus facilitates the end-to-end optimization for PAUC of deep learning. Concretely, the inner-level optimization achieves instances selection, and the outer-level optimization minimizes the loss. However, their estimation may suffer from an approximation error with true PAUC. [44, 40] proposed a smooth estimator of PAUC and provided a sound theoretical convergence guarantee of their algorithm. Nevertheless, their algorithm is limited by a slow convergence rate, especially for TPAUC.

**Generalization Analysis for Partial AUC Optimization.** [25] presented the first generalization analysis for OPAUC and derived a uniform convergence generalization bound. Following their work, a recent study [39] extended this generalization bound to TPAUC. However, limited by the pair-wise form of AUC, all of above studies require complicated decomposition. Moreover, their generalization analysis only hold for hard-threshold functions and VC-dimension. Based on our instance-wise reformulation, we show that the generalization of partial AUC is as simple as other instance-wise algorithm and can deal with real-valued score functions by Rademacher complexity.

# Appendix B    Convergence of the Bias without Regularization

Take OPAUC as an example, we will prove that the approximation induced by $r_k$ has a finite convergence rate which vanishes when $\kappa \to \infty$. For the sake of convenience, we denote the bias as:

$$\Delta_\kappa = \min_{f,(a,b)\in[0,1]^2} \max_{\gamma\in\Omega_\gamma} \min_{s'\in\Omega_{s'}} \hat{\mathbb{E}}_{\boldsymbol{z}\sim S}\Big[G_{op}^\kappa\left(f,a,b,\gamma,\boldsymbol{z},s'\right)\Big]$$
$$- \min_{f,(a,b)\in[0,1]^2} \max_{\gamma\in\Omega_\gamma} \min_{s'\in\Omega_{s'}} \hat{\mathbb{E}}_{\boldsymbol{z}\sim S}\Big[G_{op}(f,a,b,\gamma,\boldsymbol{z},s')\Big]$$

Specifically, we can also prove the following convergence condition holds without the regularization term:

**Theorem 5.** *With the assumption that $f(\boldsymbol{x}) \in [0,1], \forall\boldsymbol{x}$, we have the following convergence result:*

$$\lim_{\kappa\to\infty}\Bigg|\min_{f,(a,b)\in[0,1]^2} \max_{\gamma\in\Omega_\gamma} \min_{s'\in\Omega_{s'}} \hat{\mathbb{E}}_{\boldsymbol{z}\sim S}\Big[G_{op}^\kappa\left(f,a,b,\gamma,\boldsymbol{z},s'\right)\Big]$$
$$- \min_{f,(a,b)\in[0,1]^2} \max_{\gamma\in\Omega_\gamma} \min_{s'\in\Omega_{s'}} \hat{\mathbb{E}}_{\boldsymbol{z}\sim S}\Big[G_{op}(f,a,b,\gamma,\boldsymbol{z},s')\Big]\Bigg| \tag{33}$$
$$= 0.$$

*Moreover, we can also obtain a convergence rate:*

$$\Delta_\kappa = O(1/\kappa). \tag{34}$$

*Proof.* Denote:

$$\Delta_\kappa = \left| \min_{f,(a,b)\in[0,1]^2} \max_{\gamma\in\Omega_\gamma} \min_{s'\in\Omega_{s'}} \hat{\mathbb{E}}_{\mathbf{z}\sim S}\left[G_{op}^\kappa\left(f,a,b,\gamma,\mathbf{z},s'\right)\right] \right.$$

$$\left. - \min_{f,(a,b)\in[0,1]^2} \max_{\gamma\in\Omega_\gamma} \min_{s'\in\Omega_{s'}} \hat{\mathbb{E}}_{\mathbf{z}\sim S}\left[G_{op}(f,a,b,\gamma,\mathbf{z},s')\right] \right|. \tag{35}$$

First, we have:

$$\limsup_{\kappa\to+\infty} \Delta_\kappa \leq \underbrace{\limsup_{\kappa\to+\infty} \sup_{f,(a,b)\in[0,1]^2,\gamma\in\Omega_\gamma,s'\in\Omega_{s'},\mathbf{z}\sim\mathcal{D}_\mathcal{Z}} \left| \frac{\log(1+\exp(\kappa\cdot g))}{\kappa} - [g]_+ \right|}_{(a)}. \tag{36}$$

where $g = (f(\mathbf{x})-b)^2 + 2(1+\gamma)f(\mathbf{x}) - s'$ and $[x]_+ = \max\{x,0\}$. Since $g \in [-5,5]$ in the feasible set, we have:

$$(a) \leq \limsup_{\kappa\to+\infty} \sup_{x\in[-5,5]} \left| \frac{\log(1+\exp(\kappa\cdot x))}{\kappa} - [x]_+ \right|. \tag{37}$$

Next we prove that

$$\limsup_{\kappa\to\infty} \sup_{x\in[-5,5]} \left[ \left| \frac{\log(1+\exp(\kappa\cdot x))}{\kappa} - [x]_+ \right| \right] \leq 0. \tag{38}$$

For the sake of simplicity, we denote:

$$\ell(x) = \left| \frac{\log(1+\exp(\kappa\cdot x))}{\kappa} - [x]_+ \right|. \tag{39}$$

It is easy to see that, when $x < 0$, we have:

$$\ell(x)' = \left( \frac{\log(1+\exp(\kappa\cdot x))}{\kappa} \right)' \geq 0. \tag{40}$$

When $x > 0$, we have:

$$\ell(x)' = \left( \frac{\log(1+\exp(\kappa\cdot x))}{\kappa} - x \right)' \leq 0. \tag{41}$$

Hence, the supremum must be attained at $x = 0$. We thus have:

$$(a) \leq \limsup_{\kappa\to+\infty} \frac{\log(2)}{\kappa} = 0. \tag{42}$$

Obviously, the absolute value ensures that:

$$\liminf_{\kappa\to+\infty} \Delta_\kappa \geq 0. \tag{43}$$

The result follows from the fact:

$$0 \leq \liminf_{\kappa\to+\infty} \Delta_\kappa \leq \limsup_{\kappa\to+\infty} \Delta_\kappa \leq 0. \tag{44}$$

Moreover, from the proof above, we also obtain a convergence rate:

$$\Delta_\kappa = O(1/\kappa). \tag{45}$$

□

## Appendix C  The Constrained Reformulation

In this section, we will prove that the constrained reformulation which is used in the proof of Thm.2 and Thm.8. Our proof can be established by Lem.2, Lem.3, and Thm.6. Throughout the proof, we will define:

$$
\begin{aligned}
a^* &= \mathbb{E}_{\boldsymbol{x}\sim\mathcal{D}_\mathcal{P}}[f(\boldsymbol{x})] & &:= E_+ \\
b^* &= \mathbb{E}_{\boldsymbol{x}'\sim\mathcal{D}_\mathcal{N}}[f(\boldsymbol{x}')|f(\boldsymbol{x}')\geq\eta_\beta(f)] & &:= E_- \\
b^* - a^* & & &:= \Delta E \\
\tilde{a}^* &= \mathbb{E}_{\boldsymbol{x}\sim\mathcal{D}_\mathcal{P}}[f(\boldsymbol{x})|f(\boldsymbol{x})\leq\eta_\alpha(f)] & &:= \tilde{E}_+ \\
b^* - \tilde{a}^* & & &:= \Delta\tilde{E} \\
\mathbb{E}_{\boldsymbol{x}\sim\mathcal{D}_\mathcal{P}}[(f(\boldsymbol{x})-a)^2] & &:= E_a \\
\mathbb{E}_{\boldsymbol{x}\sim\mathcal{D}_\mathcal{P}}[(f(\boldsymbol{x})-a)^2|f(\boldsymbol{x})\leq\eta_\alpha(f)] & &:= \tilde{E}_a \\
\mathbb{E}_{\boldsymbol{x}'\sim\mathcal{D}_\mathcal{N}}[(f(\boldsymbol{x}')-b)^2|f(\boldsymbol{x}')\geq\eta_\beta(f)] & &:= E_b \\
\mathbb{E}_{\boldsymbol{x}\sim\mathcal{D}_\mathcal{P}}[f(\boldsymbol{x})^2|f(\boldsymbol{x})\leq\eta_\alpha(f)] & &:= E_{+,2} \\
\mathbb{E}_{\boldsymbol{x}'\sim\mathcal{D}_\mathcal{N}}[f(\boldsymbol{x}')^2|f(\boldsymbol{x}')\geq\eta_\beta(f)] & &:= E_{-,2}
\end{aligned}
\tag{46}
$$

**Lemma 2.** *(The Reformulation for OPAUC) For a fixed scoring function $f$, the following two problems shares the same optimum, given that the scoring function satisfies: $f(\boldsymbol{x})\in[0,1],\ \forall\boldsymbol{x}$:*

$$
(\boldsymbol{OP1})\ \min_{(a,b)\in[0,1]^2}\ \max_{\gamma\in[-1,1]}\ \mathbb{E}_{\boldsymbol{x}\sim\mathcal{D}_\mathcal{P}}[(f(\boldsymbol{x})-a)^2] + \mathbb{E}_{\boldsymbol{x}'\sim\mathcal{D}_\mathcal{N}}[(f(\boldsymbol{x}')-b)^2|f(\boldsymbol{x}')\geq\eta_\beta(f)]
$$
$$
+2\Delta E + 2\gamma\Delta E - \gamma^2
$$
$$
(\boldsymbol{OP2})\ \min_{(a,b)\in[0,1]^2}\ \max_{\gamma\in[b-1,1]}\ \mathbb{E}_{\boldsymbol{x}\sim\mathcal{D}_\mathcal{P}}[(f(\boldsymbol{x})-a)^2] + \mathbb{E}_{\boldsymbol{x}'\sim\mathcal{D}_\mathcal{N}}[(f(\boldsymbol{x}')-b)^2|f(\boldsymbol{x}')\geq\eta_\beta(f)]
$$
$$
+2\Delta E + 2\gamma\Delta E - \gamma^2
\tag{47}
$$

**Remark 2.** $(\boldsymbol{OP1})$ *and* $(\boldsymbol{OP2})$ *have the equivalent formulation:*

$$
(\boldsymbol{OP1}) \Leftrightarrow \min_{(a,b)\in[0,1]^2}\ \max_{\gamma\in[-1,1]}\ \mathbb{E}_{\boldsymbol{z}\sim\mathcal{D}_\mathcal{Z}}\Big[[(f(\boldsymbol{x})-a)^2 - 2(1+\gamma)f(\boldsymbol{x})]y/p - \gamma^2
$$
$$
[(f(\boldsymbol{x})-b)^2 + 2(1+\gamma)f(\boldsymbol{x})]\cdot[(1-y)\mathbb{I}_{f(\boldsymbol{x})\geq\eta_\beta(f)}]/[(1-p)\beta]\Big].
\tag{48}
$$

$$
(\boldsymbol{OP2}) \Leftrightarrow \min_{(a,b)\in[0,1]^2}\ \max_{\gamma\in[b-1,1]}\ \mathbb{E}_{\boldsymbol{z}\sim\mathcal{D}_\mathcal{Z}}\Big[[(f(\boldsymbol{x})-a)^2 - 2(1+\gamma)f(\boldsymbol{x})]y/p - \gamma^2
$$
$$
[(f(\boldsymbol{x})-b)^2 + 2(1+\gamma)f(\boldsymbol{x})]\cdot[(1-y)\mathbb{I}_{f(\boldsymbol{x})\geq\eta_\beta(f)}]/[(1-p)\beta]\Big].
\tag{49}
$$

*Proof.* From the proof of our main paper, we know that $(OP1)$ has a closed-form minimum:

$$E_{a^*} + E_{b^*} + (\Delta E)^2 + 2\Delta E. \tag{50}$$

Hence, we only need to prove that $(OP2)$ has the same minimum solution. By expanding $(OP2)$, we have:

$$
\min_{(a,b)\in[0,1]^2}\ \max_{\gamma\in[b-1,1]}\ \mathbb{E}_{\boldsymbol{z}\sim\mathcal{D}_\mathcal{Z}}[F_{op}(f,a,b,\gamma,\eta_\beta(f),\boldsymbol{z})] =
$$
$$
2\Delta E + \min_{a\in[0,1]} E_a + \min_{b\in[0,1]}\ \max_{\gamma\in[b-1,1]} F_0
\tag{51}
$$

where

$$F_0 := E_b + 2\gamma\Delta E - \gamma^2 \tag{52}$$

Obviously since $a$ is decoupled with $b, \gamma$, we have:

$$\min_{a \in [0,1]} E_a = E_{a^*} \tag{53}$$

Now, we solve the minimax problem of $F_0$. For any fixed feasible $b$, the inner max problem is a truncated quadratic programming, which has a unique and closed-form solution. Hence, we first solve the inner maximization problem for fixed $b$, and then represent the minimax problem as a minimization problem for $b$. Specifically, we have:

$$\left( \max_{\gamma \in [b-1,1]} 2\gamma\Delta E - \gamma^2 \right) = \begin{cases} (\Delta E)^2, & \Delta E \geq b - 1 \\ 2(b-1)\Delta E - (b-1)^2, & \text{otherwise} \end{cases} \tag{54}$$

Thus, we have:

$$\min_{b \in [0,1]} \max_{\gamma \in [b-1,1]} F_0 = \min_{b \in [0,1]} F_1 \tag{55}$$

where

$$F_1 = \begin{cases} F_{1,0}(b) := E_b + (\Delta E)^2, b - 1 \leq \Delta E \\ F_{1,1}(b) := E_{-,2} - 2bE_- + 2b - 1 + 2(b-1)\Delta E, \text{otherwise} \end{cases} \tag{56}$$

It is easy to see that both cases of $F_1$ are convex functions w.r.t $b$. So, we can find the global minimum by comparing the minimum of $F_{1,0}$ and $F_{1,1}$.

- CASE 1: $\Delta E \geq b - 1$. It is easy to see that $b^* = E_- \in (-\infty, 1 + \Delta E]$, by taking the derivative to zero, we have, the optimum value is obtained at $b = E_-$ for $F_{1,0}$.

- CASE 2: $\Delta E \leq b - 1$. Again by taking the derivative, we have:

$$F_{1,1}(b)' = -2E_- + 2 + 2\Delta E = 2 - 2E_+ \geq 0 \tag{57}$$

  We must have:

$$\inf_{b \geq 1 + \Delta E} F_{1,1}(b) \geq F_{1,1}(1 + \Delta E) = F_{1,0}(1 + \Delta E) \geq F_{1,0}(E_-) = F_{1,0}(b^*) \tag{58}$$

- Putting all together Hence the global minimum of $F_1$ is obtained at $b^*$ with:

$$F_1(b^*) = F_{1,0}(b^*) = E_{b^*} + (\Delta E)^2 \tag{59}$$

Hence, we have $(OP2)$ has the minimum value:

$$E_{a^*} + E_{b^*} + (\Delta E)^2 + 2\Delta E \tag{60}$$

$$\square$$

Now, we use a similar trick to prove the result for TPAUC:

**Lemma 3.** *(The Reformulation for TPAUC) For a fixed scoring function $f$, the following two problems shares the same optimum, given that the scoring function satisfies: $f(\boldsymbol{x}) \in [0,1], \forall \boldsymbol{x}$:*

$$(\boldsymbol{OP3}) \min_{f,(a,b) \in [0,1]^2} \max_{\gamma \in [-1,1]} \mathbb{E}_{\boldsymbol{x} \sim \mathcal{D}_{\mathcal{P}}}[(f(\boldsymbol{x}) - a)^2 | f(\boldsymbol{x}) \leq \eta_\alpha(f)]$$

$$+ \mathbb{E}_{\boldsymbol{x}' \sim \mathcal{D}_{\mathcal{N}}}[(f(\boldsymbol{x}') - b)^2 | f(\boldsymbol{x}') \geq \eta_\beta(f)]$$

$$+ 2\Delta \tilde{E} + 2\gamma\Delta\tilde{E} - \gamma^2.$$

$$(\boldsymbol{OP4}) \min_{f,(a,b) \in [0,1]^2} \max_{\gamma \in [\max\{-a,b-1\},1]} \mathbb{E}_{\boldsymbol{x} \sim \mathcal{D}_{\mathcal{P}}}[(f(\boldsymbol{x}) - a)^2 | f(\boldsymbol{x}) \leq \eta_\alpha(f)]$$

$$+ \mathbb{E}_{\boldsymbol{x}' \sim \mathcal{D}_{\mathcal{N}}}[(f(\boldsymbol{x}') - b)^2 | f(\boldsymbol{x}') \geq \eta_\beta(f)]$$

$$+ 2\Delta\tilde{E} + 2\gamma\Delta\tilde{E} - \gamma^2. \tag{61}$$

**Remark 3.** $(\boldsymbol{OP3})$ *and* $(\boldsymbol{OP4})$ *have the equivalent formulation:*

$$(\boldsymbol{OP3}) \Leftrightarrow \min_{(a,b) \in [0,1]^2} \max_{\gamma \in [-1,1]} \mathbb{E}_{\boldsymbol{z} \sim \mathcal{D}_{\mathcal{Z}}} \Big[ [(f(\boldsymbol{x}) - a)^2 - 2(1+\gamma)f(\boldsymbol{x})] \cdot [y\mathbb{I}_{f(\boldsymbol{x}) \leq \eta_\alpha(f)}]/p - \gamma^2$$

$$[(f(\boldsymbol{x}) - b)^2 + 2(1+\gamma)f(\boldsymbol{x})] \cdot [(1-y)\mathbb{I}_{f(\boldsymbol{x}) \geq \eta_\beta(f)}]/[(1-p)\beta] \Big]. \tag{62}$$

$$(OP4) \Leftrightarrow \min_{(a,b)\in[0,1]^2} \max_{\gamma\in[\max\{-a,b-1\}1]} \mathbb{E}_{\boldsymbol{z}\sim\mathcal{D}_{\mathcal{Z}}}\Big[[(f(\boldsymbol{x})-a)^2 - 2(1+\gamma)f(\boldsymbol{x})]\cdot[y\mathbb{I}_{f(\boldsymbol{x})\leq\eta_\alpha(f)}]/p$$

$$[(f(\boldsymbol{x})-b)^2 + 2(1+\gamma)f(\boldsymbol{x})]\cdot[(1-y)\mathbb{I}_{f(\boldsymbol{x})\geq\eta_\beta(f)}]/[(1-p)\beta] - \gamma^2\Big].$$
(63)

*Proof.* Again, $(OP3)$ has the minimum value:

$$\tilde{E}_{\tilde{a}^*} + E_{b^*} + (\Delta\tilde{E})^2 + 2\Delta\tilde{E} \tag{64}$$

We proof that $(OP4)$ ends up with the minimum value. By expanding $(OP4)$, we have:

$$(OP4) = 2\Delta\tilde{E} + \min_{(a,b)\in[0,1]^2} \max_{\gamma\in[\max\{-a,b-1\},1]} F_3 \tag{65}$$

where

$$F_3 := \tilde{E}_a + E_b + 2\Delta\tilde{E} + 2\gamma\Delta\tilde{E} - \gamma^2 \tag{66}$$

For any fixed feasible $a, b$, the inner max problem is a truncated quadratic programming, which has a unique and closed-form solution. Specifically, define $c = \max\{-a, b-1\}$, we have:

$$\left(\max_{\gamma\in[c,1]} 2\gamma\Delta\tilde{E} - \gamma^2\right) = \begin{cases} (\Delta\tilde{E})^2, & \Delta\tilde{E} \geq c \\ 2c\Delta\tilde{E} - c^2, & \text{otherwise} \end{cases} \tag{67}$$

Thus, we have:

$$\min_{(a,b)\in[0,1]^2} \max_{\gamma\in[c,1]} F_3 = \min_{(a,b)\in[0,1]} F_4 \tag{68}$$

where

$$F_4 = \begin{cases} F_{4,0}(a,b) := \tilde{E}_a + E_b + (\Delta\tilde{E})^2, c \leq \Delta\tilde{E} \\ F_{4,1}(a,b) := \tilde{E}_a + E_{-,2} - 2bE_- + 2(b-1)\Delta\tilde{E} + 2b - 1, b - 1 \geq \Delta\tilde{E}, -a \leq b-1 \\ F_{4,2}(a,b) := E_b + E_{+,2} - 2a\tilde{E}_+ - 2a\Delta\tilde{E}, -a \geq \Delta\tilde{E}, b - 1 \leq -a \end{cases} \tag{69}$$

It is easy to see that both cases of $F_1$ are convex functions w.r.t $b$. So, we can find the global minimum by comparing the minimum of $F_{1,0}$ and $F_{1,1}$.

- CASE 1: $\Delta\tilde{E} \geq \max\{-a, b-1\}$.

  It is easy to check that when $a = \tilde{E}_+, b = E_-$, we have $-a \leq \Delta\tilde{E}$ and $b - 1 \leq \Delta\tilde{E}$. It is easy to see that $a, b$ are decoupled in the expression of $F_{4,0}(a,b)$. By setting:

$$\frac{\partial F_{4,0}(a,b)}{\partial a} = 0,$$
$$\frac{\partial F_{4,0}(a,b)}{\partial b} = 0 \tag{70}$$

  We know that the minimum solution is attained at $a = \tilde{a}^*, b = b^*$. Then the minimum value of $F_{4,0}(a,b)$ at this range becomes:

$$\tilde{E}_{\tilde{a}^*} + E_{b^*} + (\Delta\tilde{E})^2 \tag{71}$$

  Moreover, we will also use the fact that $E_{\tilde{a}^*}$ and $E_{b^*}$ are also the global minimum for $E_a$ and $E_b$, respectively.

- CASE 2: $b - 1 \geq \Delta\tilde{E}, \ -a \leq b-1$.

  It is easy to see that $E_a \geq E_{\tilde{a}^*}$ in this case. According to the same derivation as in Lem.2 CASE 2, we have:

$$E_{-,2} - 2bE_- + 2(b-1)\Delta\tilde{E} + 2b - 1 \geq E_{b^*} + (\Delta\tilde{E})^2 \tag{72}$$

  holds when $b - 1 \geq \Delta\tilde{E}$. Recall that CASE 2 is include in the condition $b - 1 \geq \Delta\tilde{E}$. So, under the condition of CASE 2:

$$F_{4,1}(a,b) \geq \tilde{E}_{\tilde{a}^*} + E_{b^*} + (\Delta\tilde{E})^2 \tag{73}$$

- CASE 3: $-a \geq \Delta\tilde{E}, b - 1 \leq -a$

  In this case, we have $E_b \geq E_{b^*}$. It remains to check:

  $$g(a) = -2a\tilde{E}_+ - 2a\Delta\tilde{E} \tag{74}$$

  By taking derivative, we have:

  $$g'(a) = -2\tilde{E}_+ - 2\Delta\tilde{E} = -2\tilde{E}_- \leq 0. \tag{75}$$

  Similar as the proof of CASE 2, when $-a \geq \Delta\tilde{E}$, we have:

  $$g(a) \geq \tilde{E}_{\tilde{a}^*} + (\Delta\tilde{E})^2 \tag{76}$$

  and thus

  $$F_{4,2}(a,b) \geq \tilde{E}_{\tilde{a}^*} + E_{b^*} + (\Delta\tilde{E})^2 \tag{77}$$

  holds. Since the condition of CASE 3 is included in the set $-a \geq \Delta\tilde{E}$:

  $$F_{4,2}(a,b) \geq \tilde{E}_{\tilde{a}^*} + E_{b^*} + (\Delta\tilde{E})^2 \tag{78}$$

  holds under the condition of CASE 3.

- Putting altogether: The minimum value of $(OP4)$ reads:

  $$\tilde{E}_{\tilde{a}^*} + E_{b^*} + (\Delta\tilde{E})^2 + 2\Delta\tilde{E} \tag{79}$$

  which is the same as $(OP3)$.

$\square$

Finally, since for each fixed $f$ $(OP3) = (OP4)$, and $(OP1) = (OP2)$. We can then claim the following theorem:

**Theorem 6.** *(Constrainted Reformulation)*

$$\min_f(OP1) = \min_f(OP2), \quad \min_f(OP3) = \min_f(OP4) \tag{80}$$

**Remark 4.** *Since the calculation is irelevant to the definition of the expectation, the replace the population-level expectation with the empirical expectation over the training data.*

**Remark 5.** *By applying Theorem 1, we can get the reformulation result in Theorem 2*

*for* OPAUC

$$\min_{f,(a,b)\in[0,1]^2} \max_{\gamma\in[b-1,1]} \min_{s'\in\Omega_{s'}} \mathbb{E}_{\boldsymbol{z}\sim\mathcal{D}_{\mathcal{Z}}}[G_{op}(f,a,b,\gamma,\boldsymbol{z},s')] \tag{81}$$

*where*

$$G_{op}(f,a,b,\gamma,\boldsymbol{z},s') = [(f(\boldsymbol{x}) - a)^2 - 2(1+\gamma)f(\boldsymbol{x})]y/p - \gamma^2 \\ + \left(\beta s' + \left[(f(\boldsymbol{x}) - b)^2 + 2(1+\gamma)f(\boldsymbol{x}) - s'\right]_+\right)(1-y)/[\beta(1-p)]. \tag{82}$$

*for* TPAUC

$$\min_{f,(a,b)\in[0,1]^2} \max_{\gamma\in[\max\{-a,b-1\},1]} \min_{s\in\Omega_s,s'\in\Omega_{s'}} \mathbb{E}_{\boldsymbol{z}\sim\mathcal{D}_{\mathcal{Z}}}[G_{tp}(f,a,b,\gamma,\boldsymbol{z},s,s')] \tag{83}$$

*where*

$$G_{tp}(f,a,b,\gamma,\boldsymbol{z},s,s') = \left(\alpha s + r_\kappa\left((f(\boldsymbol{x}) - a)^2 - 2(1+\gamma)f(\boldsymbol{x}) - s\right)\right)y/(\alpha p) - \gamma^2 \\ + \left(\beta s' + r_\kappa\left((f(\boldsymbol{x}) - b)^2 + 2(1+\gamma)f(\boldsymbol{x}) - s'\right)\right)(1-y)/[\beta(1-p)]. \tag{84}$$

## Appendix D  Reformulation for TPAUC

According to Eq.(5), given a surrogate loss $\ell$ and the finite dataset $S$, maximizing TPAUC and $\hat{\text{AUC}}_{\alpha,\beta}(f, S)$ is equivalent to solving the following problems, respectively:

$$\min_f \mathcal{R}_{\alpha,\beta}(f) = \mathbb{E}_{\boldsymbol{x}\sim\mathcal{D}_{\mathcal{P}}, \boldsymbol{x}'\sim\mathcal{D}_{\mathcal{N}}} \left[ \mathbb{I}_{f(\boldsymbol{x})\leq\eta_\alpha(f)} \cdot \mathbb{I}_{f(\boldsymbol{x}')\geq\eta_\beta(f)} \cdot \ell(f(\boldsymbol{x}) - f(\boldsymbol{x}')) \right], \qquad (85)$$

$$\min_f \hat{\mathcal{R}}_{\alpha,\beta}(f, S) = \sum_{i=1}^{n_+^\alpha} \sum_{j=1}^{n_-^\beta} \frac{\ell\left( f(\boldsymbol{x}_{[i]}) - f(\boldsymbol{x}'_{[j]}) \right)}{n_+^\alpha n_-^\beta}. \qquad (86)$$

Similar to OPAUC, we have the following theorem shows an instance-wise reformulation of the TPAUC optimization problem:

**Theorem 7.** *Assuming that $f(\boldsymbol{x}) \in [0, 1]$, $\forall \boldsymbol{x} \in \mathcal{X}$, $F_{tp}(f, a, b, \gamma, t, t', \boldsymbol{z})$ is defined as:*

$$F_{tp}(f, a, b, \gamma, t, t', \boldsymbol{z}) = (f(\boldsymbol{x}) - a)^2 y\mathbb{I}_{f(\boldsymbol{x})\leq t}/(\alpha p) + (f(\boldsymbol{x}) - b)^2 (1 - y)\mathbb{I}_{f(\boldsymbol{x}')\geq t'}/[\beta(1 - p)]$$
$$+ 2(1 + \gamma)f(\boldsymbol{x})(1 - y)\mathbb{I}_{f(\boldsymbol{x}')\geq t'}/[\beta(1 - p)] - 2(1 + \gamma)f(\boldsymbol{x})y/p\mathbb{I}_{f(\boldsymbol{x})\leq t}/(\alpha p) - \gamma^2, \qquad (87)$$

*where $y = 1$ for positive instances, $y = 0$ for negative instances and we have the following conclusions:*

*(a) (**Population Version**.) We have:*

$$\min_f \mathcal{R}_{\alpha,\beta}(f) \Leftrightarrow \min_{f,(a,b)\in[0,1]^2} \max_{\gamma\in[-1,1]} \mathbb{E}_{\boldsymbol{z}\sim\mathcal{D}_{\mathcal{Z}}} \left[ F_{tp}(f, a, b, \gamma, \eta_\alpha(f), \eta_\beta(f), \boldsymbol{z}) \right], \qquad (88)$$

*where* $\eta_\alpha(f) = \arg\min_{\eta_\alpha\in\mathbb{R}} \left[ \mathbb{E}_{\boldsymbol{x}\sim\mathcal{D}_{\mathcal{P}}}[\mathbb{I}_{f(\boldsymbol{x})\leq\eta_\alpha}] = \alpha \right]$ *and* $\eta_\beta(f) = \arg\min_{\eta_\beta\in\mathbb{R}} \left[ \mathbb{E}_{\boldsymbol{x}'\sim\mathcal{D}_{\mathcal{N}}}[\mathbb{I}_{f(\boldsymbol{x}')\geq\eta_\beta}] = \beta \right]$.

*(b) (**Empirical Version**.) Moreover, given a training dataset $S$ with sample size $n$, denote:*

$$\hat{\mathbb{E}}_{z\sim S}[F_{tp}(f, a, b, \gamma, \hat{\eta}_\alpha(f), \hat{\eta}_\beta(f), \boldsymbol{z})] = \frac{1}{n} \sum_{i=1}^n F_{tp}(f, a, b, \gamma, \hat{\eta}_\alpha(f), \hat{\eta}_\beta(f), \boldsymbol{z}),$$

*where $\hat{\eta}_\alpha(f)$ and $\hat{\eta}_\beta(f)$ are the empirical quantile of the positive and negative instances in $S$, respectively. We have:*

$$\min_f \hat{\mathcal{R}}_{\alpha,\beta}(f, S) \Leftrightarrow \min_{f,(a,b)\in[0,1]^2} \max_{\gamma\in[-1,1]} \hat{\mathbb{E}}_{\boldsymbol{z}\sim S} \left[ F_{tp}(f, a, b, \gamma, \hat{\eta}_\alpha(f), \hat{\eta}_\beta(f), \boldsymbol{z}) \right], \qquad (89)$$

Thm.7 provides a support to convert the pair-wise loss into instance-wise loss for TPAUC. Actually, for $\mathcal{R}_{\alpha,\beta}(f)$, we can just reformulate it as an Average Top-$k$ (ATk) loss. Denote $\ell_+(\boldsymbol{x}) = (f(\boldsymbol{x}) - a)^2 - 2(1 + \gamma)f(\boldsymbol{x})$ and $\ell_-(\boldsymbol{x}') = (f(\boldsymbol{x}') - b)^2 + 2(1 + \gamma)f(\boldsymbol{x}')$. In the proof of the next theorem, we will show that $\ell_+(\boldsymbol{x})$ is an decreasing function and $\ell_-(\boldsymbol{x}')$ is an increasing function w.r.t. $f(\boldsymbol{x})$ and $f(\boldsymbol{x}')$, namely:

$$\mathbb{E}_{\boldsymbol{x}\sim\mathcal{D}_{\mathcal{P}}}[\mathbb{I}_{f(\boldsymbol{x})\leq\eta_\alpha(f)} \cdot \ell_+(\boldsymbol{x})] = \min_s \frac{1}{\alpha} \cdot \mathbb{E}_{\boldsymbol{x}\sim\mathcal{D}_{\mathcal{P}}}[\alpha s + [\ell_+(\boldsymbol{x}) - s]_+], \qquad (90)$$

$$\mathbb{E}_{\boldsymbol{x}'\sim\mathcal{D}_{\mathcal{N}}}[\mathbb{I}_{f(\boldsymbol{x}')\geq\eta_\beta(f)} \cdot \ell_-(\boldsymbol{x}')] = \min_{s'} \frac{1}{\beta} \cdot \mathbb{E}_{\boldsymbol{x}'\sim\mathcal{D}_{\mathcal{N}}}[\beta s' + [\ell_-(\boldsymbol{x}') - s']_+], \qquad (91)$$

The similar result holds for $\hat{\mathcal{R}}_{\alpha,\beta}(f, S)$. Then, we can reach to Thm.8

**Theorem 8.** *Assuming that $f(\boldsymbol{x}) \in [0, 1]$, for all $\boldsymbol{x} \in \mathcal{X}$, we have the equivalent optimization for TPAUC:*

$$\min_{f,(a,b)\in[0,1]^2} \max_{\gamma\in[-1,1]} \mathbb{E}_{\boldsymbol{z}\sim\mathcal{D}_{\mathcal{Z}}} [F_{tp}(f, a, b, \gamma, \eta_\alpha(f), \eta_\beta(f), \boldsymbol{z})]$$
$$\Leftrightarrow \min_{f,(a,b)\in[0,1]^2} \max_{\gamma\in\Omega_\gamma} \min_{s\in\Omega_s, s'\in\Omega_{s'}} \mathbb{E}_{\boldsymbol{z}\sim\mathcal{D}_{\mathcal{Z}}} [G_{tp}(f, a, b, \gamma, \boldsymbol{z}, s, s')], \qquad (92)$$

$$\min_{f,(a,b)\in[0,1]^2} \max_{\gamma\in\Omega_\gamma} \hat{\mathbb{E}}_{\boldsymbol{z}\sim S} [F_{tp}(f, a, b, \gamma, \hat{\eta}_\alpha(f), \hat{\eta}_\beta(f), \boldsymbol{z})]$$
$$\Leftrightarrow \min_{f,(a,b)\in[0,1]^2} \max_{\gamma\in\Omega_\gamma} \min_{s\in\Omega_s, s'\in\Omega_{s'}} \hat{\mathbb{E}}_{\boldsymbol{z}\sim S} [G_{tp}(f, a, b, \gamma, \boldsymbol{z}, s, s')], \qquad (93)$$

*where* $\Omega_\gamma = [\max\{b-1, -a\}, 1]$, $\Omega_s = [-4, 1]$, $\Omega_{s'} = [0, 5]$ *and*

$$
\begin{aligned}
G_{tp}(f, a, b, \gamma, \boldsymbol{z}, s, s') = &\left( \alpha s + \left[ (f(\boldsymbol{x}) - a)^2 - 2(1+\gamma)f(\boldsymbol{x}) - s \right]_+ \right) y/(\alpha p) \\
&+ \left( \beta s' + \left[ (f(\boldsymbol{x}) - b)^2 + 2(1+\gamma)f(\boldsymbol{x}) - s' \right]_+ \right) (1-y)/[\beta(1-p)] - \gamma^2.
\end{aligned}
\tag{94}
$$

Similar to OPAUC, we can get a regularized non-convex strongly-concave TPAUC optimization problem:

$$
\min_{f, (a,b) \in [0,1]^2} \max_{\gamma \in \Omega_\gamma} \min_{s \in \Omega_s, s' \in \Omega_{s'}} \mathop{\mathbb{E}}_{\boldsymbol{z} \sim \mathcal{D}_{\mathcal{Z}}} [G_{tp}^{\kappa, \omega}] \Leftrightarrow \min_{f, (a,b) \in [0,1]^2, s \in \Omega_s, s' \in \Omega_{s'}} \max_{\gamma \in \Omega_\gamma} \mathop{\mathbb{E}}_{\boldsymbol{z} \sim \mathcal{D}_{\mathcal{Z}}} [G_{tp}^{\kappa, \omega}], \tag{95}
$$

$$
\min_{f, (a,b) \in [0,1]^2} \max_{\gamma \in \Omega_\gamma} \min_{s \in \Omega_s, s' \in \Omega_{s'}} \mathop{\hat{\mathbb{E}}}_{\boldsymbol{z} \sim S} [G_{tp}^{\kappa, \omega}] \Leftrightarrow \min_{f, (a,b) \in [0,1]^2, s \in \Omega_s, s' \in \Omega_{s'}} \max_{\gamma \in \Omega_\gamma} \mathop{\hat{\mathbb{E}}}_{\boldsymbol{z} \sim S} [G_{tp}^{\kappa, \omega}], \tag{96}
$$

where $G_{tp}^{\kappa, \omega} = G_{tp}^{\kappa, \omega}(f, a, b, \gamma, \boldsymbol{z}, s, s')$.

## Appendix E  Experiment Details

### E.1  Dataset

**Binary CIFAR-10-Long-Tail Dataset.** The CIFAR-10 dataset contains 60,000 images, each of 32 * 32 shapes, grouped into 10 classes of 6,000 images. The training and test sets contain 50,000 and 10,000 images, respectively. We construct the binary datasets by selecting one super category as positive class and the other categories as negative class. We generate three binary subsets composed of positive categories, including 1) birds, 2) automobiles, and 3) cats.

**Binary CIFAR-100-Long-Tail Dataset.** The original CIFAR-100 dataset has 100 classes, with each containing 600 images. In the CIFAR-100, there are 100 classes divided into 20 superclasses. By selecting a superclass as a positive class example each time, we create CIFAR-100-LT by following the same process as CIFAR-10-LT. The positive superclasses consist of 1) fruits and vegetables, 2) insects, and 3) large omnivores and herbivores, respectively.

**Binary Tiny-ImageNet-200-Long-Tail Dataset.** There are 100,000 256 * 256 color pictures in the Tiny-ImageNet-200 dataset, divided into 200 categories, with 500 pictures per category. We chose three positive superclasses to create binary subsets: 1) dogs, 2) birds, and 3) vehicles.

All data are divided into training, validation, and test sets with proportion 0.7 : 0.15 : 0.15. In each class, sample sizes decay exponentially, and the ratio of sample sizes of the least frequent to the most frequent class is set to 0.01.

Table 4: Details of dataset.

| Dataset | Pos. Class ID | Pos. Class Name | # Pos | #Neg |
|---|---|---|---|---|
| CIFAR-10-LT-1 | 2 | birds | 1,508 | 8,907 |
| CIFAR-10-LT-2 | 1 | automobiles | 2,517 | 7,898 |
| CIFAR-10-LT-3 | 3 | birds | 904 | 9,511 |
| CIFAR-100-LT-1 | 6,7,14,18,24 | insects | 1,928 | 13,218 |
| CIFAR-100-LT-2 | 0,51,53,57,83 | fruits and vegatables | 885 | 14,261 |
| CIFAR-100-LT-3 | 15,19,21,32,38 | large omnivores herbivores | 1,172 | 13,974 |
| Tiny-ImageNet-200-LT-1 | 24,25,26,27,28,29 | dogs | 2,100 | 67,900 |
| Tiny-ImageNet-200-LT-2 | 11,20,21,22 | birds | 1,400 | 68,600 |
| Tiny-ImageNet-200-LT-3 | 70,81,94,107,111,116,121, 133,145,153,164,166 | vehicles | 4,200 | 65,800 |

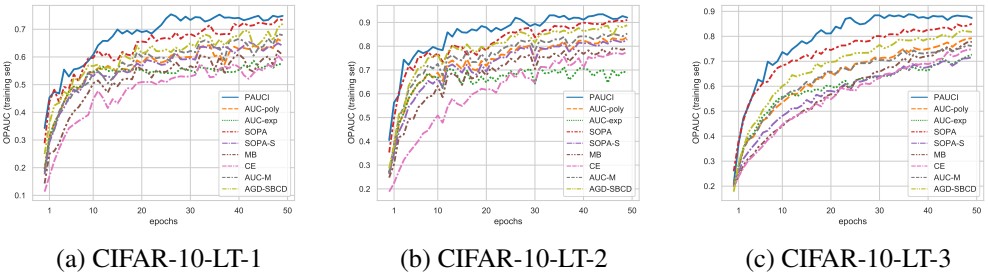

|  |  |  |
|---|---|---|
| (a) CIFAR-10-LT-1 | (b) CIFAR-10-LT-2 | (c) CIFAR-10-LT-3 |

Figure 4: Convergence of OPAUC optimization.

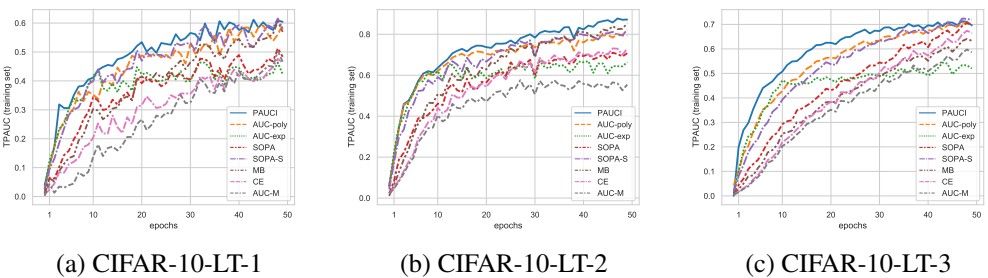

|  |  |  |
|---|---|---|
| (a) CIFAR-10-LT-1 | (b) CIFAR-10-LT-2 | (c) CIFAR-10-LT-3 |

Figure 5: Convergence of TPAUC optimization.

### E.2 Implementation Details

All experiments are conducted on an Ubuntu 16.04.1 server equipped with an Intel(R) Xeon(R) Silver 4110 CPU and four RTX 3090 GPUs, and all codes are developed in `Python 3.8` and `pytorch 1.8.2` environment. We use the ResNet-18 as a backbone. With a `Sigmoid` function, the output is scaled into $[0, 1]$. The batch size is set as $1024$. Following the previous studies [39, 43, 14], we warm up all algorithms for 10 epochs with CELoss to avoid overfitting. All models are trained using `SGD` as the basic optimizer.

### E.3 Competitors

We compare our algorithm with 6 baselines: the approximation algorithms of PAUC, which are denoted as AUC-poly [39] (poly calibrated weighting function) and AUC-exp [39] (exp calibrated weighting function); the DRO formulation of PAUC, which are denoted as SOPA [44] (exact estimator) and SOPA-S [44] (soft estimator); the large-scale PAUC optimization method, which is denoted AGD-SBCD [40]; the naive mini-batch version of empirical partial AUC optimization, which is denoted as MB [18]; the AUC minimax [41] optimization, which is denoted as AUC-M; the binary CELoss; and our method, which is denoted as PAUCI.

### E.4 Parameter Tuning

The learning rate of all methods is tuned in $[10^{-2}, 10^{-5}]$. Weight decay is tuned in $[10^{-3}, 10^{-5}]$. Specifically, $E_k$ for AUC-poly and AUC-exp is searched in $\{3, 5, 8, 10, 12, 15, 18, 20\}$. For AUC-poly, $\gamma$ is searched in $\{0.03, 0.05, 0.08, 0.1, 1, 3, 5\}$. For AUC-exp, $\gamma$ is searched in $\{8, 10, 15, 20, 25, 30\}$. For SOPA-S, we tune the KL-regularization parameter $\lambda$ in $\{0.1, 1.0, 10\}$, and we fix $\beta_0 = \beta_1 = 0.9$. For PAUCI, $k$ is tuned in $[1, 10]$, $\nu$, $\lambda$, $c_1$, $c_2$ are tuned in $[0, 1]$, $m$ is tuned in $[10, 100]$, $\kappa$ is tuned in $[2, 6]$ and $\omega$ is tuned in $[0, 4]$.

### E.5 Per-iteration Acceleration

We conduct some experiments for per-iteration complexity with a fixed epoch with varying $n_+^B$ and $n_-^B$. All experiments are conducted on an Ubuntu 16.04.1 server with an Intel(R) Xeon(R) Silver

4110 CPU. For every method, we repeat running 10000 times and record the average running time. We only record the loss calculation time and use the python package time.time() to calculate the running time. Methods with * stand for the pair-wise estimator, while methods with ** stand for the instance-wise estimator. Here is the result of the experiment. We see the acceleration is significant when the data is large.

Table 5: Pre-Iteration time complexity experiments for OPAUC (FPR $\leq 0.3$):

| unit:ms | $n_+^B = 64$ $n_-^B = 64$ | $n_+^B = 128$ $n_-^B = 128$ | $n_+^B = 256$ $n_-^B = 256$ | $n_+^B = 512$ $n_-^B = 512$ | $n_+^B = 1024$ $n_-^B = 1024$ | $n_+^B = 2048$ $n_-^B = 2048$ |
|---|---|---|---|---|---|---|
| SOPA* | 0.075 | 0.205 | 1.427 | 5.053 | 20.132 | 86.779 |
| SOPA-S* | 0.063 | 0.165 | 0.946 | 4.003 | 15.815 | 62.031 |
| AUC-poly* | 0.062 | 0.178 | 1.086 | 3.553 | 14.266 | 56.637 |
| AUC-exp* | 0.063 | 0.182 | 0.985 | 3.513 | 14.155 | 55.689 |
| AGD-SBCD* | 0.061 | 0.145 | 1.040 | 3.413 | 13.273 | 54.954 |
| MB* | 0.121 | 0.174 | 0.468 | 1.713 | 6.393 | 25.663 |
| PAUCI** | 0.026 | 0.029 | 0.033 | 0.043 | 0.072 | 0.107 |
| AUC-M** | 0.025 | 0.028 | 0.031 | 0.040 | 0.059 | 0.104 |
| CE** | 0.018 | 0.020 | 0.026 | 0.036 | 0.055 | 0.096 |

Table 6: Pre-Iteration time complexity experiments for TPAUC (FPR $\leq 0.5$, TPR $\geq 0.5$):

| unit:ms | $n_+^B = 64$ $n_-^B = 64$ | $n_+^B = 128$ $n_-^B = 128$ | $n_+^B = 256$ $n_-^B = 256$ | $n_+^B = 512$ $n_-^B = 512$ | $n_+^B = 1024$ $n_-^B = 1024$ | $n_+^B = 2048$ $n_-^B = 2048$ |
|---|---|---|---|---|---|---|
| SOPA* | 0.079 | 0.206 | 1.439 | 5.197 | 20.556 | 88.314 |
| SOPA-S* | 0.065 | 0.153 | 0.947 | 3.940 | 15.388 | 62.541 |
| AUC-poly* | 0.062 | 0.180 | 1.175 | 3.573 | 14.440 | 56.469 |
| AUC-exp* | 0.059 | 0.206 | 1.154 | 3.558 | 14.080 | 56.566 |
| MB* | 0.173 | 0.198 | 0.491 | 1.955 | 6.554 | 29.369 |
| PAUCI** | 0.030 | 0.030 | 0.038 | 0.045 | 0.071 | 0.109 |
| AUC-M** | 0.025 | 0.027 | 0.033 | 0.043 | 0.059 | 0.104 |
| CE** | 0.018 | 0.021 | 0.026 | 0.037 | 0.0535 | 0.096 |

# Appendix F  Proofs for Section 3

## F.1  Proof for Lemma 1

**Remainder of Lemma 1.** *$\sum_{i=1}^{k} x_{[i]}$ is a convex function of $(x_1, \cdots, x_n)$ where $x_{[i]}$ is the top-i element of a set $\{x_1, x_2, \cdots, x_n\}$. Furthermore, for $x_i, i = 1, \cdots, n$, we have $\frac{1}{k}\sum_{i=1}^{k} x_{[i]} = \min_s\{s + \frac{1}{k}\sum_{i=1}^{n}[x_i - s]_+\}$, where $[a]_+ = \max\{0, a\}$. The population version is $\mathbb{E}_x[x \cdot \mathbb{I}_{x \geq \eta(\alpha)}] = \min_s \frac{1}{\alpha}\mathbb{E}_x[\alpha s + [x - s]_+]$, where $\eta(\alpha) = \arg\min_{\eta \in \mathbb{R}}[\mathbb{E}_x[\mathbb{I}_{x \geq \eta}] = \alpha]$.*

*Proof.* For the summation case, please see Lemma 1 in [9] for the proof. We only proof the expectation case here. Specifically, calculating the sub-differential of the term $\mathbb{E}_x[\alpha s + [x - s]_+]$ w.r.t., $s$, we get:

$$\alpha - \mathbb{E}_x[\mathbb{I}_{x \geq s}] \in \partial \left(\mathbb{E}_x[\alpha s + [x - s]_+]\right) \tag{97}$$

Since $s$ is convex for $\alpha s + [x - s]_+$, so we can get the optimal $s$ by letting the it be 0:

$$\mathbb{E}_x[\mathbb{I}_{x \geq s}] = \alpha \tag{98}$$

It's' clear that optimal $s$ achieves $\text{top} - \alpha$ quantile. $\square$

### F.2  Proofs for OPAUC

#### F.2.1  Step 1

**Remainder of Theorem 1.** *Assuming that $f(\boldsymbol{x}) \in [0,1]$, $\forall \boldsymbol{x} \in \mathcal{X}$, $F_{op}(f, a, b, \gamma, t, \boldsymbol{z})$ is defined as:*

$$
\begin{aligned}
F_{op}(f, a, b, \gamma, t, \boldsymbol{z}) =& [(f(\boldsymbol{x}) - a)^2 - 2(1+\gamma)f(\boldsymbol{x})]y/p - \gamma^2 \\
& [(f(\boldsymbol{x}) - b)^2 + 2(1+\gamma)f(\boldsymbol{x})](1-y)\mathbb{I}_{f(\boldsymbol{x})\geq t}/(1-p)/\beta,
\end{aligned}
\tag{99}
$$

*where $y = 1$ for positive instances, $y = 0$ for negative instances and we have the following conclusions:*

(a) (**Population Version**.) We have:

$$
\min_f \mathcal{R}_\beta(f) \Leftrightarrow \min_{f, (a,b)\in[0,1]^2} \max_{\gamma\in[-1,1]} \mathbb{E}_{\boldsymbol{z}\sim\mathcal{D}_{\mathcal{Z}}} \left[ F_{op}(f, a, b, \gamma, \eta_\beta(f), \boldsymbol{z}) \right],
\tag{100}
$$

where $\eta_\beta(f) = \arg\min_{\eta_\beta\in\mathbb{R}} \mathbb{E}_{\boldsymbol{x}'\sim\mathcal{D}_{\mathcal{N}}}[\mathbb{I}_{f(\boldsymbol{x}')\geq \eta_\beta} = \beta]$.

(b) (**Empirical Version**.) Moreover, given a training dataset $S$ with sample size $n$, denote:

$$
\hat{\mathbb{E}}_{\boldsymbol{z}\sim S}[F_{op}(f, a, b, \gamma, \hat{\eta}_\beta(f), \boldsymbol{z})] = \frac{1}{n}\sum_{i=1}^{n} F_{op}(f, a, b, \gamma, \hat{\eta}_\beta(f), \boldsymbol{z}_i),
$$

where $\hat{\eta}_\beta(f)$ is the empirical quantile of the negative instances in $S$. We have:

$$
\min_f \hat{\mathcal{R}}_\beta(f, S) \Leftrightarrow \min_{f, (a,b)\in[0,1]^2} \max_{\gamma\in\Omega_\gamma} \hat{\mathbb{E}}_{\boldsymbol{z}\sim S} \left[ F_{op}(f, a, b, \gamma, \hat{\eta}_\beta(f), \boldsymbol{z}) \right],
\tag{101}
$$

*Proof.* Firstly, we give a reformulation of OPAUC:

$$
\begin{aligned}
\min_f \mathcal{R}_\beta(f) &= \min_f \mathbb{E}_{\boldsymbol{x}\sim\mathcal{D}_{\mathcal{P}}, \boldsymbol{x}'\sim\mathcal{D}_{\mathcal{N}}} \left[ \mathbb{I}_{f(\boldsymbol{x}')\geq\eta_\beta(f)} \cdot \ell(f(\boldsymbol{x}) - f(\boldsymbol{x}')) \right] \\
&= \min_f \mathbb{E}_{\boldsymbol{x}\sim\mathcal{D}_{\mathcal{P}}, \boldsymbol{x}'\sim\mathcal{D}_{\mathcal{N}}} \left[ \ell(f(\boldsymbol{x}) - f(\boldsymbol{x}'))|f(\boldsymbol{x}') \geq \eta_\beta(f) \right] \cdot \mathbb{P}_{\boldsymbol{x}'\sim\mathcal{D}_{\mathcal{N}}}[f(\boldsymbol{x}') \geq \eta_\beta(f)] \\
&= \min_f \mathbb{E}_{\boldsymbol{x}\sim\mathcal{D}_{\mathcal{P}}, \boldsymbol{x}'\sim\mathcal{D}_{\mathcal{N}}} \left[ \ell(f(\boldsymbol{x}) - f(\boldsymbol{x}'))|f(\boldsymbol{x}') \geq \eta_\beta(f) \right] \cdot \beta \\
&= \beta \cdot \min_f \mathbb{E}_{\boldsymbol{x}\sim\mathcal{D}_{\mathcal{P}}, \boldsymbol{x}'\sim\mathcal{D}_{\mathcal{N}}} \left[ \ell(f(\boldsymbol{x}) - f(\boldsymbol{x}'))|f(\boldsymbol{x}') \geq \eta_\beta(f) \right].
\end{aligned}
\tag{102}
$$

Applying the surrogate loss $(1-x)^2$ to the estimator of OPAUC, we have:

$$
\begin{aligned}
&\mathbb{E}_{\boldsymbol{x},\boldsymbol{x}'\sim\mathcal{D}_{\mathcal{P}},\mathcal{D}_{\mathcal{N}}} [(1 - (f(\boldsymbol{x}) - f(\boldsymbol{x}')))^2|f(\boldsymbol{x}') \geq \eta_\beta(f)] \\
&= 1 + \mathbb{E}_{\boldsymbol{x}\sim\mathcal{D}_{\mathcal{P}}}[f(\boldsymbol{x})^2] + \mathbb{E}_{\boldsymbol{x}'\sim\mathcal{D}_{\mathcal{N}}}[f(\boldsymbol{x}')^2|f(\boldsymbol{x}') \geq \eta_\beta(f)] - 2\mathbb{E}_{\boldsymbol{x}\sim\mathcal{D}_{\mathcal{P}}}[f(\boldsymbol{x})] \\
&\quad + 2\mathbb{E}_{\boldsymbol{x}'\sim\mathcal{D}_{\mathcal{N}}}[f(\boldsymbol{x}')|f(\boldsymbol{x}') \geq \eta_\beta(f)] - 2\mathbb{E}_{\boldsymbol{x}\sim\mathcal{D}_{\mathcal{P}}}[f(\boldsymbol{x})]\mathbb{E}_{\boldsymbol{x}'\sim\mathcal{D}_{\mathcal{N}}}[f(\boldsymbol{x}')|f(\boldsymbol{x}') \geq \eta_\beta(f)] \\
&= 1 + \mathbb{E}_{\boldsymbol{x}\sim\mathcal{D}_{\mathcal{P}}}[f(\boldsymbol{x})^2] - \mathbb{E}_{\boldsymbol{x}\sim\mathcal{D}_{\mathcal{P}}}[f(\boldsymbol{x})]^2 + \mathbb{E}_{\boldsymbol{x}'\sim\mathcal{D}_{\mathcal{N}}}[f(\boldsymbol{x}')^2|f(\boldsymbol{x}') \geq \eta_\beta(f)] \\
&\quad - \mathbb{E}_{\boldsymbol{x}'\sim\mathcal{D}_{\mathcal{N}}}[f(\boldsymbol{x}')^2|f(\boldsymbol{x}') \geq \eta_\beta(f)]^2 - 2\mathbb{E}_{\boldsymbol{x}\sim\mathcal{D}_{\mathcal{P}}}[f(\boldsymbol{x})] + 2\mathbb{E}_{\boldsymbol{x}'\sim\mathcal{D}_{\mathcal{N}}}[f(\boldsymbol{x}')|f(\boldsymbol{x}') \geq \eta_\beta(f)] \\
&\quad + (\mathbb{E}_{\boldsymbol{x}\sim\mathcal{D}_{\mathcal{P}}}[f(\boldsymbol{x})] - \mathbb{E}_{\boldsymbol{x}'\sim\mathcal{D}_{\mathcal{N}}}[f(\boldsymbol{x}')|f(\boldsymbol{x}') \geq \eta_\beta(f)])^2.
\end{aligned}
\tag{103}
$$

Note that

$$
\mathbb{E}_{\boldsymbol{x}\sim\mathcal{D}_{\mathcal{P}}}[f(\boldsymbol{x})^2] - \mathbb{E}_{\boldsymbol{x}\sim\mathcal{D}_{\mathcal{P}}}[f(\boldsymbol{x})]^2 = \min_{a\in[0,1]} \mathbb{E}_{\boldsymbol{x}\sim\mathcal{D}_{\mathcal{P}}}[(f(\boldsymbol{x}) - a)^2],
\tag{104}
$$

where the minimization is achieved by:

$$
a^* = \mathbb{E}_{\boldsymbol{x}\sim\mathcal{D}_{\mathcal{P}}}[f(\boldsymbol{x})],
\tag{105}
$$

where $a^* \in [0,1]$. Likewise,

$$
\begin{aligned}
&\mathbb{E}_{\boldsymbol{x}'\sim\mathcal{D}_{\mathcal{N}}}[f(\boldsymbol{x}')^2|f(\boldsymbol{x}') \geq \eta_\beta(f)] - \mathbb{E}_{\boldsymbol{x}'\sim\mathcal{D}_{\mathcal{N}}}[f(\boldsymbol{x}')|f(\boldsymbol{x}') \geq \eta_\beta(f)]^2 = \\
&\qquad\qquad \min_{b\in[0,1]} \mathbb{E}_{\boldsymbol{x}'\sim\mathcal{D}_{\mathcal{N}}}[(f(\boldsymbol{x}') - b)^2|f(\boldsymbol{x}') \geq \eta_\beta(f)],
\end{aligned}
\tag{106}
$$

where the minimization is get by:

$$b^* = \mathop{\mathbb{E}}_{\boldsymbol{x}' \sim \mathcal{D}_{\mathcal{N}}}[f(\boldsymbol{x}')|f(\boldsymbol{x}') \geq \eta_\beta(f)]. \tag{107}$$

where $b^* \in [0, 1]$. It's notable that

$$\left(\mathop{\mathbb{E}}_{\boldsymbol{x}' \sim \mathcal{D}_{\mathcal{N}}}[f(\boldsymbol{x}')|f(\boldsymbol{x}') \geq \eta_\beta(f)] - \mathop{\mathbb{E}}_{\boldsymbol{x} \sim \mathcal{D}_{\mathcal{P}}}[f(\boldsymbol{x})]\right)^2 = \\ \max_\gamma \left\{2\gamma \left(\mathop{\mathbb{E}}_{\boldsymbol{x}' \sim \mathcal{D}_{\mathcal{N}}}[f(\boldsymbol{x}')|f(\boldsymbol{x}') \geq \eta_\beta(f)] - \mathop{\mathbb{E}}_{\boldsymbol{x} \sim \mathcal{D}_{\mathcal{P}}}[f(\boldsymbol{x})]\right) - \gamma^2\right\}, \tag{108}$$

where the maximization can be obtained by:

$$\gamma^* = \mathop{\mathbb{E}}_{\boldsymbol{x}' \sim \mathcal{D}_{\mathcal{N}}}[f(\boldsymbol{x}')|f(\boldsymbol{x}') \geq \eta_\beta(f)] - \mathop{\mathbb{E}}_{\boldsymbol{x} \sim \mathcal{D}_{\mathcal{P}}}[f(\boldsymbol{x})]. \tag{109}$$

It's clear that $\gamma^* = b^* - a^*$. Then we can constraint $\gamma$ with range $[-1, 1]$ and get the equivalent optimization formulation:

$$\mathop{\mathbb{E}}_{\boldsymbol{x}, \boldsymbol{x}' \sim \mathcal{D}_{\mathcal{P}}, \mathcal{D}_{\mathcal{N}}}[(1 - (f(\boldsymbol{x}) - f(\boldsymbol{x}')))^2|f(\boldsymbol{x}') \geq \eta_\beta(f)] \Leftrightarrow \\ \min_{(a,b) \in [0,1]^2} \max_{\gamma \in [-1,1]} \mathop{\mathbb{E}}_{\boldsymbol{x} \sim \mathcal{D}_{\mathcal{P}}}[(f(\boldsymbol{x}) - a)^2 - 2(\gamma + 1)f(\boldsymbol{x})] - \gamma^2 \\ + \mathop{\mathbb{E}}_{\boldsymbol{x}' \sim \mathcal{D}_{\mathcal{N}}}[(f(\boldsymbol{x}') - b)^2 + 2(\gamma + 1)f(\boldsymbol{x}')|f(\boldsymbol{x}') \geq \eta_\beta(f)]. \tag{110}$$

Taking expectation *w.r.t.*, $\boldsymbol{z}$, we have:

$$\min_f \mathcal{R}_\beta(f) \Leftrightarrow \min_{f,a,b} \max_{\gamma \in [-1,1]} \mathop{\mathbb{E}}_{\boldsymbol{z} \sim \mathcal{D}_{\mathcal{Z}}}[F_{op}(f, a, b, \gamma, \eta_\beta(f), \boldsymbol{z})], \tag{111}$$

and the instance-wise function $F_{op}(f, a, b, \gamma, \eta_\beta(f), \boldsymbol{z})$ is defined by:

$$F_{op}(f, a, b, \gamma, t, \boldsymbol{z}) = [(f(\boldsymbol{x}) - a)^2 - 2(1 + \gamma)f(\boldsymbol{x})]y/p - \gamma^2 \\ [(f(\boldsymbol{x}) - b)^2 + 2(1 + \gamma)f(\boldsymbol{x})](1 - y)\mathbb{I}_{f(\boldsymbol{x}) \geq t}/(1 - p)/\beta, \tag{112}$$

where $p = \Pr[y = 1]$. The same result holds for empirical version $\mathop{\hat{\mathbb{E}}}_{\boldsymbol{z} \sim S}[F_{op}(f, a, b, \gamma, \hat{\eta}_\beta(f), \boldsymbol{z})]$. $\quad\square$

### F.2.2 Step 2

First we need the following proposition to complete the proof in this subsection.

**Proposition 1.** *If $\gamma \in \Omega_\gamma = [b - 1, 1]$, $\ell_-(\boldsymbol{x}') = (f(\boldsymbol{x}') - b)^2 + 2(1 + \gamma)f(\boldsymbol{x}')$ is an increasing function w.r.t. $f(\boldsymbol{x}')$ when $\boldsymbol{x}' \sim \mathcal{D}_{\mathcal{N}}$ and $f(\boldsymbol{x}') \in [0, 1]$.*

*Proof.* We have:

$$\frac{\partial \ell_-(\boldsymbol{x}')}{\partial f(\boldsymbol{x}')} = 2(f(\boldsymbol{x}') - b + 1 + \gamma). \tag{113}$$

Assuming that $f(\boldsymbol{x}') \in [0, 1]$, then the feasible solution of $b$ is nonnegative. When $\gamma \in [b - 1, 1]$, the negative loss function's partial derivative $\partial \ell_-(\boldsymbol{x}')/\partial f(\boldsymbol{x}') \geq 0$. Then $\ell_-(\boldsymbol{x}')$ is an increasing function *w.r.t.* $f(\boldsymbol{x}')$. $\quad\square$

**Remark 6.** *For negative instances, if the loss function is an increasing function w.r.t. the score $f(\boldsymbol{x}')$, then the top-ranked losses are equivalent to the losses of top-ranked instances.*

**Reminder of Theorem 2.** *Assuming that $f(\boldsymbol{x}) \in [0, 1]$, for all $\boldsymbol{x} \in \mathcal{X}$, we have the equivalent optimization for* OPAUC*:*

$$\min_{f,(a,b) \in [0,1]^2} \max_{\gamma \in [-1,1]} \mathop{\mathbb{E}}_{\boldsymbol{z} \sim \mathcal{D}_{\mathcal{Z}}}[F_{op}(f, a, b, \gamma, \eta_\beta(f), \boldsymbol{z})] \Leftrightarrow \min_{f,(a,b) \in [0,1]^2} \max_{\gamma \in \Omega_\gamma} \min_{s' \in \Omega_{s'}} \mathop{\mathbb{E}}_{\boldsymbol{z} \sim \mathcal{D}_{\mathcal{Z}}}[G_{op}(f, a, b, \gamma, \boldsymbol{z}, s')], \tag{114}$$

$$\min_{f,(a,b) \in [0,1]^2} \max_{\gamma \in [-1,1]} \mathop{\hat{\mathbb{E}}}_{\boldsymbol{z} \sim S}[F_{op}(f, a, b, \gamma, \hat{\eta}_\beta(f), \boldsymbol{z})] \Leftrightarrow \min_{f,(a,b) \in [0,1]^2} \max_{\gamma \in \Omega_\gamma} \min_{s' \in \Omega_{s'}} \mathop{\hat{\mathbb{E}}}_{\boldsymbol{z} \sim S}[G_{op}(f, a, b, \gamma, \boldsymbol{z}, s')], \tag{115}$$

*where* $\Omega_\gamma = [b-1, 1]$, $\Omega_{s'} = [0, 5]$ *and*

$$
\begin{aligned}
G_{op}(f, a, b, \gamma, \boldsymbol{z}, s') = {} & [(f(\boldsymbol{x}) - a)^2 - 2(1 + \gamma)f(\boldsymbol{x})]y/p - \gamma^2 \\
& + \left( \beta s' + \left[ (f(\boldsymbol{x}) - b)^2 + 2(1 + \gamma)f(\boldsymbol{x}) - s' \right]_+ \right) (1 - y)/[\beta(1 - p)].
\end{aligned}
\tag{116}
$$

*Proof.* According to the Thm.6 in Appendix.C, when we constraint $\gamma$ in range $\Omega_\gamma = [b - 1, 1]$, we have:

$$
\min_{f, (a, b) \in [0, 1]^2} \max_{\gamma \in [-1, 1]} \mathbb{E}_{z \sim \mathcal{D}_{\mathcal{Z}}}[F_{op}] \Leftrightarrow \min_{f, (a, b) \in [0, 1]^2} \max_{\gamma \in [b-1, 1]} \mathbb{E}_{z \sim \mathcal{D}_{\mathcal{Z}}}[F_{op}]
\tag{117}
$$

According to Thm.1, we have:

$$
\begin{aligned}
\mathbb{E}_{\boldsymbol{z} \sim \mathcal{D}_{\mathcal{Z}}}[F_{op}(f, a, b, \gamma, \eta_\beta(f), \boldsymbol{z})] \Leftrightarrow {} & \mathbb{E}_{\boldsymbol{x} \sim \mathcal{D}_{\mathcal{P}}}[(f(\boldsymbol{x}) - a)^2 - 2(1 + \gamma)f(\boldsymbol{x})] - \gamma^2 \\
& + \mathbb{E}_{\boldsymbol{x}' \sim \mathcal{D}_{\mathcal{N}}} \left( \left[ (f(\boldsymbol{x}') - b)^2 + 2(1 + \gamma)f(\boldsymbol{x}') \right] \cdot \mathbb{I}_{f(\boldsymbol{x}) \geq \eta_\beta(f)} \right) / \beta.
\end{aligned}
\tag{118}
$$

We denote $\ell_-(\boldsymbol{x}') = (f(\boldsymbol{x}') - b)^2 + 2(1 + \gamma)f(\boldsymbol{x}')$. The Prop.1 ensures that the negative loss function $\ell_-(\boldsymbol{x}')$ is an increasing function when $\gamma \in [b - 1, 1]$. Then we can get:

$$
\mathbb{E}_{\boldsymbol{x}' \sim \mathcal{D}_{\mathcal{N}}}[\mathbb{I}_{f(\boldsymbol{x}') \geq \eta_\beta(f)} \cdot \ell_-(\boldsymbol{x}')] = \min_s \frac{1}{\beta} \mathbb{E}_{\boldsymbol{x}' \sim \mathcal{D}_{\mathcal{N}}}[\beta s + [\ell_-(\boldsymbol{x}') - s]_+],
\tag{119}
$$

Applying Lem.1 to negative loss function, then we have:

$$
\begin{aligned}
\mathbb{E}_{\boldsymbol{z} \sim \mathcal{D}_{\mathcal{Z}}}[F_{op}(f, a, b, \gamma, \eta_\beta(f), \boldsymbol{z})] = {} & \min_{s'} \mathbb{E}_{\boldsymbol{x} \sim \mathcal{D}_{\mathcal{P}}}[(f(\boldsymbol{x}) - a)^2 - 2(1 + \gamma)f(\boldsymbol{x})] - \gamma^2 \\
& + \mathbb{E}_{\boldsymbol{x}' \sim \mathcal{D}_{\mathcal{N}}} \left( \beta s' + \left[ (f(\boldsymbol{x}') - b)^2 + 2(1 + \gamma)f(\boldsymbol{x}') - s' \right]_+ \right) / \beta.
\end{aligned}
\tag{120}
$$

Then, we get:

$$
\mathbb{E}_{\boldsymbol{z} \sim \mathcal{D}_{\mathcal{Z}}}[F_{op}(f, a, b, \gamma, \eta_\beta(f), \boldsymbol{z})] = \min_{s' \in \Omega_{s'}} \mathbb{E}_{\boldsymbol{z} \sim \mathcal{D}_{\mathcal{Z}}}[G_{op}(f, a, b, \gamma, \boldsymbol{z}, s')],
\tag{121}
$$

where

$$
\begin{aligned}
G_{op}(f, a, b, \gamma, \boldsymbol{z}, s') = {} & [(f(\boldsymbol{x}) - a)^2 - 2(1 + \gamma)f(\boldsymbol{x})]y/p - \gamma^2 \\
& + \left( \beta s' + \left[ (f(\boldsymbol{x}) - b)^2 + 2(1 + \gamma)f(\boldsymbol{x}) - s' \right]_+ \right) (1 - y)/[\beta(1 - p)].
\end{aligned}
\tag{122}
$$

We have the equivalent optimization for OPAUC:

$$
\begin{aligned}
& \min_{f, (a, b) \in [0, 1]^2} \max_{\gamma \in [-1, 1]} \mathbb{E}_{\boldsymbol{z} \sim \mathcal{D}_{\mathcal{Z}}}[F_{op}(f, a, b, \gamma, \eta_\beta(f), \boldsymbol{z})] \Leftrightarrow \\
& \min_{f, (a, b) \in [0, 1]^2} \max_{\gamma \in \Omega_\gamma} \min_{s' \in \Omega_{s'}} \mathbb{E}_{\boldsymbol{z} \sim \mathcal{D}_{\mathcal{Z}}}[G_{op}(f, a, b, \gamma, \boldsymbol{z}, s')],
\end{aligned}
\tag{123}
$$

where $\Omega_\gamma = [b - 1, 1]$, $\Omega_{s'} = [0, 5]$, $p = \mathbb{P}[y = 1]$. The same result holds for empirical version $\hat{\mathbb{E}}_{\boldsymbol{z} \sim S}[G_{op}(f, a, b, \gamma, \boldsymbol{z}, s')]$.

$\square$

## F.3 Proofs for TPAUC

### F.3.1 Step 1

**Reminder of Theorem 7.** *Assuming that $f(\boldsymbol{x}) \in [0,1]$, $\forall \boldsymbol{x} \in \mathcal{X}$, $F_{tp}(f, a, b, \gamma, t, t', \boldsymbol{z})$ is defined as:*

$$F_{tp}(f, a, b, \gamma, t, t', \boldsymbol{z}) = (f(\boldsymbol{x}) - a)^2 y \mathbb{I}_{f(\boldsymbol{x}) \leq t}/(\alpha p) + (f(\boldsymbol{x}) - b)^2 (1-y)\mathbb{I}_{f(\boldsymbol{x}') \geq t'}/[\beta(1-p)]$$
$$+ 2(1+\gamma)f(\boldsymbol{x})(1-y)\mathbb{I}_{f(\boldsymbol{x}') \geq t'}/[\beta(1-p)] - 2(1+\gamma)f(\boldsymbol{x})y\mathbb{I}_{f(\boldsymbol{x}) \leq t}/(\alpha p) - \gamma^2, \tag{124}$$

*where $y = 1$ for positive instances, $y = 0$ for negative instances and we have the following conclusions:*

*(a) (**Population Version**.) We have:*

$$\min_f \mathcal{R}_{\alpha,\beta}(f) \Leftrightarrow \min_{f,(a,b) \in [0,1]^2} \max_{\gamma \in [-1,1]} \mathbb{E}_{\boldsymbol{z} \sim \mathcal{D}_{\mathcal{Z}}} \left[ F_{tp}(f, a, b, \gamma, \eta_\alpha(f), \eta_\beta(f), \boldsymbol{z}) \right], \tag{125}$$

*where $\eta_\alpha(f) = \arg\min_{\eta_\alpha \in \mathbb{R}} \mathbb{E}_{\boldsymbol{x} \sim \mathcal{D}_{\mathcal{P}}}[\mathbb{I}_{f(\boldsymbol{x}) \leq \eta_\alpha} = \alpha]$ and $\eta_\beta(f) = \arg\min_{\eta_\beta \in \mathbb{R}} \mathbb{E}_{\boldsymbol{x}' \sim \mathcal{D}_{\mathcal{N}}}[\mathbb{I}_{f(\boldsymbol{x}') \geq \eta_\beta} = \beta]$.*

*(b) (**Empirical Version**.) Moreover, given a training dataset $S$ with sample size $n$, denote:*

$$\hat{\mathbb{E}}_{\boldsymbol{z} \sim S}[F_{tp}(f, a, b, \gamma, \hat{\eta}_\alpha(f), \hat{\eta}_\beta(f), \boldsymbol{z})] = \frac{1}{n} \sum_{i=1}^n F_{tp}(f, a, b, \gamma, \hat{\eta}_\alpha(f), \hat{\eta}_\beta(f), \boldsymbol{z})$$

*where $\hat{\eta}_\alpha(f)$ and $\hat{\eta}_\beta(f)$ are the empirical quantile of the positive and negative instances in $S$, respectively. We have:*

$$\min_f \hat{\mathcal{R}}_{\alpha,\beta}(f, S) \Leftrightarrow \min_{f,(a,b) \in [0,1]^2} \max_{\gamma \in \Omega_\gamma} \hat{\mathbb{E}}_{\boldsymbol{z} \sim S} \left[ F_{tp}(f, a, b, \gamma, \hat{\eta}_\alpha(f), \hat{\eta}_\beta(f), \boldsymbol{z}) \right], \tag{126}$$

*Proof.* Firstly, we give a reformulation of TPAUC:

$$\min_f \mathcal{R}_{\alpha,\beta}(f) = \min_f \mathbb{E}_{\boldsymbol{x} \sim \mathcal{D}_{\mathcal{P}}, \boldsymbol{x}' \sim \mathcal{D}_{\mathcal{N}}} \left[ \mathbb{I}_{f(\boldsymbol{x}) \leq \eta_\alpha(f)} \cdot \mathbb{I}_{f(\boldsymbol{x}') \geq \eta_\beta(f)} \cdot \ell(f(\boldsymbol{x}) - f(\boldsymbol{x}')) \right]$$

$$= \min_f \mathbb{E}_{\boldsymbol{x} \sim \mathcal{D}_{\mathcal{P}}, \boldsymbol{x}' \sim \mathcal{D}_{\mathcal{N}}} \left[ \ell(f(\boldsymbol{x}) - f(\boldsymbol{x}')) | f(\boldsymbol{x}') \geq \eta_\beta(f), f(\boldsymbol{x}) \leq \eta_\alpha(f) \right]$$

$$\cdot \mathop{\mathbb{P}}_{\boldsymbol{x}' \sim \mathcal{D}_{\mathcal{N}}} [f(\boldsymbol{x}') \geq \eta_\beta(f)] \cdot \mathop{\mathbb{P}}_{\boldsymbol{x} \sim \mathcal{D}_{\mathcal{P}}} [f(\boldsymbol{x}) \leq \eta_\alpha(f)]$$

$$= \min_f \mathbb{E}_{\boldsymbol{x} \sim \mathcal{D}_{\mathcal{P}}, \boldsymbol{x}' \sim \mathcal{D}_{\mathcal{N}}} \left[ \ell(f(\boldsymbol{x}) - f(\boldsymbol{x}')) | f(\boldsymbol{x}') \geq \eta_\beta(f), f(\boldsymbol{x}) \leq \eta_\alpha(f) \right] \cdot \alpha\beta$$

$$= \alpha\beta \cdot \min_f \mathbb{E}_{\boldsymbol{x} \sim \mathcal{D}_{\mathcal{P}}, \boldsymbol{x}' \sim \mathcal{D}_{\mathcal{N}}} \left[ \ell(f(\boldsymbol{x}) - f(\boldsymbol{x}')) | f(\boldsymbol{x}') \geq \eta_\beta(f), f(\boldsymbol{x}) \leq \eta_\alpha(f) \right]. \tag{127}$$

Similar to the proof of Thm.1, using the square surrogate loss, we can get the equivalent optimization formulation:

$$\min_f \mathcal{R}_{\alpha,\beta}(f) \Leftrightarrow \min_{f,(a,b) \in [0,1]^2} \max_{\gamma \in [-1,1]} \mathbb{E}_{\boldsymbol{z} \sim \mathcal{D}_{\mathcal{Z}}} [F_{tp}(f, a, b, \gamma, \eta_\alpha(f), \eta_\beta(f), \boldsymbol{z})], \tag{128}$$

and the instance-wise function $F_{tp}(f, a, b, \gamma, \eta_\alpha(f), \eta_\beta(f), \boldsymbol{z})$ is defined by:

$$F_{tp}(f, a, b, \gamma, \eta_\alpha(f), \eta_\beta(f), \boldsymbol{z})$$
$$= (f(\boldsymbol{x}) - a)^2 y \mathbb{I}_{f(\boldsymbol{x}) \leq \eta_\alpha(f)}/(\alpha p) + (f(\boldsymbol{x}) - b)^2 (1-y)\mathbb{I}_{f(\boldsymbol{x}) \geq \eta_\beta(f)}/[\beta(1-p)]$$
$$+ 2(1+\gamma)f(\boldsymbol{x})(1-y)\mathbb{I}_{f(\boldsymbol{x}) \geq \eta_\beta(f)}/[\beta(1-p)] - 2(1+\gamma)f(\boldsymbol{x})y\mathbb{I}_{f(\boldsymbol{x}) \leq \eta_\alpha(f)}/(\alpha p) - \gamma^2. \tag{129}$$

The same result holds for empirical version $\hat{\mathbb{E}}_{\boldsymbol{z} \sim S}[F_{tp}(f, a, b, \gamma, \hat{\eta}_\alpha(f), \hat{\eta}_\beta(f), \boldsymbol{z})]$. $\qquad\square$

### F.3.2 Step 2

First we need the following proposition to complete the proof in this subsection.

**Proposition 2.** *If $\gamma \in \Omega_\gamma = [\max\{b-1, -a\}, 1]$, $\ell_+(\boldsymbol{x}) = (f(\boldsymbol{x}) - a)^2 - 2(1+\gamma)f(\boldsymbol{x})$ is a decreasing function w.r.t. $f(\boldsymbol{x})$ when $\boldsymbol{x} \sim \mathcal{D}_\mathcal{P}$ and $f(\boldsymbol{x}) \in [0, 1]$.*

*Proof.* We have:

$$\frac{\partial \ell_+(\boldsymbol{x})}{\partial f(\boldsymbol{x})} = 2(f(\boldsymbol{x}) - a - 1 - \gamma). \tag{130}$$

Assuming that $f(\boldsymbol{x}) \in [0, 1]$, then the feasible solution of $a$ is nonnegative. When $\gamma \in [\max\{b - 1, -a\}, 1]$, the positive loss function's partial derivative $\partial \ell_+(\boldsymbol{x})/\partial f(\boldsymbol{x}) \leq 0$. Then $\ell_+(\boldsymbol{x})$ is an decreasing function w.r.t. $f(\boldsymbol{x})$. $\qquad \square$

**Remark 7.** *For positive instances, if the loss function is an decreasing function w.r.t. the score $f(\boldsymbol{x})$, then the top-ranked losses are equivalent to the losses of bottom-ranked instances.*

**Reminder of Theorem 8.** *Assuming that $f(\boldsymbol{x}) \in [0, 1]$ for all $\boldsymbol{x} \in \mathcal{X}$, we have the equivalent optimization for* TPAUC*:*

$$\min_{f,(a,b)\in[0,1]^2} \max_{\gamma\in[-1,1]} \mathbb{E}_{\boldsymbol{z}\sim\mathcal{D}_\mathcal{Z}} [F_{tp}(f, a, b, \gamma, \eta_\alpha(f), \eta_\beta(f), \boldsymbol{z})]$$
$$\Leftrightarrow \min_{f,(a,b)\in[0,1]^2} \max_{\gamma\in\Omega_\gamma} \min_{s\in\Omega_s,s'\in\Omega_{s'}} \mathbb{E}_{\boldsymbol{z}\sim\mathcal{D}_\mathcal{Z}} [G_{tp}(f, a, b, \gamma, \boldsymbol{z}, s, s')], \tag{131}$$

$$\min_{f,(a,b)\in[0,1]^2} \max_{\gamma\in[-1,1]} \hat{\mathbb{E}}_{\boldsymbol{z}\sim S} [F_{tp}(f, a, b, \gamma, \hat{\eta}_\alpha(f), \hat{\eta}_\beta(f), \boldsymbol{z})]$$
$$\Leftrightarrow \min_{f,(a,b)\in[0,1]^2} \max_{\gamma\in\Omega_\gamma} \min_{s\in\Omega_s,s'\Omega_{s'}} \hat{\mathbb{E}}_{\boldsymbol{z}\sim S} [G_{tp}(f, a, b, \gamma, \boldsymbol{z}, s, s')], \tag{132}$$

*where $\Omega_\gamma = [\max\{b-1, -a\}, 1]$, $\Omega_s = [-4, 1]$, $\Omega_{s'} = [0, 5]$ and*

$$G_{tp}(f, a, b, \gamma, \boldsymbol{z}, s, s') = \left(\alpha s + \left[(f(\boldsymbol{x}) - a)^2 - 2(1+\gamma)f(\boldsymbol{x}) - s\right]_+\right) y/(\alpha p)$$
$$+ \left(\beta s' + \left[(f(\boldsymbol{x}) - b)^2 + 2(1+\gamma)f(\boldsymbol{x}) - s'\right]_+\right) (1-y)/[\beta(1-p)] - \gamma^2. \tag{133}$$

*Proof.* According to the Thm.6 in Appendix.C, when we constraint $\gamma$ in range $\Omega_\gamma = [\max\{-a, b-1\}, 1]$, we have:

$$\min_{f,(a,b)\in[0,1]^2} \max_{\gamma\in[-1,1]} \mathbb{E}_{\boldsymbol{z}\sim\mathcal{D}_\mathcal{Z}} [F_{tp}] \Leftrightarrow \min_{f,(a,b)\in[0,1]^2} \max_{\gamma\in[\max\{-a,b-1\},1]} \mathbb{E}_{\boldsymbol{z}\sim\mathcal{D}_\mathcal{Z}} [F_{tp}] \tag{134}$$

According to the Thm.8, we have:

$$\mathbb{E}_{\boldsymbol{z}\sim\mathcal{D}_\mathcal{Z}} [F_{tp}(f, a, b, \gamma, \eta_\alpha(f), \eta_\beta(f), \boldsymbol{z})] \Leftrightarrow \mathbb{E}_{\boldsymbol{x}\sim\mathcal{D}_\mathcal{P}} \left([(f(\boldsymbol{x}) - a)^2 - 2(1+\gamma)f(\boldsymbol{x})] \cdot \mathbb{I}_{f(\boldsymbol{x})\leq\eta_\alpha(f)}\right)/\alpha$$
$$+ \mathbb{E}_{\boldsymbol{x}'\sim\mathcal{D}_\mathcal{N}} \left([(f(\boldsymbol{x}') - b)^2 + 2(1+\gamma)f(\boldsymbol{x}')] \cdot \mathbb{I}_{f(\boldsymbol{x}')\geq\eta_\beta(f)}\right)/\beta - \gamma^2. \tag{135}$$

When we constraint $\gamma$ in range $\Omega_\gamma = [\max\{b-1, -a\}, 1]$, Prop.1 and Prop.2 ensure that the positive and negative loss functions are monotonous. Then we can get:

$$\mathbb{E}_{\boldsymbol{x}\sim\mathcal{D}_\mathcal{P}} [\mathbb{I}_{f(\boldsymbol{x})\leq\eta_\alpha(f)} \cdot \ell_+(\boldsymbol{x})] = \min_s \frac{1}{\alpha} \cdot \mathbb{E}_{\boldsymbol{x}\sim\mathcal{D}_\mathcal{P}} [\alpha s + [\ell_+(\boldsymbol{x}) - s]_+], \tag{136}$$

$$\mathbb{E}_{\boldsymbol{x}'\sim\mathcal{D}_\mathcal{N}} [\mathbb{I}_{f(\boldsymbol{x}')\geq\eta_\beta(f)} \cdot \ell_-(\boldsymbol{x}')] = \min_{s'} \frac{1}{\beta} \cdot \mathbb{E}_{\boldsymbol{x}'\sim\mathcal{D}_\mathcal{N}} [\beta s' + [\ell_-(\boldsymbol{x}') - s']_+]. \tag{137}$$

Applying the Lem.1 to positive and negative loss, we have:

$$
\mathop{\mathbb{E}}_{\boldsymbol{z} \sim \mathcal{D}_{\mathcal{Z}}}[F_{tp}(f, a, b, \gamma, \eta_{\alpha}(f), \eta_{\beta}(f), \boldsymbol{z})] \Leftrightarrow \min_{s, s'} \mathop{\mathbb{E}}_{\boldsymbol{x} \sim \mathcal{D}_{\mathcal{P}}} \left( \alpha s + \left[ (f(\boldsymbol{x}) - a)^2 - 2(1+\gamma)f(\boldsymbol{x}) - s \right]_{+} \right) / \alpha - \gamma^2
$$
$$
+ \mathop{\mathbb{E}}_{\boldsymbol{x}' \sim \mathcal{D}_{\mathcal{N}}} \left( \beta s' + \left[ (f(\boldsymbol{x}') - b)^2 + 2(1+\gamma)f(\boldsymbol{x}') - s' \right]_{+} \right) / \beta, \tag{138}
$$

Then, we get:

$$
\mathop{\mathbb{E}}_{\boldsymbol{z} \sim \mathcal{D}_{\mathcal{Z}}}[F_{tp}(f, a, b, \gamma, \eta_{\alpha}(f), \eta_{\beta}(f), \boldsymbol{z})] \Leftrightarrow \min_{s \in \Omega_s, s' \in \Omega_{s'}} \mathop{\mathbb{E}}_{\boldsymbol{z} \sim \mathcal{D}_{\mathcal{Z}}}[G_{tp}(f, a, b, \gamma, \boldsymbol{z}, s, s')]. \tag{139}
$$

where $\Omega_{\gamma} = [\max\{b - 1, -a\}, 1], \Omega_s = [-4, 1], \Omega_{s'} = [0, 5], p = \mathbb{P}[y = 1]$ and

$$
G_{tp}(f, a, b, \gamma, \boldsymbol{z}, s, s') = \left( \alpha s + \left[ (f(\boldsymbol{x}) - a)^2 - 2(1+\gamma)f(\boldsymbol{x}) - s \right]_{+} \right) y/(\alpha p)
$$
$$
+ \left( \beta s' + \left[ (f(\boldsymbol{x}) - b)^2 + 2(1+\gamma)f(\boldsymbol{x}) - s' \right]_{+} \right) (1 - y)/[\beta(1 - p)] - \gamma^2. \tag{140}
$$

we have the equivalent optimization for TPAUC:

$$
\min_{f, (a,b) \in [0,1]^2} \max_{\gamma \in [-1,1]} \mathop{\mathbb{E}}_{\boldsymbol{z} \sim \mathcal{D}_{\mathcal{Z}}}[F_{tp}(f, a, b, \gamma, \eta_{\alpha}(f), \eta_{\beta}(f), \boldsymbol{z})]
$$
$$
\Leftrightarrow \min_{f, (a,b) \in [0,1]^2} \max_{\gamma \in \Omega_{\gamma}} \min_{s \in \Omega_s, s' \in \Omega_{s'}} \mathop{\mathbb{E}}_{\boldsymbol{z} \sim \mathcal{D}_{\mathcal{Z}}}[G_{tp}(f, a, b, \gamma, \boldsymbol{z}, s, s')], \tag{141}
$$

The same result is hold for empirical version $\mathop{\widehat{\mathbb{E}}}_{\boldsymbol{z} \sim S}[G_{tp}(f, a, b, \gamma, \boldsymbol{z}, s, s')]$. $\qquad \square$

## Appendix G   Proof of Generalization Bound

First we need the following lemma to complete the proof in this subsection.

**Lemma 4.**
$$
\max_x f(x) - \max_{x'} g(x') \leq \max_{x, x'=x} f(x) - g(x)
$$
$$
\min_x f(x) - \min_{x'} g(x') \leq \max_{x, x'=x} f(x) - g(x). \tag{142}
$$

*Proof.* Since the difference of suprema does not exceed the supremum of the difference, we have:

$$
\max_x f(x) - \max_{x'} g(x') \leq \max_x \min_{x'} f(x) - g(x') \leq \max_{x, x'=x} f(x) - g(x). \tag{143}
$$

For $\min_x f(x) - \min_{x'} g(x') \leq \max_{x, x'=x} f(x) - g(x)$, we have:

$$
\min_x f(x) - \min_{x'} g(x') \leq \min_x \max_{x'} f(x) - g(x')
$$
$$
= \max_{x'} \min_x f(x) - g(x') \leq \max_{x, x'=x} f(x) - g(x). \tag{144}
$$

$\qquad \square$

**Lemma 5.** *(Talagrand's lemma [31]) Let $\phi_1, \cdots, \phi_m$ be $l$-Lipschitz functions from $\mathbb{R}$ to $\mathbb{R}$ and $\sigma_1, \cdots, \sigma_m$ be Rademacher random variables. Then, for any hypothesis set $\mathcal{H}$ of real-valued functions, the following inequality holds:*

$$
\frac{1}{m} \mathop{\mathbb{E}}_{\boldsymbol{\sigma}} \left[ \sup_{h \in \mathcal{H}} \sum_{i=1}^{m} \sigma_i (\Phi_i \circ h)(x_i) \right] \leq \frac{l}{m} \mathop{\mathbb{E}}_{\boldsymbol{\sigma}} \left[ \sup_{h \in \mathcal{H}} \sum_{i=1}^{m} \sigma_i h(x_i) \right] = l\widehat{\mathfrak{R}}_S(\mathcal{H}). \tag{145}
$$

*In particular, if $\phi_i = \phi$ for all $i \in [m]$, then the following holds:*

$$
\widehat{\mathfrak{R}}_S(\Phi \circ \mathcal{H}) \leq l\widehat{\mathfrak{R}}_S(\mathcal{H}). \tag{146}
$$

**Lemma 6.** *Let $\boldsymbol{\sigma}$ be Rademacher random variables. Then, for any hypothesis set $\mathcal{F}$ of real-valued functions, the following inequality holds:*

$$\mathbb{E}_{\boldsymbol{\sigma}}\left[\sup_{f\in\mathcal{F}}\left|\frac{1}{n_+}\sum_{i=1}^{n_+}\sigma_i f(\boldsymbol{x}_i)\right|\right] \leq 2\hat{\mathfrak{R}}_+(\mathcal{F}) \tag{147}$$

$$\mathbb{E}_{\boldsymbol{\sigma}}\left[\sup_{f\in\mathcal{F}}\left|\frac{1}{n_-}\sum_{j=1}^{n_-}\sigma_j f(\boldsymbol{x}'_j)\right|\right] \leq 2\hat{\mathfrak{R}}_-(\mathcal{F}) \tag{148}$$

*Proof.* Assuming that $0\in\mathcal{F}$, then for any $\boldsymbol{\sigma}$ we have:

$$\sup_{f\in\mathcal{F}}\frac{1}{n_+}\sum_{i=1}^{n_+}\sigma_i f(\boldsymbol{x}_i) \geq \frac{1}{n_+}\sum_{i=1}^{n_+}\sigma_i\cdot 0 = 0 \tag{149}$$

Similarly, for any $\boldsymbol{\sigma}$ we have:

$$\sup_{f\in-\mathcal{F}}\frac{1}{n_+}\sum_{i=1}^{n_+}\sigma_i f(\boldsymbol{x}_i) \geq \frac{1}{n_+}\sum_{i=1}^{n_+}\sigma_i\cdot 0 = 0 \tag{150}$$

where $-\mathcal{F} = \{-f_i(\cdot)\}_{i=1}^{|\mathcal{F}|}$ and $f_i(\cdot)\in\mathcal{F}$. Then we have the following inequality:

$$\mathbb{E}_{\boldsymbol{\sigma}}\left[\sup_{f\in\mathcal{F}}\left|\frac{1}{n_+}\sum_{i=1}^{n_+}\sigma_i f(\boldsymbol{x}_i)\right|\right] = \mathbb{E}_{\boldsymbol{\sigma}}\left[\max\left\{\sup_{f\in\mathcal{F}}\frac{1}{n_+}\sum_{i=1}^{n_+}\sigma_i f(\boldsymbol{x}_i),\ \sup_{f\in-\mathcal{F}}\frac{1}{n_+}\sum_{i=1}^{n_+}\sigma_i f(\boldsymbol{x}_i)\right\}\right]$$

$$\overset{(*)}{\leq}\hat{\mathfrak{R}}_+(\mathcal{F}) + \hat{\mathfrak{R}}_+(-\mathcal{F}) \tag{151}$$

$$= 2\hat{\mathfrak{R}}_+(\mathcal{F})$$

$(*)$ is due to the fact that $\max\{a,b\}\leq a+b$ when $a\geq 0$, $b\geq 0$. The same result holds for negative instances. $\qquad\square$

**Lemma 7.** *Let $\boldsymbol{\sigma}$ be Rademacher random variables. Then, for any hypothesis set $\mathcal{F}$ of real-valued functions, the following inequality holds:*

$$\mathbb{E}_{\boldsymbol{\sigma}}\left[\sup_{a\in[0,1]}\frac{1}{n_+}\sum_{i=1}^{n_+}\sigma_i a^2\right] = O\left(\frac{1}{\sqrt{n_+}}\right), \tag{152}$$

$$\mathbb{E}_{\boldsymbol{\sigma}}\left[\sup_{b\in[0,1]}\frac{1}{n_-}\sum_{j=1}^{n_-}\sigma_j b^2\right] = O\left(\frac{1}{\sqrt{n_-}}\right), \tag{153}$$

*Proof.* Using the Cauchy inequality, we have:

$$\mathbb{E}_{\boldsymbol{\sigma}}\left[\sup_{a\in[0,1]}\frac{1}{n_+}\sum_{i=1}^{n_+}\sigma_i a^2\right] \leq \left(\sup_{a\in[0,1]}|a^2|\right)\cdot\mathbb{E}_{\boldsymbol{\sigma}}\left[\left|\frac{1}{n_+}\sum_{i=1}^{n_+}\sigma_i\right|\right]$$

$$\overset{(r)}{\leq} 1\cdot\sqrt{\mathbb{E}_{\boldsymbol{\sigma}}\left[\left(\frac{1}{n_+}\sum_{i=1}^{n_+}\sigma_i\right)^2\right]}$$

$$= \sqrt{\frac{1}{n_+^2}\mathbb{E}_{\boldsymbol{\sigma}}\left[\sum_{i=1}^{n_+}\sigma_i^2\right]} \tag{154}$$

$$= \sqrt{\frac{1}{n_+}} = O\left(\frac{1}{\sqrt{n_+}}\right)$$

$(r)$ is due to the fact $\sqrt{(\cdot)}$ is concave and the Jensen's inequality. The same result holds for negative instances. $\qquad\square$

## G.1 OPAUC

**Reminder of Theorem 4.** *For any $\delta > 0$, with probability at least $1 - \delta$ over the draw of an i.i.d. sample set $S$ of size $n$, for all $f \in \mathcal{F}$ we have:*

$$\min_{(a,b)\in[0,1]^2} \max_{\gamma\in\Omega_\gamma} \min_{s'\in\Omega_{s'}} \mathbb{E}_{\boldsymbol{z}\sim\mathcal{D}_{\mathcal{Z}}}[G_{op}(f,a,b,\gamma,\boldsymbol{z},s')] \leq \min_{(a,b)\in[0,1]^2} \max_{\gamma\in\Omega_\gamma} \min_{s'\in\Omega_{s'}} \hat{\mathbb{E}}_{\boldsymbol{z}\sim S}[G_{op}(f,a,b,\gamma,\boldsymbol{z},s')]$$
$$+ O(\hat{\Re}_+(\mathcal{F}) + \hat{\Re}_-(\mathcal{F})) + O(n_+^{-1/2} + \beta^{-1}n_-^{-1/2}).$$

*Proof.* According to the Lem.4, we have:

$$\sup_{f\in\mathcal{F}} \left( \min_{(a,b)\in[0,1]^2} \max_{\gamma\in\Omega_\gamma} \min_{s'\in\Omega_{s'}} \mathbb{E}_{\boldsymbol{z}\sim\mathcal{D}_{\mathcal{Z}}}[G_{op}(f,a,b,\gamma,\boldsymbol{z},s')] \right.$$

$$\left. - \min_{(a,b)\in[0,1]^2} \max_{\gamma\in\Omega_\gamma} \min_{s'\in\Omega_{s'}} \hat{\mathbb{E}}_{\boldsymbol{z}\sim S}[G_{op}(f,a,b,\gamma,\boldsymbol{z},s')] \right)$$

$$\leq \sup_{f\in\mathcal{F},(a,b)\in[0,1]^2} \left( \max_{\gamma\in\Omega_\gamma} \min_{s'\in\Omega_{s'}} \mathbb{E}_{\boldsymbol{z}\sim\mathcal{D}_{\mathcal{Z}}}[G_{op}(f,a,b,\gamma,\boldsymbol{z},s')] - \max_{\gamma\in\Omega_\gamma} \min_{s'\in\Omega_{s'}} \hat{\mathbb{E}}_{\boldsymbol{z}\sim S}[G_{op}(f,a,b,\gamma,\boldsymbol{z},s')] \right)$$

$$\leq \sup_{f\in\mathcal{F},(a,b)\in[0,1]^2,\gamma\in\Omega_\gamma} \left( \min_{s'\in\Omega_{s'}} \mathbb{E}_{\boldsymbol{z}\sim\mathcal{D}_{\mathcal{Z}}}[G_{op}(f,a,b,\gamma,\boldsymbol{z},s')] - \min_{s'\in\Omega_{s'}} \hat{\mathbb{E}}_{\boldsymbol{z}\sim S}[G_{op}(f,a,b,\gamma,\boldsymbol{z},s')] \right)$$

$$\leq \sup_{f\in\mathcal{F},(a,b)\in[0,1]^2,s'\in\Omega_{s'},\gamma\in\Omega_\gamma} \left( \mathbb{E}_{\boldsymbol{z}\sim\mathcal{D}_{\mathcal{Z}}}[G_{op}(f,a,b,\gamma,\boldsymbol{z},s')] - \hat{\mathbb{E}}_{\boldsymbol{z}\sim S}[G_{op}(f,a,b,\gamma,\boldsymbol{z},s')] \right)$$

$$\leq \sup_{f\in\mathcal{F},a\in[0,1],\gamma\in\Omega_\gamma} \left( \mathbb{E}_{\boldsymbol{x}\sim\mathcal{D}_{\mathcal{P}}} P(f,a,\gamma,\boldsymbol{x}) - \hat{\mathbb{E}}_{\boldsymbol{x}_i\sim\mathcal{P}} P(f,a,\gamma,\boldsymbol{x}_i) \right)$$

$$+ \sup_{f\in\mathcal{F},b\in[0,1],s'\in\Omega_{s'},\gamma\in\Omega_\gamma} \left( \mathbb{E}_{\boldsymbol{x}'\sim\mathcal{D}_{\mathcal{N}}} N(f,b,\gamma,\boldsymbol{x}',s') - \hat{\mathbb{E}}_{\boldsymbol{x}'_j\sim\mathcal{N}} N(f,b,\gamma,\boldsymbol{x}'_j,s') \right).$$

(155)

where $P(f,a,\gamma,\boldsymbol{x}) = (f(\boldsymbol{x}) - a)^2 - 2(1 + \gamma)f(\boldsymbol{x})$ and $N(f,b,\gamma,\boldsymbol{x}',s') = (\beta s' + [(f(\boldsymbol{x}') - b)^2 + 2(1 + \gamma)f(\boldsymbol{x}') - s']_+)/\beta$. According to the Thm 3.3 in [31], with probability at least $1 - \delta(\delta > 0)$ we have:

$$\sup_{f\in\mathcal{F},a\in[0,1],\gamma\in\Omega_\gamma} \left( \mathbb{E}_{\boldsymbol{x}\sim\mathcal{D}_{\mathcal{P}}} P(f,a,\gamma,\boldsymbol{x}) - \hat{\mathbb{E}}_{\boldsymbol{x}_i\sim\mathcal{P}} P(f,a,\gamma,\boldsymbol{x}_i) \right)$$

$$+ \sup_{f\in\mathcal{F},b\in[0,1],s'\in\Omega_{s'},\gamma\in\Omega_\gamma} \left( \mathbb{E}_{\boldsymbol{x}'\sim\mathcal{D}_{\mathcal{N}}} N(f,b,\gamma,\boldsymbol{x}',s') - \hat{\mathbb{E}}_{\boldsymbol{x}'_j\sim\mathcal{N}} N(f,b,\gamma,\boldsymbol{x}'_j,s') \right)$$

$$\leq 2\underbrace{\mathbb{E}_{\boldsymbol{\sigma}} \left[ \sup_{f\in\mathcal{F},a\in[0,1],\gamma\in\Omega_\gamma} \frac{1}{n_+} \sum_{i=1}^{n_+} \sigma_i P(f,a,\gamma,\boldsymbol{x}_i) \right]}_{(1)} + 12\sqrt{\frac{\log\frac{4}{\delta}}{2n_+}}$$

(156)

$$+ 2\underbrace{\mathbb{E}_{\boldsymbol{\sigma}} \left[ \sup_{f\in\mathcal{F},b\in[0,1],s'\in\Omega_{s'},\gamma\in\Omega_\gamma} \frac{1}{n_-} \sum_{j=1}^{n_-} \sigma_j N(f,b,\gamma,\boldsymbol{x}'_j,s') \right]}_{(2)} + \frac{15}{\beta}\sqrt{\frac{\log\frac{4}{\delta}}{2n_-}}.$$

For term $(1)$, we have:

$$\mathbb{E}_{\boldsymbol{\sigma}}\left[\sup_{f\in\mathcal{F},a\in[0,1],\gamma\in\Omega_\gamma}\frac{1}{n_+}\sum_{i=1}^{n_+}\sigma_i P(f,a,\gamma,\boldsymbol{x}_i)\right]$$

$$=\mathbb{E}_{\boldsymbol{\sigma}}\left[\sup_{f\in\mathcal{F},a\in[0,1],\gamma\in\Omega_\gamma}\frac{1}{n_+}\sum_{i=1}^{n_+}\sigma_i\left(f^2(\boldsymbol{x}_i)-2(1+\gamma+a)f(\boldsymbol{x}_i)+a^2\right)\right]$$

$$\overset{(s)}{\leq}\underbrace{\mathbb{E}_{\boldsymbol{\sigma}}\left[\sup_{f\in\mathcal{F}}\frac{1}{n_+}\sum_{i=1}^{n_+}\sigma_i f^2(\boldsymbol{x}_i)\right]}_{(a)}+\underbrace{\mathbb{E}_{\boldsymbol{\sigma}}\left[\sup_{f\in\mathcal{F}}\frac{1}{n_+}\sum_{i=1}^{n_+}\sigma_i\left(-2f(\boldsymbol{x}_i)\right)\right]}_{(b)}+\underbrace{\mathbb{E}_{\boldsymbol{\sigma}}\left[\sup_{a\in[0,1]}\frac{1}{n_+}\sum_{i=1}^{n_+}\sigma_i\left(a^2\right)\right]}_{(c)}$$

$$+\underbrace{\mathbb{E}_{\boldsymbol{\sigma}}\left[\sup_{f\in\mathcal{F},\gamma\in\Omega_\gamma}\frac{1}{n_+}\sum_{i=1}^{n_+}\sigma_i\left(-2\gamma f(\boldsymbol{x}_i)\right)\right]}_{(d)}+\underbrace{\mathbb{E}_{\boldsymbol{\sigma}}\left[\sup_{f\in\mathcal{F},a\in[0,1]}\frac{1}{n_+}\sum_{i=1}^{n_+}\sigma_i\left(-2af(\boldsymbol{x}_i)\right)\right]}_{(e)}.$$

$$\tag{157}$$

$(s)$ is due to the fact that $\sup_{a,b}a+b\leq\sup a+\sup b$. Assuming that $f(\boldsymbol{x})\in[0,1]$, $0\in\mathcal{F}$. For term (a), according to the Lem.5 and the fact that $x^2$ is 2-Lipschitz continuous within $[0,1]$, we get:

$$\mathbb{E}_{\boldsymbol{\sigma}}\left[\sup_{f\in\mathcal{F}}\frac{1}{n_+}\sum_{i=1}^{n_+}\sigma_i f^2(\boldsymbol{x}_i)\right]\leq 2\mathbb{E}_{\boldsymbol{\sigma}}\left[\sup_{f\in\mathcal{F}}\frac{1}{n_+}\sum_{i=1}^{n_+}\sigma_i f(\boldsymbol{x}_i)\right]=2\hat{\mathfrak{R}}_+(\mathcal{F}).\tag{158}$$

Using the fact that $\sigma_i$ and $-\sigma_i$ are distributed in the same way, we can write the term $(b)$ as:

$$\mathbb{E}_{\boldsymbol{\sigma}}\left[\sup_{f\in\mathcal{F}}\frac{1}{n_+}\sum_{i=1}^{n_+}\sigma_i\left(-2f(\boldsymbol{x}_i)\right)\right]=2\mathbb{E}_{\boldsymbol{\sigma}}\left[\sup_{f\in\mathcal{F}}\frac{1}{n_+}\sum_{i=1}^{n_+}\sigma_i f(\boldsymbol{x}_i)\right]=2\hat{\mathfrak{R}}_+(\mathcal{F}).\tag{159}$$

For term $(c)$, according to the Lem.7, we have:

$$\mathbb{E}_{\boldsymbol{\sigma}}\left[\sup_{a\in[0,1]}\frac{1}{n_+}\sum_{i=1}^{n_+}\sigma_i\left(a^2\right)\right]=O\left(\frac{1}{\sqrt{n_+}}\right).\tag{160}$$

For term $(d)$, we have:

$$\mathbb{E}_{\boldsymbol{\sigma}}\left[\sup_{f\in\mathcal{F},\gamma\in\Omega_\gamma}\frac{1}{n_+}\sum_{i=1}^{n_+}\sigma_i\left(-2\gamma f(\boldsymbol{x}_i)\right)\right]\leq\mathbb{E}_{\boldsymbol{\sigma}}\left[\sup_{f\in\mathcal{F},\gamma\in\Omega_\gamma}\left|\frac{1}{n_+}\sum_{i=1}^{n_+}\sigma_i\left(-2\gamma f(\boldsymbol{x}_i)\right)\right|\right]$$

$$\leq\mathbb{E}_{\boldsymbol{\sigma}}\left[\sup_{f\in\mathcal{F},\gamma\in\Omega_\gamma}|-2\gamma|\cdot\left|\frac{1}{n_+}\sum_{i=1}^{n_+}\sigma_i f(\boldsymbol{x}_i)\right|\right]$$

$$\leq 2\mathbb{E}_{\boldsymbol{\sigma}}\left[\sup_{f\in\mathcal{F}}\left|\frac{1}{n_+}\sum_{i=1}^{n_+}\sigma_i f(\boldsymbol{x}_i)\right|\right]$$

$$\overset{(*)}{\leq}4\hat{\mathfrak{R}}_+(\mathcal{F}),$$

$$\tag{161}$$

where $(*)$ follows from the Lem.6. Similarly, for term $(e)$, we have:

$$\mathbb{E}_{\boldsymbol{\sigma}}\left[\sup_{f\in\mathcal{F},a\in[0,1]}\frac{1}{n_+}\sum_{i=1}^{n_+}\sigma_i\left(-2af(\boldsymbol{x}_i)\right)\right]\leq 4\hat{\mathfrak{R}}_+(\mathcal{F})\tag{162}$$

Combining terms (a), (b), (c), (d), (e), then we get:

$$\mathbb{E}_{\boldsymbol{\sigma}}\left[\sup_{f\in\mathcal{F},a\in[0,1],\gamma\in\Omega_\gamma}\frac{1}{n_+}\sum_{i=1}^{n_+}\sigma_i P(f,a,\gamma,\boldsymbol{x}_i)\right]\leq 12\hat{\mathfrak{R}}_+(\mathcal{F})+O\left(\frac{1}{\sqrt{n_+}}\right)\tag{163}$$

For term $(2)$, we have:

$$\mathbb{E}_{\boldsymbol{\sigma}}\left[\sup_{f\in\mathcal{F},b\in[0,1],s'\in\Omega_{s'},\gamma\in\Omega_\gamma}\frac{1}{n_-}\sum_{j=1}^{n_-}\sigma_j N(f,a,\gamma,\boldsymbol{x}'_j,s')\right]$$

$$=\mathbb{E}_{\boldsymbol{\sigma}}\left[\sup_{f\in\mathcal{F},b\in[0,1],s'\in\Omega_{s'},\gamma\in\Omega_\gamma}\frac{1}{n_-}\sum_{j=1}^{n_-}\sigma_j\left(\left[f^2(\boldsymbol{x}'_j)+2(1+\gamma-b)f(\boldsymbol{x}'_j)+b^2-s'\right]_++\beta s'\right)\right]$$

$$\overset{(o')}{\le}\mathbb{E}_{\boldsymbol{\sigma}}\left[\sup_{f\in\mathcal{F},b\in[0,1],s'\in\Omega_{s'},\gamma\in\Omega_\gamma}\frac{1}{n_-}\sum_{j=1}^{n_-}\sigma_j\left(\left[f^2(\boldsymbol{x}'_j)+2(1+\gamma-b)f(\boldsymbol{x}'_j)+b^2-s'\right]_+\right)\right]+O\left(\frac{1}{\sqrt{n_-}}\right)$$

$$\overset{(l')}{\le}\mathbb{E}_{\boldsymbol{\sigma}}\left[\sup_{f\in\mathcal{F},b\in[0,1],s'\in\Omega_{s'},\gamma\in\Omega_\gamma}\frac{1}{n_-}\sum_{j=1}^{n_-}\sigma_j\left(f^2(\boldsymbol{x}'_j)+2(1+\gamma-b)f(\boldsymbol{x}'_j)+b^2-s'\right)\right]+O\left(\frac{1}{\sqrt{n_-}}\right)$$

$$\overset{(s)}{\le}\underbrace{\mathbb{E}_{\boldsymbol{\sigma}}\left[\sup_{f\in\mathcal{F}}\frac{1}{n_-}\sum_{j=1}^{n_-}\sigma_j f^2(\boldsymbol{x}'_j)\right]}_{(a')}+\underbrace{\mathbb{E}_{\boldsymbol{\sigma}}\left[\sup_{f\in\mathcal{F}}\frac{1}{n_-}\sum_{j=1}^{n_-}\sigma_j 2f(\boldsymbol{x}'_j)\right]}_{(b')}+\underbrace{\mathbb{E}_{\boldsymbol{\sigma}}\left[\sup_{s'\in\Omega_{s'},b\in[0,1]}\frac{1}{n_-}\sum_{j=1}^{n_-}\sigma_j(b^2-s')\right]}_{(c')}$$

$$+\underbrace{\mathbb{E}_{\boldsymbol{\sigma}}\left[\sup_{f\in\mathcal{F},\gamma\in\Omega_\gamma}\frac{1}{n_-}\sum_{j=1}^{n_-}\sigma_j 2\gamma f(\boldsymbol{x}'_j)\right]}_{(d')}+\underbrace{\mathbb{E}_{\boldsymbol{\sigma}}\left[\sup_{f\in\mathcal{F},b\in[0,1]}\frac{1}{n_-}\sum_{j=1}^{n_-}\sigma_j\left(-2bf(\boldsymbol{x}'_j)\right)\right]}_{(e')}+O\left(\frac{1}{\sqrt{n_-}}\right).$$

$$(164)$$

$(o')$ follows from the Lem.7 and the fact that $\sup_{a,b}\le\sup a+\sup b$. $(l')$ follows from the Lem.5 and the fact that $[\cdot]_+$ is 1-Lipschitz continuous. For terms $(a')$, $(b')$, $(c')$, $(d')$, $(e')$, we have the similar results as terms $(a)$, $(b)$, $(c)$, $(d)$, $(e)$. So we can get:

$$\mathbb{E}_{\boldsymbol{\sigma}}\left[\sup_{f\in\mathcal{F},b\in[0,1],s'\in\Omega_{s'},\gamma\in\Omega_\gamma}\frac{1}{n_-}\sum_{j=1}^{n_-}\sigma_j N(f,b,\gamma,\boldsymbol{x}'_j,s')\right]\le 12\hat{\mathfrak{R}}_-(\mathcal{F})+O\left(\frac{1}{\sqrt{n_-}}\right)\qquad(165)$$

and

$$\sup_{f\in\mathcal{F},a\in[0,1],\gamma\in\Omega_\gamma}\left(\mathbb{E}_{\boldsymbol{x}\sim\mathcal{D}_\mathcal{P}}P(f,a,\gamma,\boldsymbol{x})-\hat{\mathbb{E}}_{\boldsymbol{x}_i\sim\mathcal{P}}P(f,a,\gamma,\boldsymbol{x}_i)\right)$$

$$+\sup_{f\in\mathcal{F},b\in[0,1],s'\in\Omega_{s'},\gamma\in\Omega_\gamma}\left(\mathbb{E}_{\boldsymbol{x}'\sim\mathcal{D}_\mathcal{N}}N(f,b,\gamma,\boldsymbol{x}',s')-\hat{\mathbb{E}}_{\boldsymbol{x}'_j\sim\mathcal{N}}N(f,b,\gamma,\boldsymbol{x}'_j,s')\right)$$

$$(166)$$

$$\le 2\left(12\hat{\mathfrak{R}}_+(\mathcal{F})+O\left(\frac{1}{\sqrt{n_+}}\right)+12\hat{\mathfrak{R}}_-(\mathcal{F})+O\left(\frac{1}{\sqrt{n_-}}\right)\right)+12\sqrt{\frac{\log\frac{4}{\delta}}{2n_+}}+\frac{15}{\beta}\sqrt{\frac{\log\frac{4}{\delta}}{2n_-}}$$

$$=O(\hat{\mathfrak{R}}_+(\mathcal{F})+\hat{\mathfrak{R}}_-(\mathcal{F}))+O(n_+^{-1/2}+\beta^{-1}n_-^{-1/2})$$

For any $\delta>0$, with probability at least $1-\delta$ over the draw of an i.i.d. sample $S$ of positive instances size $n_+$ (negative $n_-$ resp.), each of the following holds for all $f\in\mathcal{F}$:

$$\min_{(a,b)\in[0,1]^2}\max_{\gamma\in\Omega_\gamma}\min_{s'\in\Omega_{s'}}\mathbb{E}_{\boldsymbol{z}\sim\mathcal{D}_\mathcal{Z}}[G_{op}(f,a,b,\gamma,\boldsymbol{z},s')]$$

$$\le\min_{(a,b)\in[0,1]^2}\max_{\gamma\in\Omega_\gamma}\min_{s'\in\Omega_{s'}}\hat{\mathbb{E}}_{\boldsymbol{z}\sim S}[G_{op}(f,a,b,\gamma,\boldsymbol{z},s')]+O(\hat{\mathfrak{R}}_+(\mathcal{F})+\beta^{-1}\hat{\mathfrak{R}}_-(\mathcal{F}))$$

$$+O(n_+^{-1/2}+\beta^{-1}n_-^{-1/2}).$$

$$\square$$

## G.2  TPAUC

**Theorem 9.** *For any $\delta > 0$, with probability at least $1 - \delta$ over the draw of an i.i.d. sample set $S$ of size $n$, for all $f \in \mathcal{F}$ we have:*

$$\min_{(a,b)\in[0,1]^2} \max_{\gamma\in\Omega_\gamma} \min_{s\in\Omega_s, s'\in\Omega_{s'}} \mathop{\mathbb{E}}_{\boldsymbol{z}\sim\mathcal{D}_{\mathcal{Z}}} [G_{tp}(f,a,b,\gamma,\boldsymbol{z},s,s')] \leq$$

$$\min_{(a,b)\in[0,1]^2} \max_{\gamma\in\Omega_\gamma} \min_{s\in\Omega_s, s'\in\Omega_{s'}} \mathop{\hat{\mathbb{E}}}_{\boldsymbol{z}\sim S} [G_{tp}(f,a,b,\gamma,\boldsymbol{z},s,s')]$$

$$+ O(\hat{\mathfrak{R}}_+(\mathcal{F}) + \hat{\mathfrak{R}}_-(\mathcal{F})) + O(\alpha^{-1} n_+^{-1/2} + \beta^{-1} n_-^{-1/2}).$$

*Proof.* According to Lem.4, we have:

$$\sup_{f\in\mathcal{F}} \left( \min_{(a,b)\in[0,1]^2} \max_{\gamma\in\Omega_\gamma} \min_{s\in\Omega_s, s'\in\Omega_{s'}} \mathop{\mathbb{E}}_{\boldsymbol{z}\sim\mathcal{D}_{\mathcal{Z}}} [G_{tp}(f,a,b,\gamma,\boldsymbol{z},s,s')] \right.$$

$$\left. - \min_{(a,b)\in[0,1]^2} \max_{\gamma\in\Omega_\gamma} \min_{s\in\Omega_s, s'\in\Omega_{s'}} \mathop{\hat{\mathbb{E}}}_{\boldsymbol{z}\sim S} [G_{tp}(f,a,b,\gamma,\boldsymbol{z},s,s')] \right)$$

$$\leq \sup_{f\in\mathcal{F},(a,b)\in[0,1]^2} \left( \max_{\gamma\in\Omega_\gamma} \min_{s\in\Omega_s, s'\in\Omega_{s'}} \mathop{\mathbb{E}}_{\boldsymbol{z}\sim\mathcal{D}_{\mathcal{Z}}} [G_{tp}(f,a,b,\gamma,\boldsymbol{z},s,s')] \right.$$

$$\left. - \max_{\gamma\in\Omega_\gamma} \min_{s\in\Omega_s, s'\in\Omega_{s'}} \mathop{\hat{\mathbb{E}}}_{\boldsymbol{z}\sim S} [G_{tp}(f,a,b,\gamma,\boldsymbol{z},s,s')] \right)$$

$$\leq \sup_{f\in\mathcal{F},(a,b)\in[0,1]^2,\gamma\in\Omega_\gamma} \left( \min_{s\in\Omega_s, s'\in\Omega_{s'}} \mathop{\mathbb{E}}_{\boldsymbol{z}\sim\mathcal{D}_{\mathcal{Z}}} [G_{tp}(f,a,b,\gamma,\boldsymbol{z},s,s')] \right.$$

$$\left. - \min_{s\in\Omega_s, s'\in\Omega_{s'}} \mathop{\hat{\mathbb{E}}}_{\boldsymbol{z}\sim S} [G_{tp}(f,a,b,\gamma,\boldsymbol{z},s,s')] \right) \tag{167}$$

$$\leq \sup_{f\in\mathcal{F},(a,b)\in[0,1]^2,s\in\Omega_s, s'\in\Omega_{s'},\gamma\in\Omega_\gamma} \left( \mathop{\mathbb{E}}_{\boldsymbol{z}\sim\mathcal{D}_{\mathcal{Z}}} [G_{tp}(f,a,b,\gamma,\boldsymbol{z},s,s')] \right.$$

$$\left. - \mathop{\hat{\mathbb{E}}}_{\boldsymbol{z}\sim S} [G_{tp}(f,a,b,\gamma,\boldsymbol{z},s,s')] \right)$$

$$\leq \sup_{f\in\mathcal{F},a\in[0,1],s\in\Omega_s,\gamma\in\Omega_\gamma} \left( \mathop{\mathbb{E}}_{\boldsymbol{x}\sim\mathcal{D}_{\mathcal{P}}} P(f,a,\gamma,\boldsymbol{x},s) - \mathop{\hat{\mathbb{E}}}_{\boldsymbol{x}_i\sim\mathcal{P}} P(f,a,\gamma,\boldsymbol{x}_i,s) \right)$$

$$+ \sup_{f\in\mathcal{F},b\in[0,1],s'\in\Omega_{s'},\gamma\in\Omega_\gamma} \left( \mathop{\mathbb{E}}_{\boldsymbol{x}'\sim\mathcal{D}_{\mathcal{N}}} N(f,b,\gamma,\boldsymbol{x}',s') - \mathop{\hat{\mathbb{E}}}_{\boldsymbol{x}'_j\sim\mathcal{N}} N(f,b,\gamma,\boldsymbol{x}'_j,s') \right).$$

where $P(f,a,\gamma,\boldsymbol{x},s) = (\alpha s + \big[(f(\boldsymbol{x})-a)^2 - 2(1+\gamma)f(\boldsymbol{x}) - s\big]_+)/\alpha$ and $N(f,b,\gamma,\boldsymbol{x}',s') = (\beta s' + \big[(f(\boldsymbol{x}')-b)^2 + 2(1+\gamma)f(\boldsymbol{x}') - s'\big]_+)/\beta$. According to the Thm 3.3 in [31], with probability at least $1 - \delta(\delta > 0)$ we have:

$$\sup_{f\in\mathcal{F},a\in[0,1],s\in\Omega_s\gamma\in\Omega_\gamma} \left( \mathop{\mathbb{E}}_{\boldsymbol{x}\sim\mathcal{D}_{\mathcal{P}}} P(f,a,\gamma,\boldsymbol{x},s) - \mathop{\hat{\mathbb{E}}}_{\boldsymbol{x}_i\sim\mathcal{P}} P(f,a,\gamma,\boldsymbol{x}_i,s) \right)$$

$$+ \sup_{f\in\mathcal{F},b\in[0,1],s'\in\Omega_{s'},\gamma\in\Omega_\gamma} \left( \mathop{\mathbb{E}}_{\boldsymbol{x}'\sim\mathcal{D}_{\mathcal{N}}} N(f,b,\gamma,\boldsymbol{x}',s') - \mathop{\hat{\mathbb{E}}}_{\boldsymbol{x}'_j\sim\mathcal{N}} N(f,b,\gamma,\boldsymbol{x}'_j,s') \right)$$

$$\leq \underbrace{2\mathop{\mathbb{E}}_{\boldsymbol{\sigma}} \left[ \sup_{f\in\mathcal{F},a\in[0,1],s\in\Omega_s,\gamma\in\Omega_\gamma} \frac{1}{n_+} \sum_{i=1}^{n_+} \sigma_i P(f,a,\gamma,\boldsymbol{x}_i,s) \right]}_{(3)} + \frac{12}{\alpha}\sqrt{\frac{\log\frac{4}{\delta}}{2n_+}} \tag{168}$$

$$+ \underbrace{2\mathop{\mathbb{E}}_{\boldsymbol{\sigma}} \left[ \sup_{f\in\mathcal{F},b\in[0,1],s'\in\Omega_{s'},\gamma\in\Omega_\gamma} \frac{1}{n_-} \sum_{j=1}^{n_-} \sigma_j N(f,b,\gamma,\boldsymbol{x}'_j,s') \right]}_{(4)} + \frac{15}{\beta}\sqrt{\frac{\log\frac{4}{\delta}}{2n_-}}.$$

For term $(3)$, we have:

$$\mathbb{E}_{\boldsymbol{\sigma}}\left[\sup_{f\in\mathcal{F},a\in[0,1],s\in\Omega_s,\gamma\in\Omega_\gamma}\frac{1}{n_+}\sum_{i=1}^{n_+}\sigma_i P(f,a,\gamma,\boldsymbol{x}_i,s)\right]$$

$$=\mathbb{E}_{\boldsymbol{\sigma}}\left[\sup_{f\in\mathcal{F},a\in[0,1],s\in\Omega_s,\gamma\in\Omega_\gamma}\frac{1}{n_+}\sum_{i=1}^{n_+}\sigma_i\left(\left[f^2(\boldsymbol{x}_i)-2(1+\gamma+a)f(\boldsymbol{x}_i)+a^2-s\right]_+ +\alpha s\right)\right]$$

$$\overset{(o^*)}{\leq}\mathbb{E}_{\boldsymbol{\sigma}}\left[\sup_{f\in\mathcal{F},a\in[0,1],s\in\Omega_s,\gamma\in\Omega_\gamma}\frac{1}{n_+}\sum_{i=1}^{n_+}\sigma_i\left(\left[f^2(\boldsymbol{x}_i)-2(1+\gamma+a)f(\boldsymbol{x}_i)+a^2-s\right]_+\right)\right]+O\left(\frac{1}{\sqrt{n_+}}\right)$$

$$\overset{(l^*)}{\leq}\mathbb{E}_{\boldsymbol{\sigma}}\left[\sup_{f\in\mathcal{F},a\in[0,1],s\in\Omega_s,\gamma\in\Omega_\gamma}\frac{1}{n_+}\sum_{i=1}^{n_+}\sigma_i\left(f^2(\boldsymbol{x}_i)-2(1+\gamma+a)f(\boldsymbol{x}_i)+a^2-s\right)\right]+O\left(\frac{1}{\sqrt{n_+}}\right)$$

$$\overset{(s)}{\leq}\underbrace{\mathbb{E}_{\boldsymbol{\sigma}}\left[\sup_{f\in\mathcal{F}}\frac{1}{n_+}\sum_{i=1}^{n_+}\sigma_i f^2(\boldsymbol{x}_i)\right]}_{(a^*)}+\underbrace{\mathbb{E}_{\boldsymbol{\sigma}}\left[\sup_{f\in\mathcal{F}}\frac{1}{n_+}\sum_{i=1}^{n_+}\sigma_i(-2f(\boldsymbol{x}_i))\right]}_{(b^*)}+\underbrace{\mathbb{E}_{\boldsymbol{\sigma}}\left[\sup_{a\in[0,1],s\in\Omega_s}\frac{1}{n_+}\sum_{i=1}^{n_+}\sigma_i(a^2-s)\right]}_{(c^*)}$$

$$+\underbrace{\mathbb{E}_{\boldsymbol{\sigma}}\left[\sup_{f\in\mathcal{F},\gamma\in\Omega_\gamma}\frac{1}{n_+}\sum_{i=1}^{n_+}\sigma_i(-2\gamma f(\boldsymbol{x}_i))\right]}_{(d^*)}+\underbrace{\mathbb{E}_{\boldsymbol{\sigma}}\left[\sup_{f\in\mathcal{F},a\in[0,1]}\frac{1}{n_+}\sum_{i=1}^{n_+}\sigma_i\left(-2af(\boldsymbol{x}_i)\right)\right]}_{(e^*)}+O\left(\frac{1}{\sqrt{n_+}}\right).$$

$$(169)$$

$(o^*)$ is similar to $(o')$. $(l^*)$ follows from the Lem.5 and the fact that $[\cdot]_+$ is 1-Lipschitz continuous. For terms $(a^*)$, $(b^*)$, $(c^*)$, $(d^*)$, $(e^*)$, we have the similar results as terms $(a')$, $(b')$, $(c')$, $(d')$, $(e')$. So we can get:

$$\mathbb{E}_{\boldsymbol{\sigma}}\left[\sup_{f\in\mathcal{F},a\in[0,1],s\in\Omega_s,\gamma\in\Omega_\gamma}\frac{1}{n_+}\sum_{i=1}^{n_+}\sigma_i P(f,a,\gamma,\boldsymbol{x}_i,s)\right]\leq 12\hat{\Re}_+(\mathcal{F})+O\left(\frac{1}{\sqrt{n_+}}\right)\qquad(170)$$

and

$$\sup_{f\in\mathcal{F},a\in[0,1],s\in\Omega_s\gamma\in\Omega_\gamma}\left(\mathbb{E}_{\boldsymbol{x}\sim\mathcal{D}_{\mathcal{P}}}P(f,a,\gamma,\boldsymbol{x},s)-\hat{\mathbb{E}}_{\boldsymbol{x}_i\sim\mathcal{P}}P(f,a,\gamma,\boldsymbol{x}_i,s)\right)$$

$$+\sup_{f\in\mathcal{F},b\in[0,1],s'\in\Omega_{s'},\gamma\in\Omega_\gamma}\left(\mathbb{E}_{\boldsymbol{x}'\sim\mathcal{D}_{\mathcal{N}}}N(f,b,\gamma,\boldsymbol{x}',s')-\hat{\mathbb{E}}_{\boldsymbol{x}'_j\sim\mathcal{N}}N(f,b,\gamma,\boldsymbol{x}'_j,s')\right)$$

$$(171)$$

$$\leq 2\left(12\hat{\Re}_+(\mathcal{F})+O\left(\frac{1}{\sqrt{n_+}}\right)+12\hat{\Re}_-(\mathcal{F})+O\left(\frac{1}{\sqrt{n_-}}\right)\right)+\frac{12}{\alpha}\sqrt{\frac{\log\frac{4}{\delta}}{2n_+}}+\frac{15}{\beta}\sqrt{\frac{\log\frac{4}{\delta}}{2n_-}}$$

$$=O(\hat{\Re}_+(\mathcal{F})+\hat{\Re}_-(\mathcal{F}))+O(\alpha^{-1}n_+^{-1/2}+\beta^{-1}n_-^{-1/2}).$$

For term $(4)$, the same result holds as term $(2)$. For any $\delta>0$, with probability at least $1-\delta$ over the draw of an i.i.d. sample $S$ of positive instances size $n_+$ (negative $n_-$ resp.), each of the following holds for all $f\in\mathcal{F}$:

$$\min_{(a,b)\in[0,1]^2}\max_{\gamma\in\Omega_\gamma}\min_{s\in\Omega_s,s'\in\Omega_{s'}}\mathbb{E}_{\boldsymbol{z}\sim\mathcal{D}_{\mathcal{Z}}}[G_{tp}(f,a,b,\gamma,\boldsymbol{z},s,s')]$$

$$\leq\min_{(a,b)\in[0,1]^2}\max_{\gamma\in\Omega_\gamma}\min_{s\in\Omega_s,s'\in\Omega_{s'}}\hat{\mathbb{E}}_{\boldsymbol{z}\sim S}[G_{tp}(f,a,b,\gamma,\boldsymbol{z},s,s')]+O(\hat{\Re}_+(\mathcal{F})+\hat{\Re}_-(\mathcal{F}))$$

$$+O(\alpha^{-1}n_+^{-1/2}+\beta^{-1}n_-^{-1/2}).$$

$\square$