# OpenReview forum: "Asymptotically Unbiased Instance-wise Regularized Partial AUC Optimization: Theory and Algorithm"
_NeurIPS.cc/2022/Conference — NeurIPS 2022 Accept_

### Official Review · Reviewer_FtWU · 2022-07-07

**Rating:** 8
**Confidence:** 5
**Soundness:** 4 excellent
**Presentation:** 3 good
**Contribution:** 4 excellent

**Summary:**

This paper focuses on optimizing the One-way (Two-way) Partial AUC metric, which is  challenging since a ranking constraint is involved in the objective function. Interestingly, this paper presents a simple instance-wise reformulation of the original objective, which is unbiased in an asymptotic sense. It turns out that the complicated problem could be solved with an accelerated minimax optimization problem. Moreover, the convergence rate can thus be improved. Empirically, the experiments also show its superiority in most cases.

**Questions:**

1) How will the reformulation improve the per-iteration complexity? Show it with experiments.

2) In Sec. 5, the generalization analysis is based on the fact that the (empirical) risk is proportional to the minimax (empirical) risk. I’m wondering if the final result is also proportional to the excess risk R- \hat{R}.

3) The batch size in this paper is set to 1024, which is quite large. I’m wondering how the authors implement this in their experiments.

4)  It is necessary to state some analysis on the  explicit upper bounds for the empirical Rademacher complexity, e.g.,  after Theorem 4.

**Limitations:**

All my concerns are presented in “Weakness” and “Questions”.  This paper focuses on designing an efficient and asymptotically unbiased algorithm for PAUC, which seems no pontential  negative social impact.

**Strengths And Weaknesses:**

Strength:

1) The reformulation of the original problem is  impressive to me, where the ranking constraints are  canceled by conditional expectation and a differentiable reformulation of the top-k (bottom-k) ranking.

2) The generalization analysis is  interesting, where the minimax reformulation can also simplify the derivation of uniform convergence bounds. Moreover, the differentiable formulation also allows the analysis to deal with real-valued hypothesis classes, which previous works often fail to do.

3) Though the convergence analysis is an existing result. It is also good to see that the convergence rate could decrease to O(T^{-3}) due to the reformulation.

Weakness:

1) It seems that there are some typos in the proof. For example, in line 529, in the decomposition of the conditional risk, I think $\ell$ should be replaced with $\ell_{0-1}$. The same problem exists in line 578.

2) In Figures 4-5, I can only see the efficiency improvement in the number of iterations. But the authors also claimed that the reformulation could improve the per-iteration efficiency, which I do agree. I think it may be better if they could give some empirical comparison in terms of this.

---

> ### Author Response · Authors · 2022-08-01
> **Author response to Reviewer FtWU (Part 2/2)**
>
> > **(Q3)** In Sec. 5, the generalization analysis is based on the fact that the (empirical) risk is proportional to the minimax (empirical) risk. I’m wondering if the final result is also proportional to the excess risk $R- \hat{R}$.
>
> **(A3)**: Taking OPAUC as an instance,  the (empirical) risk is also proportional to the minimax (empirical) risk with coefficient $\beta$. To see this, we have:
>
> $$
> \mathcal{R}_{\beta}(f) = \beta \cdot \min _{(a, b) \in[0,1]^{2}} \max _{\gamma \in \Omega _{\gamma}} \min _{s^{\prime} \in \Omega _{s^{\prime}}} \underset{z \sim \mathcal{D} _{\mathcal{Z}}}{\mathbb{E}}\left[G _{o p}\left(f, a, b, \gamma, \boldsymbol{z}, s^{\prime}\right)\right],
> $$
>
> and
>
> $$
> \hat{\mathcal{R}} _{\beta}(f) =
> \beta\cdot
> \min _{(a, b) \in[0,1]^{2}}
> \max _{\gamma \in \Omega _{\gamma} }
> \min _{s^{\prime} \in \Omega _{s^{\prime}}}
> \underset{{z \sim \mathcal{D} _{\mathcal{Z}}}}{\hat{\mathbb{E}}}\left[G _{o p}\left(f, a, b, \gamma, \boldsymbol{z}, s^{\prime}\right)\right].
> $$
>
> Then we get:
>
> $$
> \begin{aligned}
> \mathcal{R} _{\beta}(f)-\hat{\mathcal{R}} _{\beta}(f)=
> \beta\left(\min _{(a, b) \in[0,1]^{2}} \max _{\gamma \in \Omega _{\gamma}} \min _{s^{\prime} \in \Omega _{s^{\prime}}} \underset{z \sim \mathcal{D} _{\mathcal{Z}}}{\mathbb{E}}\left[G _{o p}\left(f, a, b, \gamma, \boldsymbol{z}, s^{\prime}\right)\right]\right.\\\\
> \left.-\min _{(a, b) \in[0,1]^{2}}
> \max _{\gamma \in \Omega _{\gamma} }
> \min _{s^{\prime} \in \Omega _{s^{\prime}}}
> \underset{{z \sim \mathcal{D} _{\mathcal{Z}}}}{\hat{\mathbb{E}}}\left[G _{o p}\left(f, a, b, \gamma, \boldsymbol{z}, s^{\prime}\right)\right])\right).
> \end{aligned}
> $$
>
> So the final result is also proportional to the excess risk with coefficient $\beta$. The result is similar to TPAUC.
>
> > **(Q4)** The batch size in this paper is set to 1024, which is quite large. I’m wondering how the authors implement this in their experiments.
>
> **(A4)**:  We use `torch.nn.DataParallel` to implement on four RTX 3090 GPUs.
>
> > **(Q5)** It is necessary to state some analysis on the explicit upper bounds for the empirical Rademacher complexity, e.g., after Theorem 4.
>
> **(A5)**:  Thank you so much for your suggestion! Under a standard setting, the upper bound for the empirical Rademacher complexities $ \hat{\mathfrak{R}} _+, \hat{\mathfrak{R}} _- $ are of order $O((1/n _+) ^{-1/2}), O((1/n _-) ^{-1/2})$, respectively.  This is true for many hypothesis classes, including linear models and deep neural networks (see the references below). So the overall sample complexity should be $O((1/n _+) ^{-1/2}), O((1/n _-) ^{-1/2})$. We will include this discussion after Thm.4.
>
> - Schapire, Robert E., and Yoav Freund. "Foundations of machine learning." (2012): 23-52.
>
> - Golowich N, Rakhlin A, Shamir O. Size-independent sample complexity of neural networks, COLT 2018: 297-299.

---

> ### Author Response · Authors · 2022-08-01
> **Author response to Reviewer FtWU (Part 1/2)**
>
> Thanks for your valuable suggestions! The replies to your concerns are attached below.
>
> > **(Q1)**  There seem to have typos in the proof. For example, in line 529, in the decomposition of the conditional risk, I think $\ell$ should be replaced with $\ell_{0-1}$. The same problem exists in line 578.
>
> **(A1):** We are sorry for the typos. We will correct them in the next version. Thank you so much for your cross-check!
>
> > **(Q2)** In Figures 4-5, I can only see the efficiency improvement in the number of iterations. But the authors also claimed that the reformulation could improve the per-iteration efficiency, which I do agree with. I think it may be better if they could give some empirical comparison in terms of this. How will the reformulation improve the per-iteration complexity? Show it with experiments.
>
> **(A2):** Thank you for your suggestion! We conduct some experiments for per-iteration complexity with a fixed epoch with varying $n_+, n_-$. All experiments are conducted on an Ubuntu 16.04.1 server with an Intel(R) Xeon(R) Silver 4110 CPU. For every method, we repeat running 10000 times and record the average running time. We only record the loss calculation time and use the python package `time.time()` to calculate the running time. Methods with * stand for the pair-wise estimator, while methods with ** stand for the instance-wise estimator. Here is the result of the experiment. We see the acceleration is significant when the data is large.
>
> OPAUC ($\mathrm{FPR}\leq0.3$)
>
> | unit:ms   | $\begin{aligned}n_+=64\\\\n_-=64\end{aligned}$ | $\begin{aligned}n_+=128\\\\n_-=128\end{aligned}$ | $\begin{aligned}n_+=256\\\\n_-=256\end{aligned}$ | $\begin{aligned}n_+=512\\\\n_-=512\end{aligned}$ | $\begin{aligned}n_+=1024\\\\n_-=1024\end{aligned}$ | $\begin{aligned}n_+=2048\\\\n_-=2048\end{aligned}$ |
> |:---------:|:----------------:|:------------------:|:------------------:|:------------------:|:--------------------:|:--------------------:|
> | SOPA*     | 0.075            | 0.205              | 1.427              | 5.053              | 20.132               | 86.779               |
> | SOPA-S*     | 0.063            | 0.165              | 0.946              | 4.003              | 15.815               | 62.031               |
> | AGD-SBCD*      | 0.061            | 0.145              | 1.040              | 3.413              | 13.273               | 54.954               |
> | AUC-poly* | 0.062            | 0.178              | 1.086              | 3.553              | 14.266               | 56.637               |
> | AUC-exp*  | 0.063            | 0.182              | 0.985              | 3.513              | 14.155               | 55.689               |
> | MB*       | 0.121            | 0.174              | 0.468              | 1.713              | 6.393                | 25.663               |
> | PAUCI**   | 0.026            | 0.029              | 0.033              | 0.043              | 0.072                | 0.107                |
> | AUC-M**   | 0.025            | 0.028              | 0.031              | 0.040              | 0.059                | 0.104                |
> | CE**      | 0.018            | 0.020              | 0.026              | 0.036              | 0.055                | 0.096                |
>
> TPAUC ($\mathrm{FPR}\leq0.5,\mathrm{TPR}\geq0.5$)
>
> | unit:ms   | $\begin{aligned}n_+=64\\\\n_-=64\end{aligned}$ | $\begin{aligned}n_+=128\\\\n_-=128\end{aligned}$ | $\begin{aligned}n_+=256\\\\n_-=256\end{aligned}$ | $\begin{aligned}n_+=512\\\\n_-=512\end{aligned}$ | $\begin{aligned}n_+=1024\\\\n_-=1024\end{aligned}$ | $\begin{aligned}n_+=2048\\\\n_-=2048\end{aligned}$ |
> |:---------:|:----------------:|:------------------:|:------------------:|:------------------:|:--------------------:|:--------------------:|
> | SOPA*     | 0.079            | 0.206              | 1.439              | 5.197              | 20.556               | 88.314               |
> | SOPA*     | 0.065            | 0.153              | 0.947              | 3.940              | 15.388               | 62.541               |
> | AUC-poly* | 0.062            | 0.180              | 1.175              | 3.573              | 14.440               | 56.469               |
> | AUC-exp*  | 0.059            | 0.206              | 1.154              | 3.558              | 14.080               | 56.566               |
> | MB*       | 0.173            | 0.198              | 0.491              | 1.955              | 6.554                | 29.369               |
> | PAUCI**   | 0.030            | 0.030              | 0.038              | 0.045              | 0.071                | 0.109                |
> | AUC-M**   | 0.025            | 0.027              | 0.033              | 0.043              | 0.059                | 0.104                |
> | CE**      | 0.018            | 0.021              | 0.026              | 0.037              | 0.0535               | 0.096                |

---

### Official Review · Reviewer_TVKy · 2022-07-10

**Rating:** 8
**Confidence:** 5
**Soundness:** 4 excellent
**Presentation:** 3 good
**Contribution:** 4 excellent

**Summary:**

This paper proposes novel algorithms to improve the efficiency of partial AUC Optimization. Specifically, they present a reformulation scheme to transform the pairwise indifferentiable objective function into an instance-wise differentiable with an approximation scheme. Moreover, they provide generalization and optimization guarantees for their proposed method. The extensive experiment in this paper shows that the proposed method can outperform the state-of-art most times.

**Questions:**


All my concerns are shown in the previous question. So, I will only list some suggestions.

1.The methodology could be polished to be more friendly for the general readers.
2. Since this paper is not the first PAUC optimization method. The related work should be moved to the main paper. This would be helpful for the readers to get the key contribution of the paper.
3. There are some typos to be corrected, for example:
(1) In the keywords, please change “min max” to “mini-max”
(2) Line 168, “this problem。” should be “these problems.”
(3)Line 125-129 is a bit redundant. Please rephrase it.


**Limitations:**

YES

**Strengths And Weaknesses:**

Pros:

This paper presents an efficient reformulation scheme to make a complicated problem much more practical to solve. In other words, both the number of epochs and the per-iteration running time could be reduced significantly.

This proposed method also has a strong and comprehensive theoretical guarantee in terms of convergence and generalization. Moreover, technical details are non-trivial. I believe these merits can benefit the audience from a broad range of the ML community.

The experiments are extensive. Most of the competitors are quite SOTA.

The paper presents a solid work with the possibility to be employed in real-world problems. I only have some minor concerns, which I hope can be addressed during the rebuttal.

Cons:

The math is dense even in the main paper. Though I can understand most of the details, I think the authors can add more details and intuitive content to guide readers unfamiliar with AUC.

I only see the performance comparisons in the main paper. I think efficiency is more important in this paper since the goal is to accelerate. So, I would also like to the running time comparisons in the experiments.

---

> ### Author Response · Authors · 2022-08-01
> **Author response to Reviewer TVKy**
>
> Thanks for your valuable suggestions! The replies to your concerns are attached below.
>
> > **(Q1)**  The math is dense even in the main paper. Though I can understand most of the details, I think the authors can add more details and intuitive content to guide readers unfamiliar with AUC.
>
> **(A1)**:  Thank you for your great suggestions! We will include more details about math and AUC. In addition, more intuitions about the theorems and lemmas will be introduced in Introduction and Preliminaries.
>
> > **(Q2)** I only see the performance comparisons in the main paper. I think efficiency is more important in this paper since the goal is to accelerate. So, I would also like to the running time comparisons in the experiments.
>
> **(A2)**:  Thank you for your suggestion! We conduct some experiments for per-iteration complexity with a fixed epoch with varying $n_+, n_-$. All experiments are conducted on an Ubuntu 16.04.1 server with an Intel(R) Xeon(R) Silver 4110 CPU (to get rid of the affect of parallel computing). For every method, we repeat running 10,000 times and record the average running time. We only record the loss calculation time and use the python package `time.time()` to calculate the running time. Methods with * stand for the pair-wise estimator, while methods with ** stand for the instance-wise estimator. Here is the result of the experiment. We see the acceleration is significant when the data is large.
>
>
> OPAUC ($\mathrm{FPR}\leq0.3$)
>
>
> | unit:ms   | $\begin{aligned}n_+=64\\\\n_-=64\end{aligned}$ | $\begin{aligned}n_+=128\\\\n_-=128\end{aligned}$ | $\begin{aligned}n_+=256\\\\n_-=256\end{aligned}$ | $\begin{aligned}n_+=512\\\\n_-=512\end{aligned}$ | $\begin{aligned}n_+=1024\\\\n_-=1024\end{aligned}$ | $\begin{aligned}n_+=2048\\\\n_-=2048\end{aligned}$ |
> |:---------:|:----------------:|:------------------:|:------------------:|:------------------:|:--------------------:|:--------------------:|
> | SOPA*     | 0.075            | 0.205              | 1.427              | 5.053              | 20.132               | 86.779               |
> | SOPA-S*     | 0.063            | 0.165              | 0.946              | 4.003              | 15.815               | 62.031               |
> | AGD-SBCD*      | 0.061            | 0.145              | 1.040              | 3.413              | 13.273               | 54.954               |
> | AUC-poly* | 0.062            | 0.178              | 1.086              | 3.553              | 14.266               | 56.637               |
> | AUC-exp*  | 0.063            | 0.182              | 0.985              | 3.513              | 14.155               | 55.689               |
> | MB*       | 0.121            | 0.174              | 0.468              | 1.713              | 6.393                | 25.663               |
> | PAUCI**   | 0.026            | 0.029              | 0.033              | 0.043              | 0.072                | 0.107                |
> | AUC-M**   | 0.025            | 0.028              | 0.031              | 0.040              | 0.059                | 0.104                |
> | CE**      | 0.018            | 0.020              | 0.026              | 0.036              | 0.055                | 0.096                |
>
> TPAUC ($\mathrm{FPR}\leq0.5,\mathrm{TPR}\geq0.5$)
>
> | unit:ms   | $\begin{aligned}n_+=64\\\\n_-=64\end{aligned}$ | $\begin{aligned}n_+=128\\\\n_-=128\end{aligned}$ | $\begin{aligned}n_+=256\\\\n_-=256\end{aligned}$ | $\begin{aligned}n_+=512\\\\n_-=512\end{aligned}$ | $\begin{aligned}n_+=1024\\\\n_-=1024\end{aligned}$ | $\begin{aligned}n_+=2048\\\\n_-=2048\end{aligned}$ |
> |:---------:|:----------------:|:------------------:|:------------------:|:------------------:|:--------------------:|:--------------------:|
> | SOPA*     | 0.079            | 0.206              | 1.439              | 5.197              | 20.556               | 88.314               |
> | SOPA-S*     | 0.065            | 0.153              | 0.947              | 3.940              | 15.388               | 62.541               |
> | AUC-poly* | 0.062            | 0.180              | 1.175              | 3.573              | 14.440               | 56.469               |
> | AUC-exp*  | 0.059            | 0.206              | 1.154              | 3.558              | 14.080               | 56.566               |
> | MB*       | 0.173            | 0.198              | 0.491              | 1.955              | 6.554                | 29.369               |
> | PAUCI**   | 0.030            | 0.030              | 0.038              | 0.045              | 0.071                | 0.109                |
> | AUC-M**   | 0.025            | 0.027              | 0.033              | 0.043              | 0.059                | 0.104                |
> | CE**      | 0.018            | 0.021              | 0.026              | 0.037              | 0.0535               | 0.096                |

---

### Official Review · Reviewer_m7Rj · 2022-07-11

**Rating:** 5
**Confidence:** 5
**Soundness:** 2 fair
**Presentation:** 2 fair
**Contribution:** 1 poor

**Summary:**

The paper proposes a nonconvex strongly concave min-max formulation for OPAUC and TPAUC maximization and employs a stochastic min-max algorithm with $O(\epsilon^{-3})$ complexity.

**Questions:**

1. In Table 1, the proposed algorithm is compared with [39], but difference between the paper and [39] is not mentioned in Introduction part. This paper approximates the PAUC as a non-convex strongly concave problem and then solves the approximated problem, while [39] solves the original PAUC problem. Therefore, the comparison of their convergence rates is not fair enough. Moreover, it would be more convincing to see some numerical results comparison with [39].
2. The Assumption 1 requires G has Lipschitz continuous gradients for any data, which is a very strong assumption and $L_G$ can be infinity in this assumption. Is it possible to use a weaker assumption (e.g. the expectation of gradient is Lipschitz continuous) and get the same convergence rate?
3. Can this formulation conversion be used to optimize PAUC in a range of FPR/TPR?

**Limitations:**

Please see above.

**Strengths And Weaknesses:**

Strengths:
1. The paper is well-organized and written clearly.
2. The formulation conversion of PAUC is novel.

Weaknesses:
1. The paper employs an algorithm with a very strong (bad) assumption. $L_G$ in Assumption 1 can be infinity.
2. Contribution is not significant enough.

---

> ### Author Response · Authors · 2022-08-01
> **Author response to Reviewer m7Rj (Part 2/2)**
>
> > **(Q5)**  Assumption 1 requires Lipschitz continuous gradients for every data, which is a very strong assumption, and $L_G$ can be infinity in this assumption. Is it possible to use a weaker assumption (e.g., the gradient expectation is Lipschitz continuous) and get the same convergence rate?
>
> **(A5)**: Thank you for your valuable comments! For our case, the assumption is equivalent to assuming that the scoring function has Lipschitz continuous gradients for every data. Such assumptions are widely used in recent studies about non-convex optimizations, see Ref.[1-9].
>
> Moreover, some recent studies have started discussing the condition under which the assumption holds. For example, Ref.[9] states that the gradient Lipschitz assumption for scoring function assumption is satisfied if the following conditions hold:
>
> 1. The neural network uses the smooth activation function ($\textit{i.e.}$, Linear function, Logistic function, Softplus function, Tanh function).
>
> 2. The output score is bounded with Sigmoid function $\sigma(x)=1/(1+\exp(-f(\boldsymbol{x}))$ (the input will be scaled into closed set $[0,1]$).
>
> 3. The parameter of neural network $\boldsymbol{w}$ is bounded ($\textit{e.g.}$, $\|\boldsymbol{w}\|^2\leq M$).
>
> Ref. [10] estimates the gradient Lipschitz constants of the output of the deep neural network. They propose a general estimation for the upper bound on the Lipschitz constant of the gradient of any loss function to the parameters. The results show that such constants are finite under mild conditions.
>
> ##### REFERENCE
>
> 1. Guo, Zhishuai, et al. "Randomized stochastic variance-reduced methods for multi-task stochastic bilevel optimization." *arXiv preprint arXiv:2105.02266* (2021). **[Assumption 1, 2]**
>
> 2. Hu, Quanqi, Yongjian Zhong, and Tianbao Yang. "Multi-block Min-max Bilevel Optimization with Applications in Multi-task Deep AUC Maximization." *arXiv preprint arXiv:2206.00260* (2022). **[Assumption 2.2]**
>
> 3. Jiang, Wei, et al. "Multi-block-Single-probe Variance Reduced Estimator for Coupled Compositional Optimization." *arXiv preprint arXiv:2207.08540* (2022). **[Assumption 1]**
>
> 4. Qi, Qi, et al. "Attentional biased stochastic gradient for imbalanced classification." *arXiv preprint arXiv:2012.06951* (2020). **[Assumption 1]**
>
> 5. Wang, Bokun, and Tianbao Yang. "Finite-Sum Coupled Compositional Stochastic Optimization: Theory and Applications." *International Conference on Machine Learning*. PMLR, 2022. **[Assumption 1]**
>
> 6. Luo, Luo, et al. "Stochastic recursive gradient descent ascent for stochastic nonconvex-strongly-concave minimax problems." *Advances in Neural Information Processing Systems* 33 (2020): 20566-20577. **[Assumption 2]**
>
> 7. Chen, Tianyi, Yuejiao Sun, and Wotao Yin. "A single-timescale stochastic bilevel optimization method." *arXiv preprint arXiv:2102.04671* (2021). **[Assumption 1]**
>
> 8. Gao, Hongchang, Bin Gu, and My T. Thai. "Stochastic Bilevel Distributed Optimization over a Network." *arXiv preprint arXiv:2206.15025* (2022).**[Assumption 4]**
>
> 9. Zhu, Dixian, et al. "When AUC meets DRO: Optimizing Partial AUC for Deep Learning with Non-Convex Convergence Guarantee." arXiv preprint arXiv:2203.00176 (2022). **[Assumption 2]**
>
> 10. Herrera, Calypso, Florian Krach, and Josef Teichmann. "Estimating full lipschitz constants of deep neural networks." arXiv preprint arXiv:2004.13135 (2020).
>
> > **(Q6)** Can this formulation conversion be used to optimize PAUC in a range of FPR/TPR?
>
> **(A6):** We can use the sum-of-ranged loss proposed in the following work to get a similar reformulation. However, due to the space and time limitation, we will leave it as future work.
>
> - Hu, Shu, Yiming Ying, and Siwei Lyu. "Learning by minimizing the sum of ranked range." Advances in Neural Information Processing Systems 33 (2020): 21013-21023.

---

> > ### Comment · Reviewer_m7Rj · 2022-08-04
> > **To Paper1420 Authors**
> >
> > To Authors, all Reviewers and the AC.
> >
> > Thank you for your replies. After reading your feedback and the paper again, I still have some concerns and questions.
> >
> > **1. Theorem 3 is incorrect and the convergence cannot be guaranteed.**
> >
> > In **Theorem 1** and **Theorem 2**, the constraint set $\Omega_\gamma$ is $[b-1, 1]$ which depends on $b$, a decision variable of the outer minimization problem. This is a very big issue. First, for a given $b$, the optimal $\gamma$ for the inner maximization depends on $b$. As a result, $b^*$ in the outer optimal solution is not guaranteed to be (53). Second, Algorithm 1 cannot be applied to a min-max problem where the constraint of inner maximization depends on the decision variables of the outer maximization problem. The convergence result in **Theorem 3** (Theorem 9 [15]) also fail because of that. I don't think the authors can easily fix this issue because a min-max problem with coupling constraints is a fundamentally more difficult in optimization, and [15] is not the right method to apply. Moreover, it seems restricting $\gamma$ in $[b-1, 1]$ is the key to ensure **Proposition 1** (monotonicity of $\ell_1(x')$), which is needed to develop **Theorem 2**. Given that restricting $\gamma$ in $[b-1, 1]$ is not allowed, **Theorem 2** is also not meaningful neither. The same problems also happen to two-way partial AUC maximization (**Proposition 2** and **Theorem 6**). Hence, the entire paper is wrong. I want to draw all reviewers' attention to this issue.
> >
> > **2. There are some problems with Table 1 and the comparison in Table 1 is unfair.**
> >
> > (1) In the line of "Per-Iteration Time Complexity", all of the algorithms listed in *Table 1* are stochastic algorithms. Would you please explain the reason that stochastic algorithms have $O(n\_+n\_-)$ per-iteration complexity with $n\_+$ representing the total number of positive instances?
> >
> > (2) The names of algorithms in Table 1 are not consistent with reference paper. It is confused to use different names. In [43], the algorithm for one way partial AUC is called **SOPA** or **SOPA-S** and the algorithm for two way partial AUC is called **SOTA-S**. In [39], the algorithm is called **AGD-SBCD**.
> >
> > (3) The comparison with [43] in Table 1 is unfair. For [43], the convergence rate is developed for finding a stationary point of the original problem with $[\cdot]\_+$, which is **non-smooth**. However, in this paper, the convergence rate is for finding a stationary point of a **smoothed** problem. The smoothed problem in general may **not** have the same stationary point or optimal solution as the original non-smooth problem, so  $\epsilon$ has different meanings in [43] and this paper. Hence, the convergence rates of these two algorithms are not comparable.
> >
> > (4) The comparison with [39] in Table 1 is unfair. [39] is focused on solving partial AUC when FPR is in a fixed range. The fixed range partial AUC cannot be simply classified as OPAUC. However, it is unclear the proposed algorithm can be applied to optimizing pAUC with FPR in a range $[\alpha, \beta]$. Hence, it is unfair to compare with [39] in Table 1.
> >
> > (5) The statement about the convergence for [38] is wrong. Given that [38] transfers the problem into standard minimization problems, hence, for general non-convex smooth optimization under bounded variance condition SGD has a complexity of $O(1/\epsilon^4)$ and under Lipschitz stochastic gradient condition variance reduced methods (e.g., STORM used in the paper) has a complexity of $O(1/\epsilon^3)$. It is not clear why the authors claim the complexity of $O(1/\epsilon^2)$ for [38].
> >
> > **4. Concerning the numerical result comparison with [39].**
> >
> > Thank you for implementing numerical comparison with [39]. However, [39] is designed for partial AUC **in a fixed range**, not simply OPAUC. Hence, it would be more reasonable to implement partial AUC in a fixed range. Moreover, the table shows that the algorithm in this paper can achieve better testing performance, which does not mean that this algorithm can converge faster. Convergence curves are preferable.
> >
> > **5. Formulations (8), (14), (22) are wrong. All of them miss components $p$ and $1-p$.**
> >
> > Given the above reasons, I will lower my score to 1 unless the authors can address the issues including the fatal error.

---

> > > ### Author Response · Authors · 2022-08-08
> > > **Author response to Reviewer m7Rj (Part 5/5)**
> > >
> > > > **(Q4)** There are some problems with Table 1 and the comparison in Table 1 is unfair.
> > >
> > >  **(A4)**: We have carefully revised table 1. The new version is attached as follows:
> > >
> > > >**Table 1** Comparison with existing partial AUC algorithms. The convergence rate represents the number of iterations after which an algorithm can find an $\epsilon$-stationary point, where $\epsilon$-sp is $\epsilon$-stationary point and.  $\triangle$ implies a natural result of non-convex SGD. $n_+^B$($n_-^B$ resp.) is the number of positive (negative resp.) instances for each mini-batch $B$. $\kappa \rightarrow \infty, \omega \rightarrow 0$ implies that our method is unbiased under such a asymptotic condition.
> > >
> > > ||SOPA|SOPA-S|TPAUC|Ours|
> > > |-------------------------------|--------------------|--------------------|--------------------------------|--------------------------------------------------------------------------------|
> > > |ConvergenceRate(OPAUC)|$O(\epsilon^{-4})$|$O(\epsilon^{-4})$|$O(\epsilon^{-4})^{\triangle}$|$O(\epsilon^{-3})$|
> > > |ConvergenceRate(TPAUC)|$O(\epsilon^{-6})$|$O(\epsilon^{-4})$|$O(\epsilon^{-4})^{\triangle}$|$O(\epsilon^{-3})$|
> > > |ConvergenceMeasure|$\epsilon$-sp(non-smooth)|$\epsilon$-sp|$\epsilon$-sp|$\epsilon$-sp|
> > > |Smoothness|non-smooth|smooth|smooth|smooth|
> > > |OPAUC&TPAUC|$\surd$|$\surd$|$\surd$|$\surd$|
> > > |Unbiased|$\surd$|X|X|$\begin{aligned}\kappa\rightarrow\infty\\\\ \omega\rightarrow0\end{aligned}$|
> > > |Per-IteratoinTimeComplexity|$O(n_+^Bn_-^B)$|$O(n_+^Bn_-^B)$|$O(n_+^Bn_-^B)$|$O(n_+^B+n_-^B)$|
> > >
> > >
> > >
> > > > **(Q4.1)**  $n_+$ representing the total number of positive instances?
> > >
> > >  **(A4.1)** We are sorry about the unclear notation. Now we use $n^B_+, and ~n^B_-$ to represent the number of positive/negative instances in each mini-batch.
> > >
> > > > **(Q4.2)**  The names of algorithms in Table 1 are not consistent with reference paper.
> > >
> > >  **(A4.2)** We are sorry about the inconsistent algorithm names. We have modified their names to avoid inconsistency.
> > >
> > > > **(Q4.3)**  The comparison with [43] in Table 1 is unfair.
> > >
> > >  **(A4.3)** Thank you for your correction. To make it fair, we have included more information in table 1.
> > >
> > > > **(Q4.4)**  The comparison with [39] in Table 1 is unfair.
> > >
> > >  **(A4.4)** Thank you for your correction. We won't comparison with [39] in the table. Now table 1 only contains algorithms which can optimize both OPAUC and TPAUC. Moreover, we include the empirical comparisons on OPAUC metrics within a fixed range (please see the answer for **(Q5)** below).
> > >
> > > > **(Q4.5)**  The statement about the convergence for [38] is wrong.
> > >
> > >  **(A4.5)** Thank you for your correction. The right convergence rate should be $O(\epsilon^{-4})$. We will fix it in the next version.
> > >
> > >
> > >
> > > > **(Q5)** Concerning the numerical result comparison with [39].
> > >
> > >
> > >  **(A5)** Thank you for your suggestion. We conduct the following experiments and restrict algorithms within a fixed FPR range. All experiments are conducted on an Ubuntu 16.04.1 server with an Intel(R) Xeon(R) Silver 4110 CPU and four RTX 3090 GPUS. For AGD-SBCD, all parameters are tuned by following the [39]'s experiments. Other experiment setups are the same as our paper.
> > >
> > > > **Performance Experiment**
> > > The results show that our algorithm can reach comparable performance with AGD-SBCD in this case.
> > >
> > > OPAUC ($0.1 \leq \mathrm{FPR}\leq 0.3$)
> > > | methods  | cifar-10-1 | cifar-10-2 | cifar-10-3 | cifar-100-1 | cifar-100-2 | cifar-100-3 | tniy-im200-1 | tiny-im200-2 | tiny-200-3 |
> > > | -------- | ---------- | ---------- | ---------- | ----------- | ----------- | ----------- | ------------ | ------------ | ---------- |
> > > | AGD-SBCD | 0.8482     | **0.9932** | 0.8510     | **0.9565**  | **0.9885**  | 0.8859      | 0.9147       | 0.9611       | **0.9666** |
> > > | PAUCI    | **0.8560** |   0.9907   | **0.8615** | 0.9517  | 0.9863      | **0.8918**  | **0.9231**   | **0.9673**   | 0.9542     |
> > >
> > >
> > >
> > > >### **Convergence Experiment**
> > >
> > > We show the convergence comparison here in the following table on the training data. It could be seen that our algorithm start to enter a stable state after epoch 30, while AGD-SBCD starts to enter a stable state after epoch 40.
> > >
> > >
> > > OPAUC ($0.1 \leq \mathrm{FPR}\leq 0.3$, cifar-10-1)
> > > | methods  | epochs:5   | epochs:10   | epochs:15  | epochs:20   | epochs:25   | epochs:30   | epochs:35    | epochs:40    | epochs:45  | epochs:50  |
> > > | -------- | ---------- | ---------- | ---------- | ----------- | ----------- | ----------- | ------------ | ------------ | ---------- | ---------- |
> > > | AGD-SBCD | 0.8493 | 0.8777 | 0.8894   | 0.9011  | 0.8944  | 0.8990  | 0.9033    | 0.9073   | 0.9060 | 0.9080 |
> > > | PAUCI    | 0.8624 | 0.8853 | 0.8933   | 0.9003  | 0.9089  | 0.9115  | 0.9126    | 0.9104   | 0.9139 | 0.9127 |

---

> > > > ### Comment · Reviewer_m7Rj · 2022-08-09
> > > > **To Paper1420 Authors**
> > > >
> > > > Thank you very much for your detailed response and it solves my problem! I checked the rebuttal revision and noticed that you have fixed most theorems and unfair tables. However, there are some places still needed to be fixed. If you can submit a new version with following modifications, I would raise my score.
> > > >
> > > > 1. Since you have removed the unfair comparison with [39], the complexity comparison in **abstract** should also be removed.
> > > >
> > > > 2. To be consistent with the Table 1, I would suggest you fix some typos about algorithms' names in Sections 6.2 and 6.3.
> > > >
> > > > 3. It is claimed that your algorithm can be applied to OPAUC optimization within a fixed FPR range (Line 225-227). To show that, formulation of partial AUC within a fixed FPR range should be provided in the paper. Otherwise, I would suggest removing Table 4.
> > > >
> > > > 4. The page limit of the paper is 9 pages.

---

> > > > > ### Author Response · Authors · 2022-08-09
> > > > > **Reply to Reviewer m7Rj**
> > > > >
> > > > > Thank you so much for your suggestion!
> > > > >
> > > > > 1. We have changed the words in the abstract to
> > > > >
> > > > > > we employ an efficient solver that enjoys a linear per-iteration computational complexity w.r.t. the sample size and a time-complexity of $O(\epsilon^{-1/3})$ to reach a $\epsilon$ stationary point.
> > > > >
> > > > > where the comparison is removed
> > > > >
> > > > > 2. We have corrected the name in Sec.6.2 and 6.2. Please check the main paper.
> > > > >
> > > > > 3.  We have removed Tab.4 as you suggested.
> > > > >
> > > > > 4. Now our main paper is less than 9 page. According to the instruction of NeurIPS,  references and the checklist should not be counted.

---

> > > > > > ### Comment · Reviewer_m7Rj · 2022-08-09
> > > > > > **Response to authors' reply**
> > > > > >
> > > > > > Thanks for making modifications. I raised my score since the authors provide solid proof to fix the problems and make necessary modifications in the paper.

---

> > > ### Author Response · Authors · 2022-08-08
> > > **Author response to Reviewer m7Rj (Part 4/5)**
> > >
> > > > **(Q3)** The problem of Theorem 3
> > >
> > > ##### REFERENCE
> > >
> > > > [A] Tsaknakis, Ioannis, Mingyi Hong, and Shuzhong Zhang. "Minimax problems with coupled linear constraints: computational complexity, duality and solution methods." arXiv:2110.11210 (2021).
> > >
> > > **(A3)** Thank you so much for posing such a key issue! We have realized the problem. However, according to the recent work of **Ref.[A]**. We also reformulated it as an off-the-shelf mini-max problem where the coupled constraint is replaced with the Lagrange multipliers ($\theta\_b$ for OPAUC, $\theta\_b, \theta\_a$ for TPAUC). Specifically, by Thm.2 in Ref.[A], we have, for OPAUC:
> > >
> > > $$
> > > \begin{aligned}
> > > {\min\_{f,(a,b)\in[0,1]\^2,s\in\Omega\_{s}}\max\_{\gamma\in[b-1,1]} \mathbb{E}\_{\boldsymbol{z}\sim \mathcal{D}\_\mathcal{Z}}[G\_{op}\^{\kappa,\omega}]}=\min\_{f,(a,b)\in[0,1]\^2,s\in\Omega\_{s},\theta\_b \ge 0}\max\_{\gamma\in[-1,1]} \mathbb{E}\_{\boldsymbol{z}\sim \mathcal{D}\_\mathcal{Z}}[G\_{op}\^{\kappa,\omega}] - \theta\_{b}(b-1-\gamma),
> > > \end{aligned}
> > > $$
> > >
> > > and for $\mathrm{TPAUC}$:
> > > $$
> > > {\min\_{f,(a,b)\in[0,1]\^2,s\in\Omega\_{s},s'\in\Omega\_{s'}}~~\max\_{\gamma\in[\max\\{-a,b-1\\},1]} \mathbb{E}\_{\boldsymbol{z}\sim \mathcal{D}\_\mathcal{Z}}[G\_{tp}\^{\kappa,\omega}]}= \min\_{f,(a,b)\in[0,1]\^2,s\in\Omega\_{s},s'\in\Omega\_{s'},\theta\_a \ge 0,\theta\_b \ge 0}\max\_{\gamma\in[-1,1]} \mathbb{E}\_{\boldsymbol{z}\sim \mathcal{D}\_\mathcal{Z}}[G\_{tp}\^{\kappa,\omega}] \\\\- \theta\_{b}(b-1-\gamma)- \theta\_{a}(-a-\gamma)
> > > $$
> > >
> > > Moreover, since the original loss function is bounded from below/above, it is easy to show that the optimal solution for the multipliers $\theta^*_b, \theta^*_a$ satisfies $\theta^*_b \le \infty, \theta^*_a \le \infty$. Hence, there must exist a sufficiently large $M>0$, such that:
> > >
> > > $$
> > > {\min\_{f,(a,b)\in[0,1]\^2,s\in\Omega\_{s}}\max\_{\gamma\in[b-1,1]} \mathbb{E}\_{\boldsymbol{z}\sim \mathcal{D}\_\mathcal{Z}}[G\_{op}\^{\kappa,\omega}]}=\min\_{f,(a,b)\in[0,1]\^2,s\in\Omega\_{s},\theta\_b \in [0,M\_1]}\max\_{\gamma\in[-1,1]} \mathbb{E}\_{\boldsymbol{z}\sim \mathcal{D}\_\mathcal{Z}}[G\_{op}\^{\kappa,\omega}] - \theta\_{b}(b-1-\gamma),
> > > $$
> > > and for $\mathrm{TPAUC}$:
> > > $$
> > > {\min\_{f,(a,b)\in[0,1]\^2,s\in\Omega\_{s}}~~\max\_{\gamma\in[\max\\{-a,b-1\\},1]} \mathbb{E}\_{\boldsymbol{z}\sim \mathcal{D}\_\mathcal{Z}}[G\_{tp}\^{\kappa,\omega}]}= \min\_{f,(a,b)\in[0,1]\^2,s\in\Omega\_{s},s'\in\Omega\_{s'},\theta\_a \in [0,M\_2],\theta\_b \in [0,M\_3]}~~~\max\_{\gamma\in[-1,1]} \mathbb{E}\_{\boldsymbol{z}\sim \mathcal{D}\_\mathcal{Z}}[G\_{tp}\^{\kappa,\omega}] \\\\- \theta\_{b}(b-1-\gamma)- \theta\_{a}(-a-\gamma)
> > > $$
> > >
> > > Here, to make sure that $M_1,M_2,M_3$ are large enough, we set them as $M_1 = M_2 =M_3 =10^9$.
> > >
> > >
> > > Based on the correction of $(OP2)$ and $(OP4)$, we conduct the experiments again. Here is the result (The highest and the second best results are bolded and italic, respectively), which shows that our algorithm still outperforms the competitors for the following OPAUC and TPAUC metrics.
> > >
> > > OPAUC($\mathrm{FPR}\leq0.3$)
> > > |methods|cifar-10-1|cifar-10-2|cifar-10-3|cifar-100-1|cifar-100-2|cifar-100-3|tniy-im200-1|tiny-im200-2|tiny-200-3|
> > > |--------|----------|----------|----------|-----------|-----------|-----------|------------|------------|----------|
> > > |SOPA|*0.7659*|*0.9688*|**0.7651**|*0.9108*|*0.9875*|0.8483|0.8157|0.9037|0.9066|
> > > |SOPA-S|0.7548|0.9674|*0.7542*|0.9033|0.9860|0.8449|0.8180|*0.9087*|0.9095|
> > > |AGD-SBCD|0.7526|0.9615|0.7497|0.9105|0.9814|0.8406|0.8135|0.9081|0.9057|
> > > |AUC-poly|0.7542|0.9672|0.7538|0.9027|0.9859|0.8441|0.8185|0.9084|*0.9100*|
> > > |AUC-exp|0.7347|0.9620|0.7457|0.8987|0.9850|0.8407|0.8127|0.9026|0.9049|
> > > |CE|0.7417|0.9431|0.7428|0.8903|0.9695|0.8321|0.8023|0.8917|0.8878|
> > > |MB|0.7492|0.9648|0.7500|0.9003|0.9804|**0.8575**|*0.8193*|0.9072|0.9091|
> > > |AUC-M|0.7334|0.9609|0.7442|0.8996|0.9845|0.8403|0.8102|0.9011|0.9043|
> > > |PAUCI|**0.7721**|**0.9716**|0.7399|**0.9155**|**0.9889**|*0.8492*|**0.8267**|**0.9214**|**0.9217**|
> > >
> > > TPAUC($\mathrm{FPR}\leq0.5,\mathrm{TPR}\geq0.5$)
> > > |methods|cifar-10-1|cifar-10-2|cifar-10-3|cifar-100-1|cifar-100-2|cifar-100-3|tniy-im200-1|tiny-im200-2|tiny-200-3|
> > > |--------|----------|----------|----------|-----------|-----------|-----------|------------|------------|----------|
> > > |SOPA|*0.7096*|*0.9593*|*0.7220*|*0.8714*|*0.9855*|0.7485|*0.7417*|*0.8681*|*0.8650*|
> > > |SOPA-S|0.6603|0.9456|0.6917|0.8617|0.9812|0.7419|0.7354|0.8666|0.8628|
> > > |AUC-poly|0.6804|0.9543|0.6974|0.8618|0.9835|0.7431|0.7349|0.8676|0.8627|
> > > |AUC-exp|0.6804|0.9493|0.6930|0.8613|0.9827|0.7447|0.7328|0.8672|0.8626|
> > > |CE|0.6420|0.9353|0.6798|0.8467|0.9603|0.7311|0.7223|0.8517|0.8598|
> > > |MB|0.6437|0.9492|0.6913|0.8665|0.9677|**0.7583**|0.7348|0.8651|0.8624|
> > > |AUC-M|0.6520|0.9381|0.6821|0.8505|0.9822|0.7324|0.7361|0.8517|0.8598|
> > > |PAUCI|**0.7192**|**0.9663**|**0.7305**|**0.8814**|**0.9874**|*0.7497*|**0.7618**|**0.8875**|**0.8860**|

---

> > > ### Author Response · Authors · 2022-08-08
> > > **Author response to Reviewer m7Rj (Part 3/5)**
> > >
> > >  **(A2 part 3)**
> > >
> > > >CASE 1: $\Delta \tilde{E} \ge \max\{-a,b-1\}$.
> > >
> > > It is easy to check that when $a = \tilde{E}\_+, b = E\_- $, we have $-a \le \Delta \tilde{E}$ and $b-1 \le \Delta \tilde{E}$. It is easy to see that $a,b$ are decoupled in the expression of $F\_{4,0}(a,b)$. By setting:
> > > $$
> > > \begin{aligned}
> > > \frac{\partial F_{4,0}(a,b)}{\partial a} = 0, \\
> > > \frac{\partial F_{4,0}(a,b)}{\partial b} = 0
> > > \end{aligned}
> > > $$
> > > We know that the minimum solution is attained at $a= \tilde{a}^*$, $b= b^*$. Then the minimum value of $F\_{4,0}(a,b)$ at this range becomes:
> > >
> > > $$
> > > \tilde{E}\_{\tilde{a}^*} + E\_{b^*} + (\Delta \tilde{E})^2
> > > $$
> > >
> > > Moreover, we will also use the fact that **$\tilde{E}\_{\tilde{a}^\*}$ and $E\_{b^\*}$ are also the global minimum for $E\_a$ and $E\_b$, respectively**.
> > >
> > > >CASE 2: $b-1 \ge \Delta \tilde{E},~ -a \le b-1$.
> > >
> > > It is easy to see that $E\_a \ge E\_{\tilde{a}^*}$ in this case.  According to the same derivation as in Lemma A case 2, we have:
> > > $$
> > > E_{-,2} -2b E_- + 2(b-1) \Delta \tilde{E} + 2b -1 \ge E_{b^*} + (\Delta \tilde{E})^2
> > > $$
> > > holds when $b -1 \le \Delta \tilde{E}$.
> > >
> > > Recall that CASE 2 is include in the condition $b -1 \le \Delta \tilde{E}$. So, under the condition of CASE 2:
> > > $$
> > > F\_{4,1}(a,b) \ge \tilde{E}\_{\tilde{a}^*} + E\_{b^*} + (\Delta \tilde{E})^2
> > > $$
> > >
> > > >CASE 3: $-a \ge \Delta \tilde{E},~ b-1 \le -a$.
> > >
> > > In this case, we have $E_b \ge E_{b^*}$. It remains to check:
> > > $$
> > > g(a) =  -2a \tilde{E}_+ -a \Delta \tilde{E}
> > > $$
> > > By taking derivative, we have:
> > >
> > > $$
> > > g'(a) =  -2\tilde{E}\_+ - \Delta \tilde{E} = -\tilde{E}\_+ -E\_+ \le 0.
> > > $$
> > >
> > > Similar as the proof of CASE 2, when $-a \ge \Delta \tilde{E}$, we have:
> > > $$
> > > g(a) \ge \tilde{E}\_{\tilde{a}^*} + (\Delta \tilde{E})^2
> > > $$
> > > and thus
> > > $$
> > > F\_{4,2}(a,b) \ge \tilde{E}\_{\tilde{a}^*} + E\_{b^*} + (\Delta \tilde{E})^2
> > > $$
> > > holds.
> > >
> > > Since the condition of CASE 3 is included in the set $-a \ge \Delta \tilde{E}$:
> > > $$
> > > F\_{4,2}(a,b) \ge \tilde{E}\_{\tilde{a}^*} + E\_{b^*} + (\Delta \tilde{E})^2
> > > $$
> > > holds under the condition of CASE 3.
> > >
> > > > Putting altogether:
> > >
> > > The minimum value of $(OP4)$ reads:
> > > $$
> > > \tilde{E}\_{\tilde{a}^*} + E\_{b^*} + (\Delta \tilde{E})^2 + 2\Delta \tilde{E}
> > > $$
> > > which is the same as $(OP3)$.
> > >
> > > ##### **PROOF COMPLETED**
> > >
> > > Finally, since for each fixed $f$ $(OP3) = (OP4)$, and $(OP1) = (OP2)$ . We can then claim the following theorem:
> > >
> > > > Theorem A (Constrainted Reformulation)
> > > $\min_f (OP1) = \min_f (OP2), ~~ \min_f (OP3) = \min_f (OP4)$
> > >
> > > > **Remark C** Since the calculation is irelevant to the definition of the expectation, the replace the population-level expectation with the empirical expectation over the training data.
> > >
> > > > **Remark D** By applying Theorem 1, we can get the reformulation result in Theorem 2: Note that we have also corrected the problem raised by Reviewer jtnh.
> > > > - for $\mathrm{OPAUC}$
> > >    $$\min\_{f,(a,b)\in[0,1]\^2} \max\_{\gamma\in [b-1,1]}\min\_{s'\in\Omega\_{s'}}\mathbb{E}\_{\boldsymbol{z}\sim \mathcal{D}\_\mathcal{Z}} [G\_{op}(f,a,b,\gamma,\boldsymbol{z},s')]$$
> > >    where
> > >    $$\begin{aligned}
> > >     G\_{op}(f,a,b,\gamma,\boldsymbol{z},s')&=[(f(\boldsymbol{x})-a)\^2-
> > >     2(1+\gamma)f(\boldsymbol{x})]y/p\\\\
> > >     & +
> > >     \left(\beta s' +\left[(f(\boldsymbol{x})-b)\^2+2(1+\gamma) f(\boldsymbol{x})-s'\right]\_+\right)(1-y)/[\beta (1-p)]  -\gamma\^2.
> > >     \end{aligned}
> > >   $$
> > > > - for $\mathrm{TPAUC}$
> > >    $$\min\_{f,(a,b)\in[0,1]\^2 } \max\_{\gamma\in [\max\\{-a,b-1\\}]}\min\_{s\in\Omega\_{s},s'\in\Omega\_{s'}}\mathbb{E}\_{\boldsymbol{z}\sim \mathcal{D}\_\mathcal{Z}} [G\_{tp}(f,a,b,\gamma,\boldsymbol{z},s,s')]$$
> > >       where
> > >    $$\begin{aligned}
> > >     G\_{tp}(f,a,b,\gamma,\boldsymbol{z},s,s')&=\left(\alpha s + r\_{\kappa}\left((f(\boldsymbol{x})-a)\^2-
> > >     2(1+\gamma)f(\boldsymbol{x})-s\right)\right)y/(\alpha p) -\gamma\^2\\\\
> > >     & +
> > >     \left(\beta s' +r\_{\kappa}\left((f(\boldsymbol{x})-b)\^2+2(1+\gamma) f(\boldsymbol{x})-s'\right)\right)(1-y)/[\beta (1-p)].
> > >     \end{aligned}
> > >     $$

---

> > > ### Author Response · Authors · 2022-08-08
> > > **Author response to Reviewer m7Rj (Part 2/5)**
> > >
> > >  **(A2 part 2)**
> > >
> > > Thus, we have:
> > > $$
> > > \min_{b\in[0,1]} \max_{\gamma\in[b-1,1]} F_0=
> > > \min_{b \in [0,1]} F_1
> > > $$
> > > where
> > > $$
> > > F_1 = \begin{cases}
> > > &F_{1,0}(b) := E_b + (\Delta E)^2, & b-1 \le \Delta E \\\\
> > > & F_{1,1}(b) := E_{-,2}- 2bE_- + 2b-1 + 2(b-1)\Delta E, & \text{otherwise}
> > > \end{cases}
> > > $$
> > > It is easy to see that both cases of $F_1$ are convex functions w.r.t $b$. So, we can find the global minimum by comparing the minimum of $F_{1,0}$ and $F_{1,1}$.
> > >
> > > >CASE 1: $\Delta E \ge b-1$.
> > >
> > > It is easy to see that $b_0 = E_- \in (-\infty, 1+\Delta E]$, by taking the derivative to zero, we have, the optimum value is obtained at $b= E_-$ for $F_{1,0}$.
> > >
> > > >CASE 2: $\Delta E \le b-1$.
> > >
> > > Again by taking the derivative, we have:
> > >
> > > $$
> > > F_{1,1}(b)' = -2E_- + 2 + 2 \Delta E = 2-2E_+ \ge 0
> > > $$
> > > We must have:
> > >
> > > $$\inf\_{b > 1 + \Delta E} F\_{1,1}(b) \ge F\_{1,1}(1+\Delta E) = F\_{1,0}(1+\Delta E) > F\_{1,0}(E\_-) =F\_{1,0}(b\^*)$$
> > >
> > > > Putting all together
> > >
> > > Hence the global minimum of $F_1$ is obtained at $b^*$ with:
> > > $$
> > > F_1(b^*) = F_{1,0}(b^*) = E_{b^*}+ (\Delta E)^2
> > > $$
> > >
> > > Hence, we have $(OP2)$ has the minimum value:
> > > $$
> > >  E_{a^*} + E_{b^*} + (\Delta E)^2 + 2 \Delta E
> > > $$
> > >
> > > ### PROOF COMPLETED
> > >
> > > Now, we use a similar trick to prove the result for TPAUC:
> > >
> > > > **Lemma B (The Reformulation for TPAUC)** For a **fixed** scoring function $f$, the following two problems shares the same optimum, given that the scoring function satisfies: $f(\boldsymbol{x}) \in [0,1],~ \forall \boldsymbol{x}$:
> > > $$
> > > \begin{aligned}
> > > &\boldsymbol{(OP3)} \min\_{f,(a,b)\in[0,1]\^2}\max\_{\gamma \in [-1,1]}
> > > \mathbb{E}\_{\boldsymbol{x}\sim\mathcal{D}\_\mathcal{P}}[(f(\boldsymbol{x})-a)\^2|
> > > f(\boldsymbol{x})\leq\eta\_\alpha(f)]+
> > > \mathbb{E}\_{\boldsymbol{x}'\sim\mathcal{D}\_\mathcal{N}}[(f(\boldsymbol{x}')-b)\^2|
> > > f(\boldsymbol{x}')\geq\eta\_\beta(f)]\\\\
> > > +2\Delta \tilde{E} + 2\gamma \Delta \tilde{E}-\gamma\^2
> > > \\\\
> > > &\boldsymbol{(OP4)} \min\_{f,(a,b)\in[0,1]\^2}\max\_{\gamma \in [\max\\{-a,b-1\\},1]}
> > > \mathbb{E}\_{\boldsymbol{x}\sim\mathcal{D}\_\mathcal{P}}[(f(\boldsymbol{x})-a)\^2|
> > > f(\boldsymbol{x})\leq\eta\_\alpha(f)]+
> > > \mathbb{E}\_{\boldsymbol{x}'\sim\mathcal{D}\_\mathcal{N}}[(f(\boldsymbol{x}')-b)\^2|
> > > f(\boldsymbol{x}')\geq\eta\_\beta(f)]\\\\
> > >  +2\Delta \tilde{E} + 2\gamma \Delta \tilde{E}-\gamma\^2
> > > \end{aligned}
> > > $$
> > >
> > > > **Remark B** $\boldsymbol{(OP3)}$ and $\boldsymbol{(OP4)}$ have the equivalent formulation:
> > > $$
> > > \begin{aligned}
> > > \boldsymbol{(OP3)} \Leftrightarrow
> > > \min\_{(a,b)\in[0,1]\^2}\max\_{\gamma \in [-1, 1]} &\mathbb{E}\_{\boldsymbol{z}\sim \mathcal{D}\_\mathcal{Z}}\Big[\left[(f(\boldsymbol{x})-a)\^2-
> > > 2(1+\gamma)f(\boldsymbol{x})\right]y/(\alpha p)\\\\
> > > & +
> > > \left[(f(\boldsymbol{x})-b)\^2+2(1+\gamma) f(\boldsymbol{x})\right]\(1-y\)/[\beta(1-p)] -\gamma\^2\Big].
> > > \end{aligned}
> > > $$
> > > $$
> > > \begin{aligned}
> > > \boldsymbol{(OP4)} \Leftrightarrow
> > > \min\_{(a,b)\in[0,1]\^2}\max\_{\gamma \in [\max\\{-a,b-1\\} 1]} &\mathbb{E}\_{\boldsymbol{z}\sim \mathcal{D}\_\mathcal{Z}}\Big[\left[(f(\boldsymbol{x})-a)\^2-
> > > 2(1+\gamma)f(\boldsymbol{x})\right]y/(\alpha p)\\\\
> > > & +
> > > \left[(f(\boldsymbol{x})-b)\^2+2(1+\gamma) f(\boldsymbol{x})\right]\(1-y\)/[\beta(1-p)] -\gamma\^2\Big].
> > > \end{aligned}
> > > $$
> > >
> > > ##### **PROOF**
> > > Again, $(OP3)$ has the minimum value:
> > >
> > > $$\tilde{E}\_{\tilde{a}\^*} + E _{b ^\*} + \(\Delta \tilde{E}\)\^2 + 2 \\Delta \tilde{E}$$
> > >
> > > We prove that $(OP4)$ ends up with the minimum value.
> > >
> > > By expanding $(OP4)$, we have:
> > > $$
> > > \begin{aligned}
> > > (OP4)= 2\Delta \tilde{E} + \min\_{(a,b)\in[0,1]\^2}\max\_{\gamma \in [\max\\{-a,b-1\\},1]} F\_3
> > > \end{aligned}
> > > $$
> > > where
> > > $F\_3:=  \tilde{E}\_a+ E\_{b} +2\Delta \tilde{E} + 2\gamma \Delta \tilde{E}-\gamma\^2$
> > >
> > >  For any fixed feasible $a,b$, the inner max problem is a truncated quadratic programming, which has a unique and closed-form solution.
> > >
> > > Specifically, define $c = \max\\{-a,b-1\\}$, we have:
> > > $$
> > > (\max_{\gamma \in [c,1]} 2\gamma \Delta \tilde{E}-\gamma^2) =
> > > \begin{cases}
> > > (\Delta \tilde{E})^2,  & \Delta \tilde{E} \ge c \\\\
> > > 2c\Delta \tilde{E} - c^2, & \text{otherwise}
> > > \end{cases}
> > > $$
> > > Thus, we have:
> > > $$
> > > \min_{(a,b)\in[0,1]^2} \max_{\gamma\in[c,1]} F_3=
> > > \min_{(a,b) \in [0,1]} F_4
> > > $$
> > > where
> > > $$
> > > F\_4 = \begin{cases}
> > > &F\_{4,0}(a,b) := \tilde{E}\_a + E\_b + (\Delta \tilde{E})\^2, & c \le \Delta \tilde{E} \\\\
> > > & F\_{4,1}(a,b) := \tilde{E}\_a + E\_{-,2} -2b E\_- + 2(b-1) \Delta \tilde{E} + 2b -1 ,  & b-1 \ge \Delta \tilde{E},~ -a \le b-1 \\\\
> > > & F\_{4,2}(a,b) := E\_b + E\_{+,2}  -2a \tilde{E}\_+ -a \Delta \tilde{E},  & -a \ge \Delta \tilde{E},~ b-1 \le -a
> > > \end{cases}
> > > $$
> > >
> > > It is easy to see that both cases of $F_4$ are convex functions w.r.t $b$. So, we can find the global minimum by comparing the minimum of $F\_{4,0}$,$F\_{4,1}$ and $F\_{4,2}$.

---

> > > ### Author Response · Authors · 2022-08-08
> > > **Author response to Reviewer m7Rj (Part 1/5)**
> > >
> > > Thank you so much for your timely feedback as well as the inspiring questions.
> > >
> > > > **(Q1)** The factor of $p$, $1-p$
> > >
> > >  **(A1)** Thank you so much for your correction. There is a typo in the main paper, for which we are sorry. Actually, in the appendix we have corrected it. We will correct it in the new version.
> > >
> > >
> > > > **(Q2)** The problem of Theorem 1
> > >
> > >  **(A2 part 1)**
> > > We are so grateful for the reviewer's double-check! The reviewer's concern seems to be about the correctness of using the constraint $\gamma \in [b-1,1]$ to replace $\gamma \in [-1,1]$ in the reformulation.
> > > We are sorry for omitting the proof of the details. In fact, we can show that this is correct. Our proof can be established by Lemma A, Lemma B, and Theorem A. We will add it in the appendix in the new version.
> > >
> > > > **Throughout the proof, we will define:**
> > > $$
> > > \begin{aligned}
> > > &a^* = \mathbb{E}\_{\boldsymbol{x}\sim\mathcal{D}\_{\mathcal{P}}}[f(\boldsymbol{x})] &:= E\_+\\\\
> > > &b^* = \mathbb{E}\_{\boldsymbol{x}'\sim\mathcal{D}\_{\mathcal{N}}}[f(\boldsymbol{x}')|f(\boldsymbol{x}')\ge \eta\_\beta(f)] &:=E\_- \\\\
> > > &b^* - a^* &:= \Delta E\\\\
> > > &\tilde{a}^* = \mathbb{E}\_{\boldsymbol{x}\sim\mathcal{D}\_{\mathcal{P}}}[f(\boldsymbol{x})|f(\boldsymbol{x})\le \eta\_\alpha(f)] &:= \tilde{E}\_+\\\\
> > > &b^* - \tilde{a}^* &:= \Delta \tilde{E}\\\\
> > > & \mathbb{E}\_{\boldsymbol{x}\sim\mathcal{D}\_\mathcal{P}}[(f(\boldsymbol{x})-a)^2]& :={E}\_a\\\\
> > > & \mathbb{E}\_{\boldsymbol{x}\sim\mathcal{D}\_\mathcal{P}}[(f(\boldsymbol{x})-a)^2|
> > > f(\boldsymbol{x})\leq\eta\_\alpha(f)]& :=\tilde{E}\_a\\\\
> > > & \mathbb{E}\_{\boldsymbol{x}'\sim\mathcal{D}\_{\mathcal{N}}}[(f(\boldsymbol{x}')-b)^2|f(\boldsymbol{x}')\geq\eta\_{\beta}(f)]  &:= E\_b \\\\
> > > &\mathbb{E}\_{\boldsymbol{x}\sim\mathcal{D}\_\mathcal{P}}[f(\boldsymbol{x})^2|
> > > f(\boldsymbol{x})\leq\eta\_\alpha(f)] &:=E\_{-,2}\\\\
> > > & \mathbb{E}\_{\boldsymbol{x}'\sim\mathcal{D}\_{\mathcal{N}}}[f(\boldsymbol{x}')^2|f(\boldsymbol{x}')\geq\eta\_{\beta}(f)] &:=E\_{+,2}
> > > \end{aligned}
> > > $$
> > >
> > > > **Lemma A (The Reformulation for OPAUC)** For a **fixed** scoring function $f$, the following two problems shares the same optimum, given that the scoring function satisfies: $f(\boldsymbol{x}) \in [0,1],~ \forall \boldsymbol{x}$:
> > > $$
> > > \begin{aligned}
> > > &\boldsymbol{(OP1)} \min\_{(a,b)\in[0,1]^2}\max\_{\gamma \in [-1,1]}
> > > \mathbb{E}\_{\boldsymbol{x}\sim\mathcal{D}\_\mathcal{P}}[(f(\boldsymbol{x})-a)^2] +
> > > \mathbb{E}\_{\boldsymbol{x}'\sim\mathcal{D}\_\mathcal{N}}[(f(\boldsymbol{x}')-b)^2|
> > > f(\boldsymbol{x}')\geq\eta\_\beta(f)] \\\\
> > > +2\Delta E + 2\gamma \Delta E-\gamma^2
> > > \\\\
> > > &\boldsymbol{(OP2)} \min\_{(a,b)\in[0,1]^2}\max\_{\gamma \in [b-1,1]}
> > > \mathbb{E}\_{\boldsymbol{x}\sim\mathcal{D}\_\mathcal{P}}[(f(\boldsymbol{x})-a)^2] +
> > > \mathbb{E}\_{\boldsymbol{x}'\sim\mathcal{D}\_\mathcal{N}}[(f(\boldsymbol{x}')-b)^2|
> > > f(\boldsymbol{x}')\geq\eta\_\beta(f)] \\\\
> > > +2\Delta E + 2\gamma \Delta E-\gamma^2
> > > \end{aligned}
> > > $$
> > >
> > >
> > > > **Remark A** $\boldsymbol{(OP1)}$ and $\boldsymbol{(OP2)}$ have the equivalent formulation:
> > > $$
> > > \begin{aligned}
> > > \boldsymbol{(OP1)}\Leftrightarrow \min\_{(a,b)\in[0,1]\^2}\max\_{\gamma \in [-1,1]} &\mathbb{E}\_{\boldsymbol{z}\sim\mathcal{D}\_{\mathcal{Z}}}\Big[\left[(f(\boldsymbol{x})-a)\^2-
> > > 2(1+\gamma)f(\boldsymbol{x})\right]y/p\\\\
> > > & +
> > > \left[(f(\boldsymbol{x})-b)\^2+2(1+\gamma) f(\boldsymbol{x})\right]\(1-y\)/[\beta(1-p)] -\gamma\^2\Big].
> > > \end{aligned}
> > > $$
> > > $$
> > > \begin{aligned}
> > > \boldsymbol{(OP2)}\Leftrightarrow \min\_{(a,b)\in[0,1]\^2}\max\_{\gamma \in [b-1,1]}& \mathbb{E}\_{\boldsymbol{z}\sim\mathcal{D}\_{\mathcal{Z}}}\Big[\left[(f(\boldsymbol{x})-a)\^2-
> > > 2(1+\gamma)f(\boldsymbol{x})\right]y/p\\\\
> > > & + \left[(f(\boldsymbol{x})-b)\^2+2(1+\gamma) f(\boldsymbol{x})\right]\(1-y\)/[\beta(1-p)] -\gamma\^2\Big].
> > > \end{aligned}
> > > $$
> > >
> > >  **PROOF**
> > > From the proof of our main paper, we know that $(OP1)$ has a closed-form minimum:
> > > $$
> > >  E_{a^*} + {E}_{b^*} + (\Delta E)^2 + 2 \Delta E.
> > > $$
> > >
> > > Hence, we only need to prove that $(OP2)$ has the same minimum solution.
> > >
> > > By expanding $(OP2)$, we have:
> > > $$
> > > \begin{aligned}
> > > \min\_{(a,b)\in[0,1]\^2}\max\_{\gamma\in[b-1,1]}
> > > \mathbb{E}\_{\boldsymbol{z}\sim\mathcal{D}\_{\mathcal{Z}}}[
> > > F\_{op}(f,a,b,\gamma,\eta\_\beta(f),\boldsymbol{z})] &= \\\\
> > > 2\Delta E + \min\_{a \in [0,1]}E\_a  + \min\_{b\in[0,1]}\max\_{\gamma \in [b-1,1]} F\_0
> > > \end{aligned}
> > > $$
> > > where
> > > $$
> > > F_0:= E_b + 2\gamma\Delta E - (\Delta E)^2
> > > $$
> > >
> > > Obviously, since $a$ is decoupled with $b,\gamma$, we have:
> > > $$
> > > \min_{a \in [0,1]} E_a  = E_{a^*}
> > > $$
> > >
> > >
> > > Now, we solve the mini-max problem of $F_0$. For any fixed feasible $b$, the inner max problem is a truncated quadratic programming with a unique and closed-form solution. Hence, we first solve the inner maximization problem for fixed $b$, and then represent the mini-max problem as a minimization problem for $b$.
> > >
> > > Specifically, we have:
> > > $$
> > > (\max_{\gamma\in[b-1,1]} 2\gamma \Delta E-\gamma^2) =
> > > \begin{cases}
> > > (\Delta E)^2,  & \Delta E \ge b-1 \\\\
> > > 2(b-1)\Delta E - (b-1)^2, & \text{otherwise}
> > > \end{cases}
> > > $$

---

> ### Author Response · Authors · 2022-08-01
> **Author response to Reviewer m7Rj (Part 1/2)**
>
> Thanks for your valuable suggestions! The replies to your concerns are attached below.
>
> > **(Q1)** The contribution of this paper
>
> **(A1)**: Thank you so much for your question! The key contribution is two-fold:
>
> 1. We propose a new mini-max reformulation of the OPAUC (TPAUC) optimization problem. With this reformulation, we can find an efficient solution to an approximated problem with time complexity $O(\epsilon^{-3})$ to find an $\epsilon$-stationary point. Moreover, the approximated problem is shown to have asymptotic unbiasedness under certain conditions.
>
> 2. Going beyond efficiency, the reformulation also provides a new path to derive generalization bounds on top of the mini-max problem. With this technique, we derive the upper bounds of the OPAUC (TPAUC) generalization error for **real-valued function classe**s (The following Existing studies can only deal with **0-1 classifier**).
>
>    - Narasimhan, Harikrishna, and Shivani Agarwal. "Support vector algorithms for optimizing the partial area under the ROC curve." Neural computation 29.7 (2017): 1919-1963.
>
>    - Yang, Zhiyong, et al. "When all we need is a piece of the pie: A generic framework for optimizing two-way partial auc." International Conference on Machine Learning. PMLR, 2021.
>
>      Moreover, we can finish the analysis with much simpler steps since the interdependency of pairwise loss terms is eliminated in the instance-wise form.
>
> > **(Q2)** The difference from [39]
>
> **(A2)**:  Thank you for your valuable comments. The difference is four-fold:
>
> 1. We use different techniques from [39] to reformulate the original problem. [39] uses the DC technique to reformulate the objective as a difference of **pairwise** convex functions. In our paper, we reformulate the original problem as a mini-max problem of an **instance-wise** objective function.
>
> 2. From the efficiency perspective, since we are dealing with instance-wise functions, our algorithm has an $O(n_++n_-)$ per-iteration complexity. Moreover, for the convergence rate, our algorithm enjoys an $O(\epsilon^{-3})$ time complexity to achieve an $\epsilon$-stationary point. By contrast, the DC algorithm in [39] has an $O(n_+n_-)$ per-iteration complexity, and an $O(\epsilon^{-6})$ convergence rate.  Hence, our algorithm is more efficient than DC. We agree that the comparison is slightly unfair due to **3** below. Hence, we will use performance comparison to compare them directly.
>
> 3. Our algorithm is solving a smooth approximated problem while [39] is solving a non-smooth version of the original problem. As a result, both algorithms have approximation errors. For our algorithm, the error comes from the bias of the approximation. While for [39], the error comes from the suboptimal definition of the nearly $\epsilon$-critical point. Moreover, we can give an error analysis for the approximation (**please see the response for R1 Q2 for details.**), while such a guarantee is a bit unclear for [39].
>
> 4. Besides optimization, we can also use the reformulated problem to improve generalization analysis.  **Please see the reply to your question Q1 for more details.**
>
> > **(Q3)** This paper solves the approximated problem, while [39] solves the original PAUC problem. The comparison of their convergence rates is not fair.
>
> **(A3)**:  We agree with the reviewer that the convergence rate comparison is a bit unfair since the two methods are solving different problems. However, as shown in reply to your question Q2, since [39] is solving a non-smooth problem, it also induces approximation error. Specifically, it comes from using the nearly $\epsilon$-stationary point instead of the exact $\epsilon$-stationary point. So we will compare them directly according to their performance. See the reply to the question below.
>
> > **(Q4)** It would be more convincing to see some numerical result comparison with [39]
>
> **(A4):**
>  Since [39] is designed for OPAUC, we conduct the comparison on all datasets in terms of OPAUC. The results show that our algorithm can achieve better performance.
> OPAUC ($\mathrm{FPR}\leq0.3$)
>
> |       | cifar-10-1 | cifar-10-2 | cifar-10-3 | cifar-100-1 | cifar-100-2 | cifar-100-3 | tiny-im-1 | tiny-im-2 | tiny-im-3 |
> |:-----:|:----------:|:----------:|:----------:|:-----------:|:-----------:|:-----------:| :---------:|:---------:|:---------:|
> | PAUCI | 0.7713     | 0.9718     | 0.7736     | 0.9199      | 0.9885      | 0.8538      | 0.8223    | 0.9124    | 0.9141    |
> | DCA   | 0.7526     | 0.9615     | 0.7497     | 0.9105      | 0.9814      | 0.8406      |0.8135    | 0.9081    | 0.9057    |

---

### Official Review · Reviewer_jtnh · 2022-07-11

**Rating:** 6
**Confidence:** 2
**Soundness:** 3 good
**Presentation:** 2 fair
**Contribution:** 3 good

**Summary:**

In this paper, they introduce a new method to optimize OPAUC and TPAUC. After presenting its derivation, they provide an implementation with SGD and analyze its convergence rate and generalization. Finally, they provide experiments comparing their new method to pre-existing.

**Questions:**

1. In the Notation section, is $\mathcal{D}_Z$ understood to be a mixture of $\mathcal{D}_P \times \{1\}$ and $\mathcal{D}_N \times \{0\}$? If so, at what mixture proportion?
2. In line 100, it is said that ``Note that (3) and (5) are hard to optimize since it is complicated to determine 100 the positive quantile $\eta_\alpha(f)$ and the negative quantile $\eta_\beta(f)$.'' This may be a silly question, but can't you just sort to find the largest $n^\alpha_+$ and $n^\beta_-$ scores? This would be like $O(n\log n)$ time, which wouldn't be much worse than the $O(n)$ time.
3. Can some intuition be provided behind $F_{op}$; i.e., what roles do all the introduced variables play? Does the equivalence of (9) hold for each $f$? That is, is it true that $\mathcal{R}\_\beta(f) = \min\_{a,b} \max\_{\gamma} \mathbb{E}\_z[F\_{op}(\dots)]$?
4. In Line 115: Does $\hat{\mathbb{E}}_{x' \sim \mathcal{D}\_\mathcal{N}}$ mean empirical expectation over negative data points? If so, this seems inconsistent with the definition of $\eta_\beta(f)$ from before.
5. Again in Line 115: Isn't $\mathbb{I}[f(x') \geq \eta_\beta]$ always 0 or 1? How can it equal $\beta$?
6. In the equation block in line 116: should it be $z_i$ instead of $z$ in the sum?
7. The claim of asymptotic unbiasedness seems to come from the pointwise convergence of the objective $G_{op}^\kappa$ as $\kappa \rightarrow \infty$. However, I'm a bit worried that pointwise convergence of the objective may not necessarily entail convergence of the minimizer. It is especially worrying that $\omega$ grows with $\kappa$ in the training algorithm, so the regularization effect may dominate or cause some asymptotic bias. Can the claim of asymptotic unbiasedness be more fleshed out?
8. As $G$ is a proxy function, are there conditions on the original problem that would make $G$ automatically satisfy Assumption 1?

**Limitations:**

Main limitations are discussed in the questions section (in particular, questions 3 and 7).

**Strengths And Weaknesses:**

Strengths:
* Cool reformulation ideas
* Decent experiments showing the strengths of their method

Weaknesses:
* Unclear or sloppy notation (see questions 1, 4, 5, 6)

---

> ### Author Response · Authors · 2022-08-01
> **Author response to Reviewer jtnh (Part 4/4)**
>
> **(A7 Part 2)**:
>
> **(2) Convergence with the Regularization Term**
>
> The regularization term is introduced to ensure that the objective is concave $\textit{w.r.t.}$ $\gamma$ when all the other variables are fixed. This is true when $\kappa\leq 2+ 2\omega$ (Recall that $\kappa$ is the hyperparameter of $r_\kappa$, and $\omega$ is the regularization weight in the loss function). In practice, we do not need a large $\kappa$ to achieve unbiasedness. So the required $\omega$ also has a limited magnitude.  Next, we show this claim empirically.
>
>  Since $g \in [-5,5]$, we can check the uniform approximation ability of a given $\kappa$ with the following error:
>
> $$
> err(\kappa) =
> \int_{-5}^{5}\left(
> \frac{\log (1+\exp (\kappa \cdot g))}{\kappa}-[g]_+
> \right)^2dg.
> $$
>
> Recall that  $\kappa\leq 2+ 2\omega, 0 \le \omega$, given a choice of $\kappa$, we can choose  $\omega$ as:
>
> $$
> \omega=[\kappa/2-1]_+.
> $$
>
> The following table shows how $err(\kappa), \omega$ evolves when $\kappa$ grows from 1 to 12.
>
> | $\kappa$      | 1        | 2        | 3        | 4        | 5        | 6        |
> |:-------------|:-------- |:-------- | -------- |:-------- |:-------- |:-------- |
> | $\omega$      | 0        | 0        | 0.5      | 1        | 1.5      | 2        |
> | $err(\kappa)$ | 0.600983 | 0.075128 | 0.022260 | 0.009391 | 0.004808 | 0.002782 |
>
> | $\kappa$      | 7        | 8        | 9        | 10       | 11       | 12       |
> |:-------------|:-------- |:-------- |:-------- |:-------- |:-------- |:-------- |
> | $\omega$      | 2.5      | 3        | 3.5      | 4        | 4.5      | 5        |
> | $err(\kappa)$ | 0.001752 | 0.001173 | 0.000824 | 0.000601 | 0.000451 | 0.000347 |
>
> It is easy to see that having a $\omega$ around 6 is sufficient to get a  good approximation. Hence, the regularization term will not dominate the loss in most cases.
>
> > **(Q8)**  As $G$ is a proxy function, are there conditions on the original problem that would make $G$ automatically satisfy Assumption 1?
>
> **(A8)**: Thank you so much for your question! First, this is a widely-used assumption for non-convex optimization literature such as Ref. [1-9].
>
> Moreover, some recent studies also verify the feasibility of the gradient Lipschitz assumption of deep learning models. For example, Ref.[9] states that the gradient Lipschitz assumption is satisfied if the following conditions hold:
>
> 1. The neural network uses the smooth activation function ($\textit{i.e.}$, Linear function, Logistic function, Softplus function, Tanh function).
>
> 2. The output score is bounded with Sigmoid function $\sigma(x)=1/(1+\exp(-f(\boldsymbol{x}))$ (the input will be scale into closed set [0,1]).
>
> 3. The parameter of neural network $\boldsymbol{w}$ is bounded ($\textit{e.g.}$, $\|\boldsymbol{w}\|^2\leq M$).
>
> Ref. [10] estimates the gradient Lipschitz constants of the deep neural network. They propose a general estimation for the upper bound on the Lipschitz constant of the gradient of any loss function for the parameters. The results show that such constants are finite under mild conditions.
>
> ###### REFERENCE
>
> 1. Guo, Zhishuai, et al. "Randomized stochastic variance-reduced methods for multi-task stochastic bilevel optimization." *arXiv preprint arXiv:2105.02266* (2021). **[Assumption 1, 2]**
>
> 2. Hu, Quanqi, Yongjian Zhong, and Tianbao Yang. "Multi-block Min-max Bilevel Optimization with Applications in Multi-task Deep AUC Maximization." *arXiv preprint arXiv:2206.00260* (2022). **[Assumption 2.2]**
>
> 3. Jiang, Wei, et al. "Multi-block-Single-probe Variance Reduced Estimator for Coupled Compositional Optimization." *arXiv preprint arXiv:2207.08540* (2022). **[Assumption 1]**
>
> 4. Qi, Qi, et al. "Attentional biased stochastic gradient for imbalanced classification." *arXiv preprint arXiv:2012.06951* (2020). **[Assumption 1]**
>
> 5. Wang, Bokun, and Tianbao Yang. "Finite-Sum Coupled Compositional Stochastic Optimization: Theory and Applications." *International Conference on Machine Learning*. PMLR, 2022. **[Assumption 1]**
>
> 6. Luo, Luo, et al. "Stochastic recursive gradient descent ascent for stochastic nonconvex-strongly-concave minimax problems." *Advances in Neural Information Processing Systems* 33 (2020): 20566-20577. **[Assumption 2]**
>
> 7. Chen, Tianyi, Yuejiao Sun, and Wotao Yin. "A single-timescale stochastic bilevel optimization method." *arXiv preprint arXiv:2102.04671* (2021). **[Assumption 1]**
>
> 8. Gao, Hongchang, Bin Gu, and My T. Thai. "Stochastic Bilevel Distributed Optimization over a Network." *arXiv preprint arXiv:2206.15025* (2022).**[Assumption 4]**
>
> 9. Dixian Zhu, Gang Li, Bokun Wang, Xiaodong Wu, Tianbao Yang *Proceedings of the 39th International Conference on Machine Learning*, PMLR 162:27548-27573, 2022. **[Assumption 2]**
>
> 10. Fazlyab, Mahyar, et al. "Efficient and accurate estimation of lipschitz constants for deep neural networks." Advances in Neural Information Processing Systems 32 (2019).

---

> > ### Comment · Reviewer_jtnh · 2022-08-05
> > **Response to authors**
> >
> > Thank you for all the responses! I have a couple remaining questions/concerns:
> >
> > **Regarding A1**
> > Taking a look at Appendix E, I see that there is $p=\mathbb{P}(y=1)$, but this doesn't seem defined when $\mathcal{D}\_\mathcal{Z}$ is; furthermore, this parameter $p$ exists in the Thm 1 statement in the appendix, but not in the main text. Am I supposed to implicitly take it to equal 1/2?
> >
> > **Regarding A3**
> > Having looked at Appendix E, I think there may be some factors missing in $F\_{op}$?
> >
> > From Eq. 57,
> > $$F\_{op} = (f(x)-a)^2y/p + (f(x)-b)^2(1-y)/(1-p)I\\{f(x) \geq \eta\_\beta(f)\\} + 2(1+\gamma)f(x)(1-y)/(1-p)I\\{f(x) \geq \eta\_\beta(f)\\} - 2(1+\gamma)f(x)y/p - \gamma^2$$
> >
> > Taking expectation wrt $z$, we get
> > $$\mathbb{E}\_{x \sim D\_P}[(f(x)-a)^2] + \beta \mathbb{E}\_{x \sim D\_N}[(f(x)-b)^2] + 2\beta(1+\gamma)\mathbb{E}\_{x \sim D\_N}[f(x)] + 2(1+\gamma)\mathbb{E}\_{x \sim D\_P}[f(x)] - \gamma^2$$
> >
> > However, based on the preceding discussion, we should also have factors of $\beta$ on the first, fourth, and fifth terms. Am I missing something somewhere? Also, it's not clear what setting of $p$ recovers the statement of Theorem 1 in the main text; the most likely candidate would be $p=1/2$ (and then dividing the objective by half), but that would yield an extra factor of $2$ on the $\gamma^2$ term.
> >
> > **Regarding A7**
> > Based on this response, is it correct to say that with regularization, it isn't necessarily known whether or not you still have asymptotic unbiasedness?

---

> > > ### Author Response · Authors · 2022-08-08
> > > **Author response to Reviewer jtnh**
> > >
> > > Thank you so much for your careful and valuable feedbacks! The responses are as follows:
> > >
> > > > **(Q1)**: Taking a look at Appendix E, I see that there is $p=\mathbb{P}(y=1)$, but this doesn't seem defined when $\mathcal{D}_{\mathcal{Z}}$ is; furthermore, this parameter $p$ exists in the Thm.1 statement in the appendix, but not in the main text. Am I supposed to implicitly take it to equal 1/2?
> > >
> > >  **(A1)**: We are sorry for the notation typo. $p$ should exist in Thm.1 in the main text.
> > >
> > > > **(Q2)**: Having looked at Appendix E, I think there may be some factors missing in $F_{op}$?
> > >
> > >  **(A2)**: Thank you for your correction. There are indeed some factors missing in $F_{op}$. Actually, $1/\beta$ should appear for all the terms involving the conditional expectation. We will fix this issue in the next version. The correct definition for $\mathrm{OPAUC}$ should be:
> > >
> > > $
> > > \min\_{(a,b)\in[0,1]\^2}\max\_{\gamma\in[-1,1]} \underset{\boldsymbol{z}\sim\mathcal{D}\_{\mathcal{Z}}}{\mathbb{E}} [F\_{op}(f,a,b,\gamma,\eta\_{\beta}(f),\boldsymbol{z})],
> > > $
> > >
> > > where
> > >
> > > $
> > > \begin{aligned}
> > > F\_{op}(f,a,b,\gamma,\eta\_{\beta}(f),\boldsymbol{z})&=(f(\boldsymbol{x})-a)^2y/p+
> > > (f(\boldsymbol{x})-b)^2(1-y)\mathbb{I}\_{f(\boldsymbol{x})\geq\eta\_{\beta}(f)}/[\beta(1-p)]\\\\
> > > &\quad+2(1+\gamma)f(\boldsymbol{x})(1-y)\mathbb{I}\_{f(\boldsymbol{x})\geq\eta\_{\beta}(f)}/[\beta(1-p)] -
> > > 2(1+\gamma)f(\boldsymbol{x})y/p -\gamma^2.
> > > \end{aligned}
> > > $
> > >
> > > This is the right expression which equals to
> > >
> > > $
> > > \begin{aligned}
> > > &\min\_{(a,b)\in[0,1]^2}\max\_{\gamma\in[-1,1]} \underset{\boldsymbol{x}\sim\mathcal{D}\_{\mathcal{P}}}{\mathbb{E}} [(f(\boldsymbol{x})-a)^2-2(\gamma+1)f(\boldsymbol{x})]  -\gamma^2 \\\\
> > > &+ \underset{\boldsymbol{x}'\sim\mathcal{D}\_{\mathcal{N}}}{\mathbb{E}} [(f(\boldsymbol{x}')-b)^2+2(\gamma+1)f(\boldsymbol{x}')|f(\boldsymbol{x}')\geq \eta\_{\beta}(f)].
> > > \end{aligned}
> > > $
> > >
> > > We also correct the expressions in Thm.1-3, including TPAUC.
> > >
> > > To correct the experiments, we conduct the experiments with consideration of factors missing $\beta$. Here is the result (The highest and the second-best results are bolded and italic, respectively), which shows that our algorithm still outperforms the competitors for the following OPAUC and TPAUC metrics.
> > >
> > > > **OPAUC ($\mathrm{FPR}\leq 0.3$)**
> > >
> > > |methods|cifar-10-1|cifar-10-2|cifar-10-3|cifar-100-1|cifar-100-2|cifar-100-3|tiny-im200-1|tiny-im200-2|tiny-200-3|
> > > |--------|----------|----------|----------|-----------|-----------|-----------|------------|------------|----------|
> > > |SOPA|*0.7659*|*0.9688*|**0.7651**|*0.9108*|*0.9875*|0.8483|0.8157|0.9037|0.9066|
> > > |SOPA-S|0.7548|0.9674|*0.7542*|0.9033|0.9860|0.8449|0.8180|*0.9087*|0.9095|
> > > |AGD-SBCD|0.7526|0.9615|0.7497|0.9105|0.9814|0.8406|0.8135|0.9081|0.9057|
> > > |AUC-poly|0.7542|0.9672|0.7538|0.9027|0.9859|0.8441|0.8185|0.9084|*0.9100*|
> > > |AUC-exp|0.7347|0.9620|0.7457|0.8987|0.9850|0.8407|0.8127|0.9026|0.9049|
> > > |CE|0.7417|0.9431|0.7428|0.8903|0.9695|0.8321|0.8023|0.8917|0.8878|
> > > |MB|0.7492|0.9648|0.7500|0.9003|0.9804|**0.8575**|*0.8193*|0.9072|0.9091|
> > > |AUC-M|0.7334|0.9609|0.7442|0.8996|0.9845|0.8403|0.8102|0.9011|0.9043|
> > > |PAUCI|**0.7721**|**0.9716**|0.7399|**0.9155**|**0.9889**|*0.8492*|**0.8267**|**0.9214**|**0.9217**|
> > >
> > > >**TPAUC ($\mathrm{FPR}\leq 0.5, \mathrm{TPR}\geq 0.5$)**
> > >
> > > |methods|cifar-10-1|cifar-10-2|cifar-10-3|cifar-100-1|cifar-100-2|cifar-100-3|tniy-im200-1|tiny-im200-2|tiny-200-3|
> > > |--------|----------|----------|----------|-----------|-----------|-----------|------------|------------|----------|
> > > |SOPA|*0.7096*|*0.9593*|*0.7220*|*0.8714*|*0.9855*|0.7485|*0.7417*|*0.8681*|*0.8650*|
> > > |SOPA-S|0.6603|0.9456|0.6917|0.8617|0.9812|0.7419|0.7354|0.8666|0.8628|
> > > |AUC-poly|0.6804|0.9543|0.6974|0.8618|0.9835|0.7431|0.7349|0.8676|0.8627|
> > > |AUC-exp|0.6804|0.949|0.6930|0.8613|0.9827|0.7447|0.7328|0.8672|0.8626|
> > > |CE|0.6420|0.9353|0.6798|0.8467|0.9603|0.7311|0.7223|0.8517|0.8598|
> > > |MB|0.6437|0.9492|0.6913|0.8665|0.9677|**0.7583**|0.7348|0.8651|0.8624|
> > > |AUC-M|0.6520|0.9381|0.6821|0.8505|0.9822|0.7324|0.7361|0.8517|0.8598|
> > > |PAUCI|**0.7192**|**0.9663**|**0.7305**|**0.8814**|**0.9874**|*0.7497*|**0.7618**|**0.8875**|**0.8860**|
> > >
> > > > **(Q3)**: Based on this response, is it correct to say that with regularization, it isn't necessarily known whether or not you still have asymptotic unbiasedness?
> > >
> > >  **(A3)**: Thank you for your question. The answer is Yes. The regularization might lead to extra bias. However, a regularization scheme is a popular trick for machine learning methods. As a very general result, it will inevitably induce bias. However, it is known to be a necessary building block to stabilize the solutions and improve generalization performance.

---

> > > > ### Comment · Reviewer_jtnh · 2022-08-08
> > > > **Response to authors**
> > > >
> > > > Thanks for the response and all the clarifications!
> > > >
> > > > I will increase my score due to the corrections and clarifications made.

---

> ### Author Response · Authors · 2022-08-01
> **Author response to Reviewer jtnh (Part 3/4)**
>
> > **(Q4)** In Line 115: Does $\hat{\mathbb{E}} _{\boldsymbol{x}'\sim\mathcal{D} _{\mathcal{N}}}$ mean empirical expectation over negative data points? If so, this seems inconsistent with the definition of $\eta _{\beta}(f)$ from before.
>
> **(A4)**: We are sorry for this typo. It should be $\mathbb{E} _{\boldsymbol{x}' \sim \mathcal{D} _{\mathcal{N}}}$ in line 115.
>
> > **(Q5)** Again in Line 115: Isn't $\mathbb{I}[f(x'\geq\eta _{\beta})]$ always 0 or 1? How can it equal $\beta$?
>
> **(A5)**: We are sorry for this typo. It should be
>
> $$
> \eta _{\beta}(f)=\arg \min _{\eta _{\beta} \in \mathbb{R}} \Big[ \mathbb{E} _{\boldsymbol{x}^{\prime} \sim \mathcal{D}
>  _{\mathcal{N}}}\Big[\mathbb{I}\left[f\left(\boldsymbol{x}^{\prime}\right) \geq \eta _{\beta}\right]\Big]=\beta\Big].
> $$
>
> > **(Q6)**  In the equation block in the line 116: should it be $z_i$ instead of $z$ in the sum?
>
> **(A6)**: Yes, it should be replaced with $z_i$ here. Thank you so much for your suggestions!
>
> > **(Q7)**  The claim of asymptotic unbiasedness seems to come from the point-wise convergence of the objective $G_{op}^{\kappa}$ as $\kappa \to \infty$. However, I'm a bit worried that point-wise convergence of the objective may not necessarily convergence of minimizer. It is especially worrying that $\omega$ grows with $\kappa$ in the training algorithm, so the regularization effect may dominate or cause some asymptotic bias. Can the claim of asymptotic unbiasedness be more fleshed out?
>
> **(A7 Part 1)**: Thank you so much for such an inspiring question! We will answer it with two sub-problems. We will attach the following arguments to the appendix in the next version.
>
> ### （1）Uniform Convergence without regularization.
>
> Thank you so much for posing this question. Note that the parameters $a,b,\gamma,s'$ are chosen from tight feasible sets and that the scoring function outputs are assumed to be located in $[0,1]$. In this sense, we can also prove that a stronger convergence result holds without the regularization term:
>
> $$
> \begin{aligned}
> &\lim _{\kappa \rightarrow \infty}
> \bigg|
> \min _{f,(a, b)\in[0,1] ^2}
> \max _{\gamma\in\Omega _{\gamma}}
> \min _{s ^{\prime}\in\Omega _{s'}}
> \hat{\mathbb{E}} _{\boldsymbol{z} _{i} \sim S}\Big[
> G _{o p}^{\kappa}\left(f, a, b, \gamma, \boldsymbol{z} _{i}, s ^{\prime}\right)
> \Big]\\\\
> &~~~~~~~~~~~~-
> \min _{ f,(a, b)\in[0,1] ^2}
> \max _{\gamma\in\Omega _{\gamma}}
> \min _{s ^{\prime}\in\Omega _{s'}}
> \hat{\mathbb{E}} _{\boldsymbol{z} _{i} \sim S}\Big[
> G _{o p}(f,a, b, \gamma, \boldsymbol{z} _{i}, s ^{\prime})\Big]
> \bigg| \\\\
> &=0. ~~~~~~~~~~~~~~~~~~~~~~~~~~~~~~~~~~~~~~~~~~~~~~~~~~~~~~~~~~~~~~~~~~~~~~~~~~~~~~~~~~~~~~~~~~~~~~~~~~~~~~~~~~~~~~~~~(*)
> \end{aligned}
> $$
>
> We prove Eq.$(*)$  in the following arguments:
>
> ### PROOF:
>
> Denote:
>
> $$
> \begin{aligned}
> \Delta=&\bigg|
> \min _{f,(a, b)\in[0,1] ^2}
> \max _{\gamma\in\Omega _{\gamma}}
> \min _{s ^{\prime}\in\Omega _{s'}}
> \hat{\mathbb{E}} _{\boldsymbol{z} _{i} \sim S}\Big[
> G _{o p} ^{\kappa}\left(f, a, b, \gamma, \boldsymbol{z} _{i}, s ^{\prime}\right)
> \Big]\\\\
> &-
> \min _{ f,(a, b)\in[0,1] ^2}
> \max _{\gamma\in\Omega _{\gamma}}
> \min _{s ^{\prime}\in\Omega _{s'}}
> \hat{\mathbb{E}} _{\boldsymbol{z} _{i} \sim S}\Big[
> G _{o p}(f,a, b, \gamma, \boldsymbol{z} _{i}, s^{\prime})\Big]
> \bigg|.
> \end{aligned}
> $$
>
> First, we have:
>
> $$
> \begin{aligned}
> & \limsup _{\kappa \rightarrow +\infty}\Delta \leq & \underbrace{
> \limsup _{\kappa \rightarrow +\infty}
> \sup _{f, (a, b)\in[0,1]^2, \gamma\in \Omega _{\gamma}, s^{\prime}\in \Omega _{s'},
> \boldsymbol{z}\sim\mathcal{D} _{\mathcal{Z}}}
> \left|
> \frac{\log(1+\exp(\kappa\cdot g))}{\kappa} -
> [g] _+
> \right|} _{(a)}.
> \end{aligned}
> $$
>
> where $g=(f(\boldsymbol{x})-b)^2+2(1+\gamma)f(\boldsymbol{x})-s'$ and $[x]_+=\max\\{x,0\\}$. Since $g \in[-5,5]$ in the feasible set, we have:
>
> $$
> (a)\le \limsup _{\kappa \rightarrow +\infty}
> \sup _{x\in[-5,5]}
> \left|
> \frac{\log(1+\exp(\kappa\cdot x))}{\kappa} -
> [x] _+
> \right|.
> $$
>
> Next, we prove that
>
> $$
> \underset{\kappa\to\infty}{\limsup}
> \underset{x \in [-5,5]}{\sup}
> \left[\left|
> \frac{\log(1+\exp(\kappa\cdot x))}{\kappa} -
> [x]_+
> \right|\right] \le 0.
> $$
>
> For the sake of simplicity, we denote:
>
> $$
> \ell(x) = \left|
> \frac{\log(1+\exp(\kappa\cdot x))}{\kappa} -
> [x]_+
> \right|.
> $$
>
> When $x<0$, we have:
>
> $$
> \ell(x)^\prime = \left(\frac{\log(1+\exp(\kappa \cdot x))}{\kappa}\right)^\prime \ge 0.
> $$
>
> When $x>0$, we have:
>
> $$
> \ell(x)'=\left(\frac{\log(1+\exp(\kappa \cdot x))}{\kappa}-x\right)^\prime \le 0.
> $$
>
> Hence, the supermum must be attained at $x=0$. We have:
>
> $$
> (a)\le \limsup_{\kappa\rightarrow +\infty} \frac{\log(2)}{\kappa}= 0 .
> $$
>
> The absolute value ensures that:
>
> $$
> \liminf_{\kappa\rightarrow +\infty}\Delta\ge 0.
> $$
>
> The result follows from the fact:
>
> $$
> 0 \le \liminf_{\kappa \rightarrow +\infty} \Delta \le \limsup_{\kappa \rightarrow +\infty} \Delta \le 0.
> $$
>
> ### PROOF COMPLETED
>
> Moreover, from the proof above, we also obtain a  convergence rate:
>
> $$
> \Delta = O(1/\kappa).
> $$

---

> ### Author Response · Authors · 2022-08-01
> **Author response to Reviewer jtnh (Part 2/4)**
>
> **(A3 Part 2)**:
>
> For the original objective function, we have:
>
> $$
> \begin{aligned}
> &\underset{\boldsymbol{x}, \boldsymbol{x} ^{\prime} \sim \mathcal{D} _{\mathcal{P}}, \mathcal{D} _{\mathcal{N}}}{\mathbb{E}}\left[\left(1-\left(f(\boldsymbol{x})-f\left(\boldsymbol{x} ^{\prime}\right)\right)\right) ^{2} \mid f\left(\boldsymbol{x} ^{\prime}\right) \geq \eta _{\beta}(f)\right]\\\\
> =1 &+
> \underbrace{
>     \mathbb{E} _{\boldsymbol{x} \sim \mathcal{D} _{\mathcal{P}}}
>     \left[f(\boldsymbol{x}) ^{2}\right]
>     -
>     \big(\mathbb{E} _{\boldsymbol{x} \sim \mathcal{D} _{\mathcal{P}}}
>     [f(\boldsymbol{x})]\big) ^{2}
> } _{(1)}\\\\
> &+
> \underbrace{
>     \mathbb{E} _{\boldsymbol{x} ^{\prime} \sim \mathcal{D} _{\mathcal{N}}}
>     \left[f\left(\boldsymbol{x} ^{\prime}\right)^{2}
>     \mid f\left(\boldsymbol{x} ^{\prime}\right) \geq \eta _{\beta}(f)\right]
>     -
>     \bigg(
>     \mathbb{E} _{\boldsymbol{x}^{\prime} \sim \mathcal{D} _{\mathcal{N}}}
>     \Big[f\left(\boldsymbol{x} ^{\prime}\right)
>     \mid f\left(\boldsymbol{x} ^{\prime}\right) \geq \eta _{\beta}(f)\Big]
>     \bigg)^{2}\\
> } _{(2)}\\\\
> & +
> \underbrace{
> \bigg(
> \mathbb{E} _{\boldsymbol{x} \sim \mathcal{D} _{\mathcal{P}}}
> \Big[f(\boldsymbol{x})\Big]
> -
> \mathbb{E} _{\boldsymbol{x}^{\prime} \sim \mathcal{D} _{\mathcal{N}}}
> \Big[f\left(\boldsymbol{x} ^{\prime}\right)
> \mid f\left(\boldsymbol{x} ^{\prime}\right) \geq \eta _{\beta}(f)\Big]
> \bigg)^{2}
> -2 \mathbb{E} _{\boldsymbol{x} \sim \mathcal{D} _{\mathcal{P}}}
> \Big[f(\boldsymbol{x})\Big]
> +2 \mathbb{E} _{\boldsymbol{x} ^{\prime} \sim \mathcal{D} _{\mathcal{N}}}
> \Big[f\left(\boldsymbol{x} ^{\prime}\right)
> \mid f\left(\boldsymbol{x} ^{\prime}\right) \geq \eta _{\beta}(f)\Big].
> } _{(3)}
> \end{aligned}
> $$
>
> It is easy to see that the pairwise form comes from $(1),(2),(3)$.
>
> For $(1)$, according to the definition of the variance, we have:
>
> $$
> {\underset{\boldsymbol{x} \sim
> \mathcal{D} _{\mathcal{P}}}{\mathbb{E}}
> \left[f(\boldsymbol{x})^{2}\right]-
> \underset{\boldsymbol{x}
> \sim \mathcal{D} _{\mathcal{P}}}{\mathbb{E}}
> [f(\boldsymbol{x})]^{2}}=\min _{a \in[0,1]}
> \underset{\boldsymbol{x} \sim
> \mathcal{D} _{\mathcal{P}}}{\mathbb{E}}
> \left[(f(\boldsymbol{x})-a)^{2}\right],
> $$
>
> with the optimal $a$ being:
>
> $$
> a^{*}=\underset{\boldsymbol{x}
> \sim \mathcal{D}_{\mathcal{P}}}{\mathbb{E}}[f(\boldsymbol{x})].
> $$
>
> Similarly, for $(2)$, we have:
>
> $$
> \begin{array}{c}
> {\underset{\boldsymbol{x}^{\prime}
> \sim
> \mathcal{D} _{\mathcal{N}}}{\mathbb{E}}
> \left[f\left(\boldsymbol{x}^{\prime}\right)^{2}
> \mid
> f\left(\boldsymbol{x}^{\prime}\right) \geq \eta _{\beta}(f)\right]-
> \underset{\boldsymbol{x}^{\prime} \sim \mathcal{D} _{\mathcal{N}}}
> {\mathbb{E}}\Big[f\left(\boldsymbol{x}^{\prime}\right) \mid
> f\left(\boldsymbol{x}^{\prime}\right) \geq \eta _{\beta}(f)\Big]^{2}= }\\
> \min _{b \in[0,1]} \underset{\boldsymbol{x}^{\prime} \sim \mathcal{D} _{\mathcal{N}}}{\mathbb{E}}\left[\left(f\left(\boldsymbol{x}^{\prime}\right)-b\right)^{2} \mid f\left(\boldsymbol{x}^{\prime}\right) \geq \eta _{\beta}(f)\right],
> \end{array}
> $$
>
> with the optimal $b$ being:
>
> $$
> b^{*}=\underset{\boldsymbol{x}^{\prime} \sim \mathcal{D} _{\mathcal{N}}}{\mathbb{E}}\left[f\left(\boldsymbol{x}^{\prime}\right) \mid f\left(\boldsymbol{x}^{\prime}\right) \geq \eta _{\beta}(f)\right] .
> $$
>
> For (3), we have:
>
> $$
> \begin{array}{l}
> {\left(\underset{\boldsymbol{x}^{\prime} \sim \mathcal{D}\_{\mathcal{N}}}{\mathbb{E}}\left[f\left(\boldsymbol{x}^{\prime}\right) \mid f\left(\boldsymbol{x}^{\prime}\right) \geq \eta\_{\beta}(f)\right]-\underset{\boldsymbol{x} \sim \mathcal{D}\_{\mathcal{P}}}{\mathbb{E}}[f(\boldsymbol{x})]\right)^{2} }=\\\\
> \max _{\gamma}\left\\{2 \gamma\left(\underset{\boldsymbol{x}^{\prime} \sim \mathcal{D} _{\mathcal{N}}}{\mathbb{E}}\left[f\left(\boldsymbol{x}^{\prime}\right) \mid f\left(\boldsymbol{x}^{\prime}\right) \geq \eta _{\beta}(f)\right]-\underset{\boldsymbol{x} \sim \mathcal{D} _{\mathcal{P}}}{\mathbb{E}}[f(\boldsymbol{x})]\right)-\gamma^{2}\right\\},
> \end{array}
> $$
>
> with the optimal solution being:
>
> $$
> \gamma^{*}=\underset{\boldsymbol{x}^{\prime} \sim \mathcal{D} _{\mathcal{N}}}{\mathbb{E}}\left[f\left(\boldsymbol{x}^{\prime}\right) \mid f\left(\boldsymbol{x}^{\prime}\right) \geq \eta _{\beta}(f)\right]-\underset{\boldsymbol{x} \sim \mathcal{D} _{\mathcal{P}}}{\mathbb{E}}[f(\boldsymbol{x})].
> $$
>
> So far, we can plug in the intermediate results to get an instance-wise mini-max reformulation. For more details, please refer to **Appendix E.1.1** to see the proof of Thm.1.
>
> Based on the reformulation, we can use the monotonicity constraint and the differentiable reformulation of the top-k-sum problem to introduce the variable $s^{\prime}$ to deal with the constraint $f(\boldsymbol{x}') \ge \eta_\beta(f)$. Please see the proof of Prop.1 and Thm.2 in **Appendix E.1.2** for more details.
>
> In the reformulation, $f$ is  always  fixed. So, we will have
>
> $$
> \mathcal{R} _\beta(f)=\min _{a,b}\max_\gamma\mathbb{E} _z[F _{op}(...)].
> $$

---

> ### Author Response · Authors · 2022-08-01
> **Author response to Reviewer jtnh (Part 1/4)**
>
> Thank you for your valuable comments. The replies are attached below.
>
> > **(Q1)** In the Notation section, is $\mathcal{D} _{\mathcal{Z}}$ understood to be a mixture of $\mathcal{D} _{\mathcal{P}}\times 1$ and $\mathcal{D} _{\mathcal{N}}\times 0$? If so, at what mixture proportion?
>
> **(A1):** We are sorry for the unclear notation. The definition of $\mathcal{D} _{\mathcal{Z}}$ should be:
>
> $$
> \begin{array}{lll}
> \underset{z}{\mathbb{P}}(\cdot)= & \mathbb{P} _{x \mid y=1}(\cdot) \cdot \mathbb{P}(y=1)  +&\mathbb{P} _{x \mid y=0}(\cdot) \cdot \mathbb{P}(y=0). \\\\
> \downarrow&\downarrow&\downarrow\\\\
> \mathcal{D} _{\mathcal{Z}} & \mathcal{D} _{\mathcal{P}} & \mathcal{D} _{\mathcal{N}}
> \end{array}
> $$
>
> In other words, it is the joint distribution of the feature $x$ and label $y$.
>
> > **(Q2)** In line 100, it is said that "Note that (3) and (5) are hard to optimize since it is complicated to determine the positive quantile $\eta _{\alpha}(f)$ and the negative quantile $\eta _{\beta}(f)$." This may be a silly question, but can't you just sort to find the largest $n _+^\alpha$ and $n _{-}^\beta$ scores? This would be like $O(nlogn)$ time, which wouldn't be much worse than the $O(n)$ time.
>
> **(A2)**: Thank you so much for your nice question! We agree with you that ranking has a nearly linear complexity. However, the sorting operation is not differentiable since the operation is not continuous. Hence, its convergence guarantee is hard to obtain. Moreover, even if we want to calculate an estimated gradient, we have to use the full-batch data. These two factors together make it almost impossible to optimize the original objective function based on off-the-shelf deep learning tools.
>
> > **(Q3)** Can some intuition be provided behind $F_{op}$; i.e., what roles do all the introduced variables play? Does the equivalence of (9) hold for each $f$? That is, is it true that $\mathcal{R} _\beta(f)=\min _{a,b}\max _\gamma\mathbb{E} _z[F _{op}(...)]$?
>
> **(A3 Part 1)**: The functionality of the reformulation is two-fold:
>
> 1. To reformulate the pairwise objective function into an **instance-wise** form, so that the optimization problem could be solved with the off-the-shelf tools.
>
> 2. To reformulate the partial ranking with a differentiable optimization problem, so that we can calculate the overall gradient.
>
> In this sense, we employ four auxiliary variables in the reformulation, *i.e.*, $a,b,\gamma, s'$. The first three variables are used to convert **pairwise** calculations into **instance-wise** optimization. While $s^{\prime}$ is used to reformulate the partial ranking operation as a differentiable optimization problem. The detailed proofs are shown in **Appendix E.1**. Here, we only present a quick review of the key steps where the auxiliary variables are introduced into the formulation.

---

### Author Response · Authors · 2022-08-09
**New Version of Our Paper Uploaded**

Dear the ACs, and the Reviewers,
Thank you so much for your valuable comments! They really helped us improve our manuscript! We have added all our revisions in the new version of our paper, including both the main file and the appendix in the supplementary materials.

---

> ### Comment · Reviewer_TVKy · 2022-08-09
> **About the new version**
>
> The authors have addressed all my concerns. I'm very appreciated to see the new version with so many revisons done! So, I'm happy to keep my score as a strong accept.

---

### Meta-Review · Area_Chair_cLCb · 2022-08-23

**Recommendation:** Accept
**Confidence:** Certain

**Metareview:**

The paper presented a novel reformulation of maximzing PAUC in an asymptotically unbiased and instance-wise manner. Based on this formulation, the authors presented an efficient stochastic min-max algorithm for  OPAUC and TPAUC maximization.  Convergence and generalization analysis were conducted.  The concerns and questions are well addressed in the rebuttal.  Following the recommendation from the reviewers, I recommend its acceptance.

**Award:**

No

---

### Decision · Program_Chairs · 2022-09-14

Accept